# Toward merging MOPEX and CAMELS hydrometeorological datasets: compatibility and statistical comparison

Katharine Sink, Tom Brikowski

Department of Sustainable Earth Systems Sciences, University of Texas at Dallas, Richardson, 75048, United States

*Correspondence to*: Katharine Sink (katharine.sink@utdallas.edu)

**Abstract.** This study compares two large hydrometeorological datasets, the Model Parameter Estimation Experiment (MOPEX), and the Catchment Attributes and Meteorology for Large-sample Studies (CAMELS), with the aim of quantifying differences that might impact their mergers. This comparison focuses on 47 shared watersheds within the continental United States spanning daily, monthly, seasonal, and annual scales for the overlapping water years of 1981 to 2000. Results indicate significant differences

between the datasets at daily timesteps, highlighting the challenge of high temporal resolution data reconciliation; however, compatibility markedly improves with temporal aggregation at monthly, seasonal, and annual scales. Systematic biases are evident, with MOPEX showing a warm bias for temperature and CAMELS displaying a wet bias for precipitation. For future studies analysing monthly or annual runoff trends, no corrections to the raw data are necessary, as the biases do not significantly affect large-scale temporal analyses. Studies focusing on fine-scale hydrological characteristics, such as daily precipitation events, the

frequency of wet and dry days per month or single-basin dynamics, may require a statistical bias correction to ensure accuracy. Uncertainty is inherent in all climate datasets due to differences in data sources, interpolation methods, and spatial coverage. The transition from MOPEX to CAMELS does not notably introduce additional uncertainty beyond what is already present in the original datasets. The variability between the datasets is comparable to the inherent variability within each individual dataset and is neither a useful criterion for dataset selection nor a barrier to potential merger. As a result, the overall uncertainty in annual or

decadal modeling outcomes remains essentially the same, regardless of which dataset is used. That said, model outputs should be calibrated against observational reference data to account for systematic errors. Statistical analyses demonstrate that both datasets are representative of climatic conditions, trends, and extreme events. Our findings validate the results of previous research employing either dataset. Furthermore, this study serves as a foundation for the merging and extension of MOPEX and CAMELS datasets without any alterations, providing a comprehensive, long-term dataset suitable for hydrological modeling and climate

analyses while maintaining comparability across basin and temporal scales.

## 1 Introduction

Comprehensive historical datasets are crucial for investigating and projecting surface water availability given the complex response of watersheds to natural and anthropogenic forcings. In particular, comparative and large sample hydrology (LSH) rely on large datasets comprised of numerous catchments to derive relationships, develop new models and uncertainty estimates, and classify

locations that span different climatic and physiographic regions (Addor et al., 2020; Gupta et al., 2014), yet significant discrepancies make combining and comparing such datasets difficult. Indeed Addor et al. (2020) state the "lack of common standards impedes the comparison of basins from different datasets".

This paper explores and attempts to resolve the principal issues confronting the merger of two of the most commonly used LSH datasets for the continental United States, (CONUS), the Model Parameter Estimation Experiment, MOPEX (Duan et al., 2006; Schaake et al., 2006), and the Catchments Attributes and Meteorology for Large-sample Studies, CAMELS (Addor et al., 2017; Newman et al., 2015). In general, there is an abundance of data available for climate variables, streamflow, and catchment characteristics, including ground and remote-sensed parameters; however, varying spatial and temporal resolutions among variables such as precipitation and temperature often hinder intercomparison and merging of datasets (Guo, 2017). A wide range of data sources with varying analysis and derivation methods can introduce uncertainty, especially when metadata (Kelleher and Braswell, 2021) or uncharacterized anthropogenic influences are excluded (Addor et al., 2020).

MOPEX and CAMELS are two prominent datasets that encompass a combination of daily temperature, precipitation, potential evapotranspiration, and streamflow values for selected catchments. Additionally, these datasets provide essential catchment characteristics such as area, elevation, vegetation, and soil texture, employing the United States Geological Survey (USGS) hydrologic unit code (HUC) subbasin classification (Seaber et al., 1987). While the consolidation of attributes and hydroclimatic data simplifies the acquisition process, challenges arise due to differences in spatial coverage and data sources, which currently limit the opportunity to effectively utilize both the MOPEX and CAMELS datasets simultaneously or confirm findings and expand on studies employing either dataset.

Researchers often face the necessity of choosing one data set over the other, leading to a situation where the unique strengths and limitations of each data set influence the selection process. Numerous studies have engaged in the generalization and categorization of watersheds within the CONUS using either the MOPEX or CAMELS dataset, which underscores the widespread impact and influence of these two large-sample datasets, making them arguably the most prolific resources within hydrological studies focused on the CONUS. Their prevalence in hydrologic studies is reflected in the citation counts data derived from *Clarivate Web of Science* (Clarivate, 2024); with MOPEX (Duan et al., 2006) currently cited in 489 scientific papers and CAMELS (Addor et al., 2017) cited in 352. Here we undertake a unique comparative study between the MOPEX and CAMELS datasets, using exploratory data analysis to evaluate their comparability, accuracy, and implications for past, present, and future research. The results aim to bolster confidence in analytical and modeling outcomes derived from either dataset, thereby fostering robust hydrological research, and supporting effective water resource management in the CONUS.

This study compares daily precipitation and temperature data derived from land surface stations across the country. MOPEX includes data for 431 watersheds from 1948 to 2003 and CAMELS covers 671 basins from 1980 to 2014. There are 52 overlapping basins between the two datasets. This evaluation is conducted over water years common to both datasets (1981–2000), emphasizing 47 common subbasins. Many previous dataset comparison studies have addressed global climate datasets (Essou et al., 2016; Newman et al., 2019), precipitation (Buban et al., 2020; Levy et al., 2017; Muche et al., 2020; Prat and Nelson, 2015; Sitterson et al., 2020; Sun et al., 2018), temperature (Oubeidillah et al., 2014), and evapotranspiration products (Carter et al., 2018; Chao et al., 2021; Han et al., 2015). These studies contribute to the ongoing efforts to advance the understanding of hydrological processes and improve the reliability of hydrologic models (Gupta et al., 2014); however, there has yet to be a study comparing these two large sample watershed-based datasets. Our findings show that while MOPEX and CAMELS exhibit systematic biases, they can still be merged or reliably compared without requiring corrections beyond smaller time scales (i.e. a single day, month, or season). Statistical adjustments to daily data depend on study objectives, as no single method fits all needs. Raw data or direct model outputs typically require bias correction, and we intend for our results to help researchers determine necessary adjustments using appropriate methods, such as equidistant quantile matching (EDCDFm) for

temperature and quantile delta mapping (QDM) or PresRATe for precipitation (Lehner et al., 2023; Pierce et al., 2015). To support long-term hydrological analyses, all basins will be extended to 2023 using Daymet, yielding a combined dataset of up to 1,050 basins (in progress, Sink et al., 2025). When MOPEX is extended using Daymet, slight shifts in these biases are expected, but the dataset's overall reliability remains intact.

## 2 Hydrometeorological datasets

### 2.1 MOPEX

The MOPEX intercomparison project was conceived by several organizations including the World Meteorological Organization (WMO), the International Association of Hydrogeologists (IAH) Predictions in Ungauged Basins (PUB) initiative, and the Global Energy and Water Cycle Experiment (GEWEX) in 1996 (Duan et al., 2006). Its aim was to establish guidelines for parameter estimation techniques while simultaneously decreasing uncertainty (Schaake et al., 2006). MOPEX contains precipitation, minimum and maximum temperature, and streamflow data for 431 CONUS basins on a daily time step for 1948–2003. MOPEX variables are based on weather station observations from the National Climatic Data Center (NCDC) and Natural Resources Conservation Service (NRCS) SNOTEL network, which were then averaged by catchment area using an inverse distance weighting method. For more details regarding data selection and processing, refer to Duan et al. (2006) and Schaake et al. (2006).

### 2.2 CAMELS

CAMELS, sponsored by the US Bureau of Reclamation and the US Army Corps of Engineers, consists of three daily forcing data sets from Daymet Version 2 (Thornton et al., 2012), Maurer (Maurer et al., 2002), and North American Land Data Assimilation System (NLDAS), (Xia et al., 2012) along with benchmark model performance results using the coupled Snow-17 snow model and the Sacramento Soil Moisture Accounting Model (SAC-SMA), using each of the three forcing datasets, for 671 basins within the CONUS covering the years 1980–2014 (Newman et al., 2015). CAMELS contains precipitation, temperature, and streamflow data on daily time steps in addition to detailed soil characterizations and geology. The CAMELS Daymet Version 2 forcing dataset is used in this study and interpolates observations to a 1 km x 1 km grid using a Gaussian weighting process (Thornton et al., 2021), which are simply averaged over the catchment area in CAMELS. For an in-depth discussion regarding data selection and processing for CAMELS, refer to Addor et al. (2017) and Newman et al. (2015).

### 2.3 Dataset comparison

Both datasets selected basins with apparently minimal anthropogenic impacts, highlight processing methods, and provide access to basin characteristics including boundary files, fractional spatial coverage of soil type, vegetation type, land cover, area, and elevation (Table 1). The documentation of catchment attributes, along with daily data for streamflow, temperature, precipitation, and potential evapotranspiration, significantly streamlines the initial phases of data investigation, consolidation, and processing, making the datasets exceptionally valuable for research and analysis.

**Table 1. Comparisons between MOPEX and CAMELS. Acronyms are Hydro-Climatic Data Network (HCDN), National Climatic Data Center (NCDC), Cooperative Observer Program (COOP), Snow Telemetry Network (SNOTEL), Parameter-elevation Regressions on Independent Slopes Model (PRISM), North American Land Data Assimilation System (NLDAS), National Oceanic and Atmospheric Administration (NOAA), Sacramento Soil Moisture Accounting Model (SAC-SMA), State Soil Geographic (STATSGO) database, Global Lithological Map (GLiM), Global Hydrogeology Maps (GLHYMPS) of permeability and porosity, Moderate Resolution Imaging Spectroradiometer (MODIS), International Geosphere-Biosphere Programme (IGBP), University of Maryland (UMD), Normalized Difference Vegetation Index (NDVI).**

| Characteristic | MOPEX | CAMELS |
|---|---|---|
| Basins | 431 | 671 |
| Temporal coverage | 1948–2003 | 1980–2014 |
| Streamflow (daily) | USGS HCDN (Slack and Landwehr, 1992) | USGS HCDN-2009 (Lins, 2012) |
| Precipitation (daily) | NCDC COOP, SNOTEL | Daymet, Maurer, NLDAS |
| Temperature (daily) | NCDC COOP, SNOTEL | Daymet, Maurer, NLDAS |
| Potential Evapotranspiration | NOAA (Farnsworth et al., 1982) | Priestly-Taylor |
| Actual Evapotranspiration | | SAC-SMA model |
| Soil Properties | STATSGO (Miller and White, 1998) | STATSGO, Pelletier et al. (2016) |
| Geology | | GLiM, GLHYMPS |
| Greenness Fraction (NDVI) | NLDAS | MODIS |
| Vegetation Type | IGBP, UMD | MODIS |

For this study, temperature and precipitation values from the datasets were evaluated on daily, monthly, seasonal, and annual temporal scales between 1981 and 2000, based on water years spanning 1 October 1980 to 30 September 2000. Derived variables were omitted for most analyses in this study because evapotranspiration, when calculated using the water balance, will only differ based on the precipitation since both MOPEX and CAMELS obtain the other balance component, streamflow, from the USGS National Water Information System (NWIS). Potential evapotranspiration values are highly dependent on the estimation method used and require additional information such as wind speed, solar radiation, and temperature (Andréassian et al., 2004; Lemaitre-Basset et al., 2022; Pimentel et al., 2023). Potential evapotranspiration values can also be estimated during modeling.

This study provides researchers with detailed analyses regarding the uncertainties within the datasets and between them for a 20-year period through quantitative measurements of dispersion, distribution, central tendency, interval estimates, and statistical tests. To obtain a longer record, the datasets will be extended to the present using Daymet (Thornton et al., 2021) and the results from this study can be applied to additional basins by climate region. The merged MOPEX and CAMELS datasets will incorporate up to 1,050 watersheds, temporally extended from 1948 to 2023. (in progress, Sink et al., 2025).

## 2.4 Study Area

MOPEX contains 431 catchments and CAMELS contains 671 (red and blue points, respectively, Fig. 1) within the CONUS. The spatial coverage differs between the two with CAMELS deliberately incorporating more basins within the Great Plains and southwestern US (Addor et al., 2017; Newman et al., 2015). Each catchment is identified based on the USGS NWIS stream gauge identification number (Table 2), representing its downstream outlet.

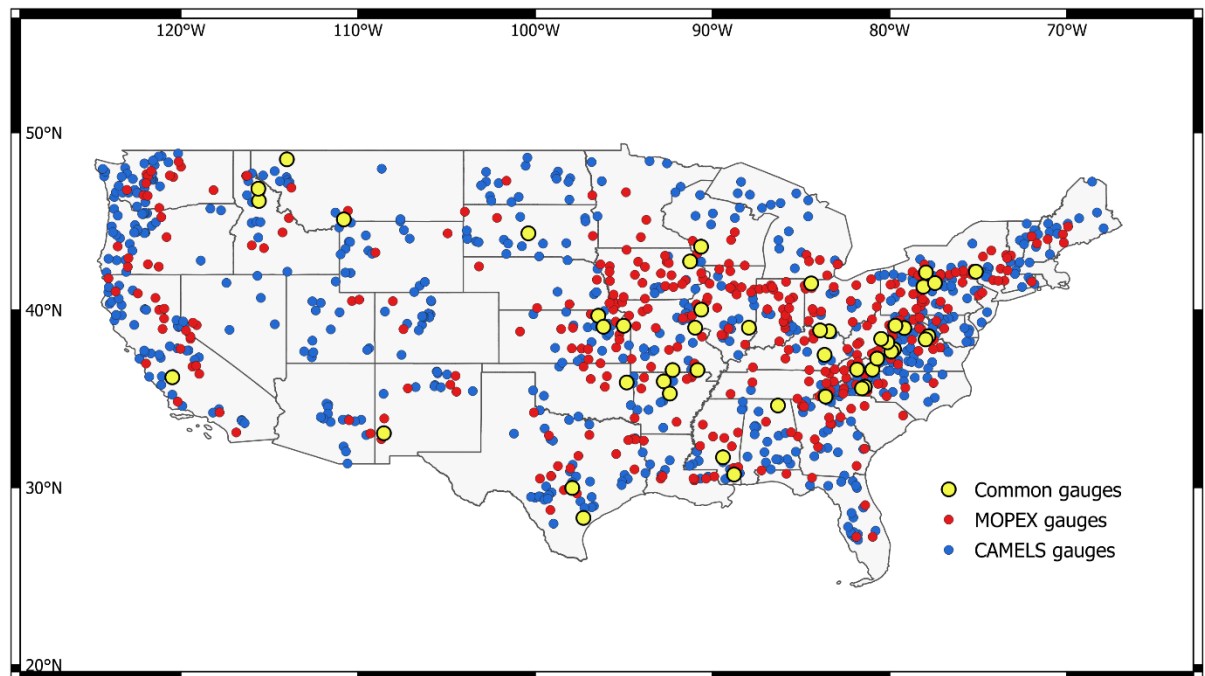

**Figure 1. Locations of 431 USGS NWIS stream gauges in MOPEX (red points), 671 gauges in CAMELS datasets (blue points), and common gauges (yellow points) within the CONUS that appear in both the MOPEX and CAMELS datasets.**

The datasets have 52 basins in common, 47 of which were used in this study (yellow points, Fig. 1). Five watersheds were omitted

from this study because of incomplete streamflow records, or the gauge catchment was only a portion of the watershed. Catchment

climate variables precipitation (PRCP) and temperature (TAIR) data were area weighted (average of observations values over the

area of the basin) using the Hydro-Climatic Data Network (HCDN) basin delineations (Slack and Landwehr, 1994)

**Table 2. Common watersheds between MOPEX and CAMELS. Basins are described by GaugeID (NWIS identification) along with the station name, location (city, state), latitude (decimal degrees), longitude (decimal degrees), elevation (meters), area (square kilometers)**
**and climate. Basins are grouped by climate type and then sorted by increasing gauge identification number.**

| GaugeID | Station Name | Location | Latitude | Longitude | Elevation (m) | Area (km²) | Climate |
|---|---|---|---|---|---|---|---|
| 06441500 | Bad River | Fort Pierre, SD | 44.33 | -100.38 | 683.42 | 8,152.55 | Arid |
| 08171300 | Blanco River | Kyle, TX | 29.98 | -97.91 | 379.23 | 1,067.47 | Arid |
| 08189500 | Mission River | Refugio, TX | 28.29 | -97.28 | 67.31 | 1,808.29 | Arid |
| 09430500 | Gila River | Gila, NM | 33.06 | -108.54 | 2,227.36 | 4,804.93 | Arid |
| 11224500 | Los Gatos Creek | Coalinga, CA | 36.21 | -120.47 | 658.03 | 247.44 | Arid |
| 01664000 | Rappahannock River | Remington, VA | 38.53 | -77.81 | 216.10 | 1,605.10 | Continental |
| 01667500 | Rapidan River | Culpepper, VA | 38.35 | -77.98 | 193.47 | 1,209.75 | Continental |
| 02016000 | Cowpasture River | Clifton Forge, VA | 37.79 | -79.76 | 645.04 | 1,194.55 | Continental |
| 02018000 | Craig Creek | Parr, VA | 37.67 | -79.91 | 648.68 | 852.34 | Continental |
| 03173000 | Walker Creek | Bane, VA | 37.27 | -80.71 | 750.95 | 773.32 | Continental |
| 03237500 | Ohio Brush Creek | West Union, OH | 38.80 | -83.42 | 272.36 | 1,003.21 | Continental |
| 03238500 | White Oak Creek | Georgetown, OH | 38.86 | -83.93 | 285.37 | 568.50 | Continental |
| 03346000 | North Fork Embarras River | Oblong, IL | 39.01 | -87.95 | 173.50 | 814.69 | Continental |
| 04185000 | Tiffin River | Stryker, OH | 41.50 | -84.43 | 250.64 | 1,064.00 | Continental |
| 05408000 | Kickapoo River | La Farge, WI | 43.57 | -90.64 | 348.35 | 689.33 | Continental |
| 05412500 | Turkey River | Garber, IA | 42.74 | -91.26 | 327.65 | 3,858.21 | Continental |

| GaugeID | Station Name | Location | Latitude | Longitude | Elevation (m) | Area (km$^2$) | Climate |
|---------|--------------|----------|----------|-----------|---------------|-----------|---------|
| 05514500 | Cuivre River | Troy, MO | 39.01 | -90.98 | 226.22 | 2,407.41 | Continental |
| 05585000 | La Moine River | Ripley, IL | 40.02 | -90.63 | 197.96 | 3,354.61 | Continental |
| 06191500 | Yellowstone River | Corwin Springs, MT | 45.11 | -110.79 | 2,547.95 | 6,783.59 | Continental |
| 06885500 | Black Vermillion River | Frankfort, KS | 39.68 | -96.44 | 394.81 | 1,062.87 | Continental |
| 06888500 | Mill Creek | Paxico, KS | 39.06 | -96.15 | 412.34 | 842.35 | Continental |
| 06892000 | Stranger Creek | Tonganoxie, KS | 39.12 | -95.01 | 304.55 | 1,092.72 | Continental |
| 07057500 | North Fork River | Tecumseh, MO | 36.62 | -92.25 | 324.68 | 1,456.44 | Continental |
| 01423000 | West Branch Delaware River | Walton, NY | 42.17 | -75.14 | 593.67 | 859.68 | Temperate |
| 01543500 | Sinnemahoning Creek | Sinnemahoning, PA | 41.32 | -78.10 | 547.44 | 1,778.26 | Temperate |
| 01548500 | Pine Creek | Cedar Run, PA | 41.52 | -77.45 | 546.71 | 1,557.05 | Temperate |
| 01606500 | South Branch Potomac River | Petersburg, WV | 38.99 | -79.18 | 836.38 | 1,684.55 | Temperate |
| 02143000 | Henry Fork | Henry River, NC | 35.68 | -81.40 | 399.35 | 216.67 | Temperate |
| 02143040 | Jacob Fork | Ramsey, NC | 35.59 | -81.57 | 411.33 | 66.48 | Temperate |
| 02472000 | Leaf River | Collins, MS | 31.71 | -89.41 | 123.24 | 1,927.13 | Temperate |
| 02479300 | Red Creek | Vestry, MS | 30.74 | -88.78 | 65.19 | 1,144.20 | Temperate |
| 03069500 | Cheat River | Parsons, WV | 39.12 | -79.68 | 961.46 | 1,856.85 | Temperate |
| 03164000 | New River | Galax, VA | 36.65 | -80.98 | 766.60 | 2,952.74 | Temperate |
| 03182500 | Greenbrier River | Buckeye, WV | 38.19 | -80.13 | 934.51 | 1,364.97 | Temperate |
| 03186500 | Williams River | Dyer, WV | 38.38 | -80.48 | 1,057.61 | 329.68 | Temperate |
| 03281500 | South Fork Kentucky River | Booneville, KY | 37.48 | -83.68 | 376.49 | 1,838.22 | Temperate |
| 03473000 | South Fork Holston River | Damascus, VA | 36.65 | -81.84 | 916.29 | 784.81 | Temperate |
| 03504000 | Nantahala River | Rainbow Springs, NC | 35.13 | -83.62 | 1,039.71 | 134.52 | Temperate |
| 03574500 | Paint Rock River | Woodville, AL | 34.62 | -86.31 | 337.61 | 813.80 | Temperate |
| 04221000 | Genesee River | Wellsville, NY | 42.12 | -77.96 | 658.41 | 750.88 | Temperate |
| 07056000 | Buffalo River | St. Joe, AR | 35.98 | -92.75 | 459.08 | 2,149.36 | Temperate |
| 07068000 | Current River | Doniphan, MO | 36.62 | -90.85 | 293.50 | 5,318.59 | Temperate |
| 07197000 | Baron Fork | Eldon, OK | 35.92 | -94.84 | 348.86 | 808.45 | Temperate |
| 07261000 | Cadron Creek | Guy, AR | 35.30 | -92.40 | 197.55 | 445.81 | Temperate |
| 12358500 | Middle Fork Flathead River | West Glacier, MT | 48.50 | -114.01 | 1,559.24 | 2,939.19 | Temperate |
| 13337000 | Lochsa River | Lowell, ID | 46.15 | -115.59 | 1,548.18 | 3,053.42 | Temperate |
| 13340600 | North Fork Clearwater River | Canyon Ranger Station, ID | 46.84 | -115.62 | 1,417.79 | 3,354.62 | Temperate |

## 3 Methodology

### 3.1 Climate characterization of the watersheds

Understanding how catchments partition annual precipitation into runoff and evapotranspiration under varying climatic conditions is crucial for hydrological modeling and water resource management. The Budyko function describes the long-term water and energy balance using annual evaporative (evapotranspiration/precipitation) and aridity (potential evapotranspiration/precipitation) indices (Budyko, 1974). The annual indices were determined for both datasets and subsequently combined during KMeans clustering to obtain the overall climate representation for each basin. KMeans clustering, an unsupervised machine learning algorithm, that seeks to minimize the within cluster sum of squares (Hartigan and Wong, 1979), was utilized to divide the 47 selected MOPEX-CAMELS shared basins into three climate groups based on their annual evaporative and aridity indices, with a classification accuracy of 84 %. The arid (aridity index > 1.5), continental (aridity index 1.5 to 0.82), and temperate (aridity index < 0.82) zones represent the three KMeans groups. For this study, the basin climate region classifications (arid, continental,

temperate) are based on the KMeans clustering results, which agree closely (but not perfectly) with the Köppen-Geiger (Beck et al., 2018)climate classification (Fig. 2).

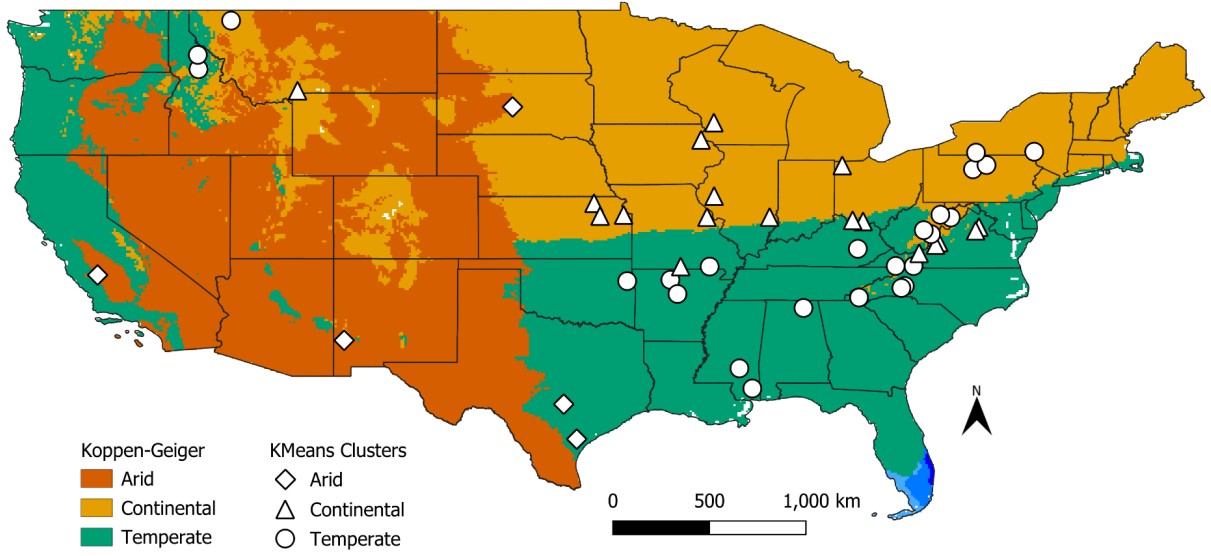

**Figure 2. Regions of the CONUS divided into Köppen Geiger climate classification (arid, continental, temperate) are represented by vermillion, orange, and bluish green respectively. The blue colors in southern Florida represent regions within the tropical climate group, which is not represented in this study. The results of the KMeans clustering are based on the annual aridity and evaporative indices for MOPEX-CAMELS shared basins shown by point symbols (diamond, triangle, circle). Climate groupings for analyses are represented by the KMeans clusters which agree closely but imperfectly with the Köppen-Geiger classification.**

Terrestrial evapotranspiration is difficult to measure directly but can be evaluated using lysimeters or eddy covariance towers on small, local scales. ET can be estimated on a larger scale using satellite remote sensing or land surface models but carry with them inherent biases due to varying algorithms, spatial resolutions, calibration, and input data (Long et al., 2014). Many studies have shown that derived ET products fail to reconcile the terrestrial water budget on multiple temporal scales (Carter et al., 2018). A water balance approach is commonly used on a catchment scale, with observed streamflow obtained from a measured outlet (Han et al., 2015). A water balance sets ET (mm) equal to the precipitation (mm) minus basin runoff (mm), with water storage assumed to be zero on an annual scale.

The MOPEX dataset does not contain daily ET. Studies that have made use of MOPEX data obtain ET via the water balance approach using the precipitation and observed runoff (Berghuijs et al., 2014; Coopersmith et al., 2012; Sawicz et al., 2014). As mentioned previously, CAMELS provides three different daily forcing datasets (Daymet, Maurer, NLDAS), which do not contain ET, in addition to three Sacramento Soil Moisture and Accounting Model (SAC-SMA) generated time series from each of the forcing datasets. Daily ET values from the model output time series using Daymet forcing variables (CAMELS SAC-SMA) were compared to the water balance derived ET using CAMELS catchment averaged Daymet precipitation and USGS runoff values (CAMELS WB) to evaluate any notable differences between methods and facilitate comparison of MOPEX and CAMELS. The CAMELS SAC-SMA model derived ET values are typically greater than the values derived from the CAMELS WB, which will become more prominent at an annual scale, as plotted in Fig. 3.

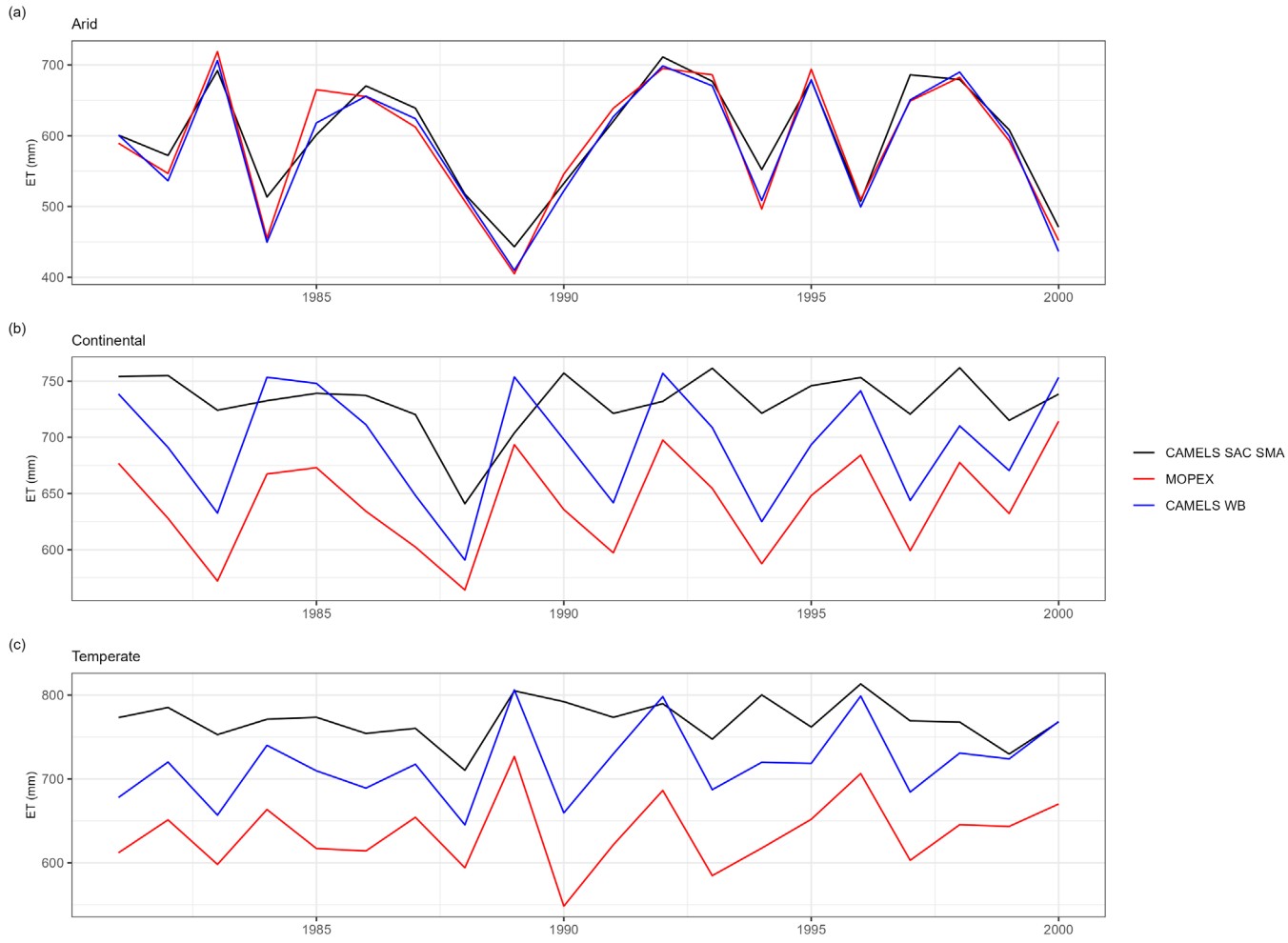

**Figure 3. Total annual evapotranspiration for a) arid, b) continental, and c) temperate regions. The annual values are the overall mean of all basin totals in a region. The model output ET (CAMELS SAC-SMA), water balance derived MOPEX ET (MOPEX), and water balance derived CAMELS ET (CAMELS WB) are shown in each plot.**

When annual differences between CAMELS SAC-SMA model estimated ET and CAMELS WB estimated ET are averaged, SAC-SMA model estimations are approximately 13 mm larger in arid regions (Fig. 3a), 36 mm larger in continental regions (Fig. 3b), and 50 mm larger in temperate regions (Fig. 3c). Higher ET values lead to reduced runoff. As shown in Fig. 4, estimated ET values from CAMELS SAC-SMA model were subtracted from the provided CAMELS (Daymet) precipitation data to calculate estimated runoff (SAC_RUN), which was then compared to observed runoff (OBS_RUN). Incorporating ET values from the model output time series as an input variable to a hydrologic model may result in slightly lower discharge estimates, primarily reflecting the influence of ET values rather than actual runoff conditions.

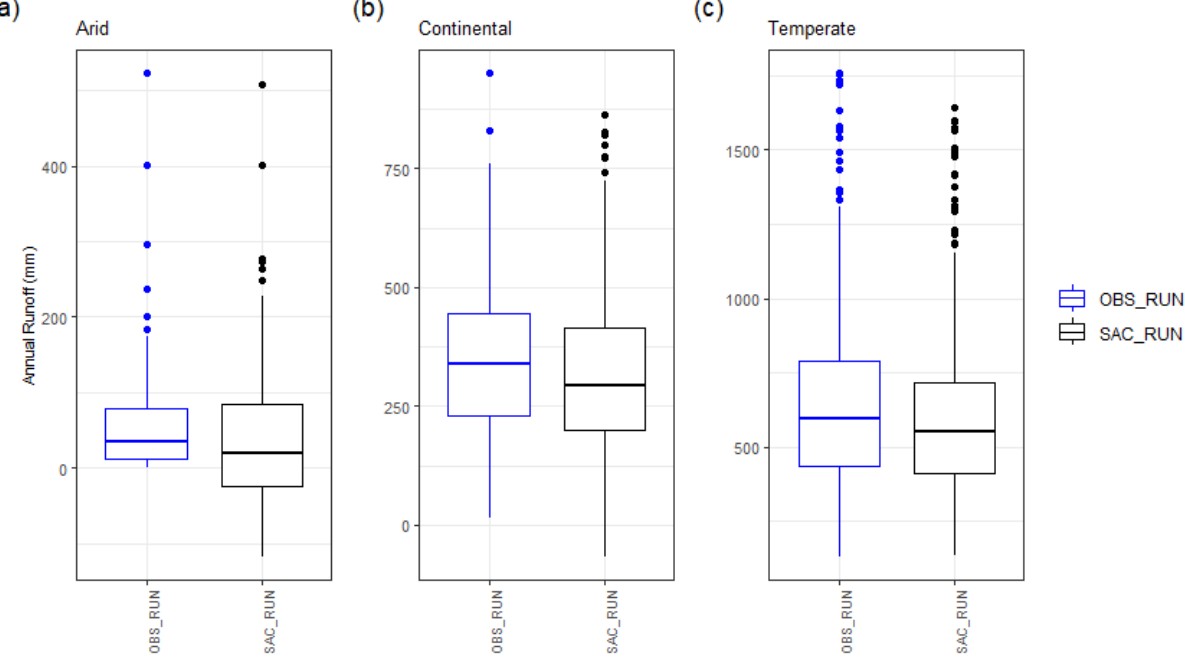

Figure 4. Total annual runoff for a) arid, b) continental, and c) temperate regions. The boxplots represent the annual totals for all basins in a region, with measured observed runoff (OBS_RUN) and water balance calculated runoff (PRCP minus ET) using CAMELS SAC-SMA model output ET (SAC_RUN).

A Budyko diagram, plotting evaporative versus aridity indices, clarifies the predominant hydrologic processes versus climate type (Fig. 5) for the common basins. The CAMELS SAC-SMA evapotranspiration (solid symbols, Fig. 5) exhibits large discrepancies from CAMELS WB (open symbols, Fig. 5) and MOPEX ET values for most catchments (arrows, Fig. 5). Furthermore, several CAMELS SAC-SMA gauges plotted above the water limit (i.e. to extreme values in the Budyko context) and were 10 to 12 % larger than the water-balance-calculated evapotranspiration indices. The higher model-derived ET for CAMELS could reflect additional non-precipitation sources of water to the catchment, but that was not evaluated in this study. The largest discrepancies between model-derived ET/P and water balance derived ET/P for CAMELS for each climate region are 46 %, 12 %, and 60 % for arid, continental, and temperate regions respectively. Average discrepancies between CAMELS evapotranspiration values are largest in arid regions, 12 %, followed by average discrepancies of 11 % in temperate regions, and 5 % in continental regions.

Differences between water-balance calculated ET for MOPEX versus CAMELS vary by climate type and may partly result from variations in sample distribution. Most of the shared watersheds fall into temperate and continental climates, but the western US is not as heavily represented based on the distribution of the catchments and the restriction to shared basins. Only eight shared catchments lie west of the hundredth meridian (Fig. 1). The arid region basins lie close to the water limit (ET/P = 1, Fig. 5), while the temperate region basins are close to the energy limit (PET/P = 1). The continental region catchments can be seen as a transitional climate which can be either energy or water limited. Annual MOPEX and CAMELS evaporative and aridity indices are plotted separately to highlight the improvements when utilizing the water balance evapotranspiration values for CAMELS. The largest difference between MOPEX and CAMELS evaporative indices using evapotranspiration water balance values is 16.2 % in temperate regions with an overall average difference of 4.6 % for all 47 basins. The mean difference between MOPEX and

CAMELS evaporative indices, with water balance calculated evapotranspiration, is 1.89 %, 2.53 %, and 7.14 % for arid, continental, and temperate regions respectively.

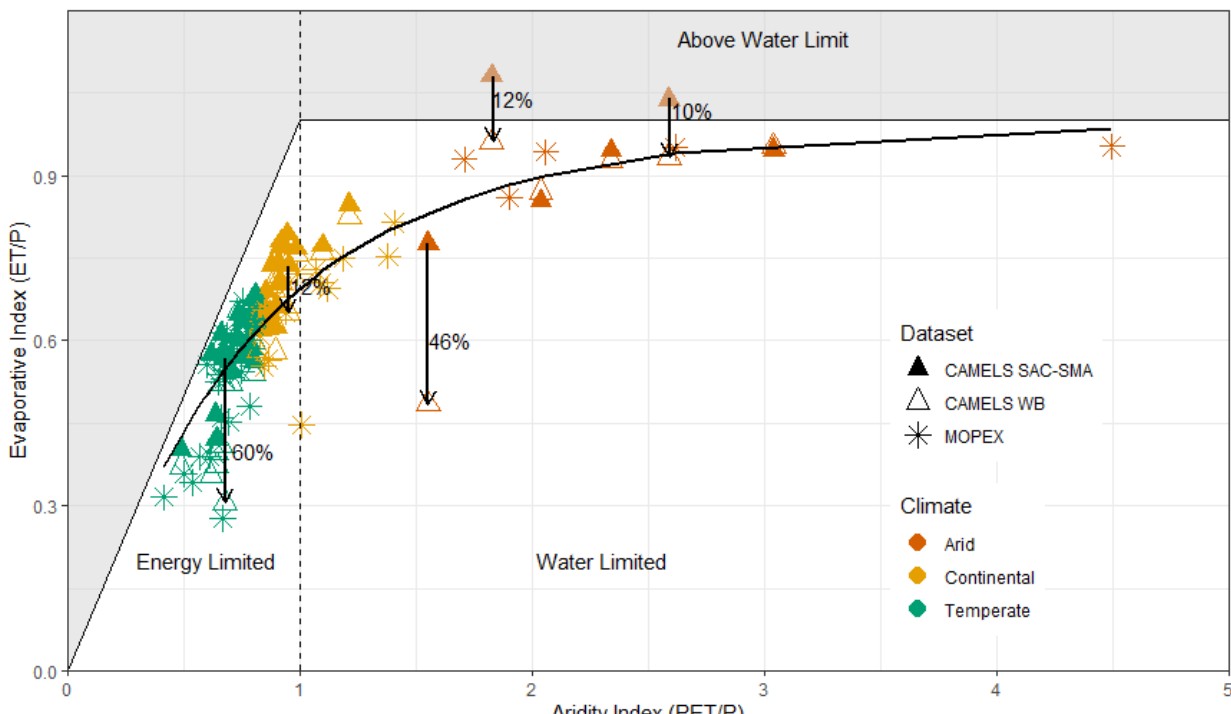

**Figure 5. Budyko diagram with the aridity and evaporative indices plotted for each of the 47 catchments (1981-2000). The overall aridity index and evaporative index is plotted for each catchment for the three ET values, resulting in 141 points. The three ET values include MOPEX (asterisk), CAMELS (solid triangle) with SAC-SMA derived ET, and CAMELS WB (open triangle) with water balance calculated evapotranspiration. Evaporative index values > 1 are non-physical.**

Further research using the CAMELS dataset should apply the water balance approach instead of SAC-SMA derived ET to avoid decreased runoff and vertical displacement in Budyko space that are artifacts of model derived ET values. The water balance ET values were calculated using precipitation that does not include snowmelt; however, most of the larger discrepancies are present in arid regions (vermillion, Fig. 5) where snowmelt is negligible.

**3.2 Exploratory data analysis**

All statistical analyses were conducted using R Statistical Software (v. 4.3.3; R Core Team, 2024). When basins are consolidated by climate region, the number of values used in calculations are dependent on the number of gauges unless otherwise specified (Table 3). Each gauge has 7,305 daily observations beginning on 1 October 1980 and ending on 30 September 2000. Monthly values are based on water years which begin in October of the previous calendar year and end in September of the current calendar

240    year. Seasons are winter (December, January, February), spring (March, April, May), summer (June, July, August), and fall (September, October, November) and the months are grouped by water year, resulting in all four seasons within each water year.

**Table 3. Number of observations used for various statistical analyses on temporal scales per dataset.**

| Time | Range | Per Gauge | Arid (5 gauges) | Continental (18 gauges) | Temperate (24 gauges) |
|---|---|---|---|---|---|
| Days | 1 October 1980–30 September 2000 | 7,305 | 36,525 | 131,490 | 175,320 |
| Months | October–September | 240 | 1,200 | 4,320 | 5,760 |
| Seasons | Fall 1981–Fall 2000 | 80 | 400 | 1,440 | 1,920 |
| Water Years | 1981–2000 | 20 | 100 | 360 | 480 |

### 3.2.1 Uncertainty and variability within datasets

The central tendency (mean, median), variability (variance, standard deviation, coefficient of variation), and distribution (skewness) of precipitation and temperature were independently evaluated for MOPEX and CAMELS. Uncertainty for the mean value was determined using two-sided confidence intervals computed via the bootstrap method. Bootstrapping is a statistical technique that estimates the sampling distribution of a statistic by iteratively resampling, with replacement, from the observed data when the population or sample distribution is unknown (Helsel et al., 2020). This nonparametric method utilizes the observed data to derive the robust estimates and sampling distributions. In this study, bootstrapping was implemented using the *Hmisc* R package (Harrell Jr, 2024) to calculate the mean value for daily, monthly, seasonal, and annual precipitation and temperature for MOPEX and CAMELS, separately. Analyses involved 10,000 resamples, and the two-sided 95-percent confidence intervals were determined by the 0.025 and 0.975 quantiles. This approach provides a robust method for estimating the uncertainty and variability associated with the mean values on different temporal scales.

### 3.2.2 Uncertainty and variability between datasets

Several hypothesis tests were conducted to compare observations between MOPEX and CAMELS. The nonparametric (binomial) sign test is used to compare two groups and assess whether one group is consistently higher than the other (Helsel et al., 2020). For a two-sided test, the null hypothesis posits that about half of the differences will be positive and half will be negative, resulting in a median difference of zero between paired observations. For context, paired observations compare the same date (day), month, season, or water year from each dataset. To conduct this test, MOPEX values were subtracted from CAMELS values, where a positive (negative) difference indicates that the CAMELS value is greater than (less than) the MOPEX value, with no consideration for the magnitude. The differences were computed for daily (7,305 pairs), monthly (240 pairs), seasonal (80 pairs), and annual (20 pairs) precipitation and temperature values for each basin ("Per Gauge" column, Table 3). Given that temperature may include negative instances, strict inequalities were applied. Subsequently, the outcomes were assigned a positive ($n_+$), negative ($n_-$), or zero value, and the values were tallied. A binomial distribution was used to calculate the probability of observing a value of n, which is 0.5. A 95 % confidence interval results in a significance level of $p<0.05$. Hypothesis testing and significance make use of the *rstatix* (Kassambara, 2023) and *stats* R packages.

Independent difference hypothesis tests included the Fligner-Killeen test (Fligner and Killeen, 1976) and the t-test. The non-parametric Fligner-Killeen test was conducted to check whether MOPEX and CAMELS have equal variances, with the null hypothesis assuming variances are equal across all samples. It is less sensitive to departures from normality compared to the Bartlett and Levene tests (Helsel et al., 2020). The absolute value of the residuals (AVR) is calculated in Eq. 1 from each group median for $j=1$ to $k$ groups and $i=1$ to $n_j$ observations where

$$AVR_{ij} = \left| x_{ij} - median_j \right| \tag{1}$$

The AVR is ranked and weighted, resulting in a set of scores. A linear-rank test is then computed on the set of scores (Helsel et al., 2020).

Welch's t-test (Welch, 1951) is a modification of the Student's t-test that does not assume equal variance. The null hypothesis posits that the two group means are identical. The test statistic, $t$, is calculated as shown in Eq. 2

$$t = \frac{m_A - m_B}{\sqrt{\frac{S_A^2}{n_A} + \frac{S_B^2}{n_B}}} \qquad (2)$$

Where $S_A$ and $S_B$ are the standard deviation of the two groups A and B, along with the means $m_A$ and $m_B$. And the degrees of freedom, $df$, is calculated as shown in Eq. 3

$$df = \frac{\left(\frac{S_A^2}{n_A} + \frac{S_B^2}{n_B}\right)^2}{\left(\frac{S_A^4}{n_A^2(n_A-1)} + \frac{S_B^4}{n_B^2(n_B-1)}\right)} \qquad (3)$$

Statistical significance for the Fligner-Killeen test and t-test are based on a p-value less than 0.05.

Bias, the mean absolute error (MAE), and standard error (SE) were also used to assess the variability within each group. The standard error provides an estimate of the standard deviation of the sampling distribution of the difference between means. The margin of error (MOE) was also determined based on a 95-percent confidence interval with a critical value ($\alpha$) of 1.96. The critical value is multiplied by the standard error of the difference of the means, which provides the confidence interval for the true difference between the means. The nonparametric Spearman rank correlation coefficient ($\rho$) was also employed to assess the strength of association between variables. This method is robust to the distribution of data and is less influenced by outliers.

**3.3 Validation**

Machine learning (ML) techniques, such as linear regression, random forest, gradient boosting, and support vector regression, offer a valuable alternative to physically based models by capturing relationships between input and output variables. While they do not rely on detailed hydrological processes, these models can still provide robust predictions and allow for comparative analysis of different datasets (Herrera et al., 2022). Using ML models as a proxy is increasingly common in hydrological research, as these models can efficiently handle high-dimensional data and learn intricate patterns without explicitly modeling physical processes
(Kratzert et al., 2019). ML models have been shown to perform well in a range of hydrological applications, especially in data-rich contexts. For this study, we employed ML models to evaluate the potential influences of MOPEX and CAMELS precipitation and temperature biases on predicted runoff.

Hydrologic models rely on parameterization and assumptions about physical processes while ML models learn directly from data,
reducing dependence on prior assumptions and allowing for a purely data-driven evaluation (Nearing et al., 2021). ML models can highlight inconsistencies or biases in input datasets by comparing their predictive performance across datasets. If one dataset consistently leads to better predictions, it may indicate better representativeness or higher quality. Traditional hydrologic models typically require extensive calibration and long run times, especially for larger scale applications, but ML models, once trained, can make predictions rapidly and do not require manual calibration (Kratzert et al., 2019). ML models can also be trained separately
on different temporal scales, allowing for direct comparisons without modifying model structures. By evaluating performance

metrics across datasets, ML provides an objective assessment of whether precipitation and temperature inputs are sufficient to capture runoff variability (Yokoo et al., 2022).

Four different ML models were implemented in R to estimate runoff from precipitation and mean air temperature using the *e1071* (Meyer et al., 2024), *gbm* (Ridgeway et al., 2024), *randomForest* (Breiman et al., 2024), and *caret* (Kuhn et al., 2024) packages. Linear regression models the relationship between a dependent variable and one or more independent variables by fitting a linear equation (Xu and Liang, 2021). Random forest is an ensemble learning method that constructs multiple decision trees and averages their predictions to improve accuracy and reduce overfitting (Breiman, 2001). Gradient boosting builds models sequentially, optimizing for errors in previous iterations by combining weak learners to create a stronger predictive model (Xu and Liang, 2021). Support vector regression (SVR) maps input data into a higher-dimensional space and finds the ideal hyperplane, separating the data points into different classes, and minimizes prediction error while maintaining generalization (Shmilovici, 2023). These models provide a diverse approach to estimating runoff, ranging from simple linear relationships to more complex, non-linear learning techniques.

MOPEX and CAMELS precipitation and temperature values were used as input to predict runoff at daily, monthly, seasonal, and annual time scales. Precipitation and temperature data were transformed into common scales using min-max normalization. Datasets were then split into training and test sets, with 80 % of the data allotted to training and 20 % to testing. Rather than partitioning the data into multiple subsets, each ML model was run 10 times, resampling and randomly splitting into testing and training sets (Domingos, 2012). Predicted runoff values were then compared to actual observed runoff to assess model accuracy using root mean square error (RMSE), MAE, R2, and bias as performance metrics. Model results were then compared across MOPEX and CAMELS datasets to determine their consistency, assess whether they provide compatible inputs for runoff estimation, and the influence of potential systematic biases in the input data.

SVR was also able to compare MOPEX and CAMELS datasets as a simple binary classification problem using the *e1071* (Meyer et al., 2023) R package. The two datasets were merged into a composite dataset for each climate region and temporal aggregation, and each was identified by either a zero (CAMELS) or one (MOPEX), representing the target variable. The composite dataset was then split into training and test sets, with 75 % of the data allotted to training and 25 % to testing. Data were randomly selected to avoid any potential bias due to formatting, etc. SVR models were trained on the composite datasets to classify the binary label (MOPEX or CAMELS) using precipitation, temperature, and evapotranspiration values as predictor variables. Classification was performed separately for all three climate regions at daily, monthly, seasonal, and water year aggregations. If the datasets are similar, then the model should have difficulty differentiating between them, yielding a classification probability near 50 %, akin to a random guess A double mass curve was also used to check the consistency of the data by plotting the cumulative annual precipitation of CAMELS versus MOPEX. If the data are proportional, then the points will plot as a straight line (Searcy et al., 1960).

**4 Results**

Evaluation and comparison of the internal uncertainty and variability of individual dataset parameters are key to understanding the consistency between the MOPEX and CAMELS datasets, and the potential for merging and extending these datasets. For each dataset, climate parameter variability primarily depends on level of aggregation (daily, monthly, seasonal, annual) and secondarily

on climate type. Between datasets, potentially important biases in climate variables are evident, varying by climate type and aggregation level. This paper presents a thorough exploratory data analysis and supports the main finding that the two datasets exhibit similar uncertainty and variability, both within and between them. By considering multiple statistics, we can evaluate the representativeness of each dataset and identify any systematic differences that may need further investigation. If both datasets exhibit similar means and variability within a climate region, it suggests that their distributions are comparable. Differences in variance and skewness, on the other hand, highlight potential biases between the datasets. Though there are consistent biases, they are minimal for aggregations beyond a daily time step, making them suitable for combined application in climate studies and hydrologic modeling at monthly, seasonal, or annual aggregations. Although efforts were made to distinguish results for internal analyses within datasets and intercomparisons between datasets, the results are often presented together to provide a clearer understanding of how each dataset behaves independently, while also enabling direct cross-dataset evaluation. Consequently, some overlap does occur.

**4.1 Uncertainty and variability within datasets**

The internal uncertainty and variability of the MOPEX and CAMELS datasets were assessed using median, mean, variance, standard deviation, skewness, coefficient of variation, and confidence intervals for each climate region at daily, monthly, seasonal, and annual scales. Precipitation statistics shown in Table 4 were determined for each individual basin, with minimum and maximum values representing all basins within each climate region (number of observations, Table 3). Due to the large differences between temporal precipitation totals, the ranges for each statistic were normalized by finding the difference between the maximum and minimum values and then dividing the difference by the mean of the maximum and minimum values. The normalized ranges (NR) were then used to assess the variability of each statistic within the dataset across the different temporal aggregations and are summarized in Table 4. When calculating the normalized range, a minimum value of zero or close to zero (i.e. median or skew) will conflate the range, making it appear larger than it truly is. For this reason, normalized daily median ranges and normalized skew values are ignored.

**Table 4. Minimum (min), maximum (max), and normalized range of median, mean, variance, standard deviation, and skewness for MOPEX (M) and CAMELS (C) total daily, monthly, seasonal, and annual precipitation (PRCP) totals in mm. Values are based on all basins within a climate region. The normalized ranges (NR) are based on the maximum and minimum values (maximum – minimum / mean (maximum + minimum)).**

| | PRCP mm | Median | | | Mean | | | Variance | | | Standard Deviation | | | Skewness | |
|---|---|---|---|---|---|---|---|---|---|---|---|---|---|---|---|
| | | Min | Max | NR | Min | Max | NR | Min | Max | NR | Min | Max | NR | Min | Max |
| ARID CAMELS | Day | 0.00 | 0.00 | - | 1.30 | 2.50 | 0.63 | 13.93 | 60.91 | 1.26 | 3.73 | 7.80 | 0.71 | 4.01 | 5.84 |
| | Month | 11.50 | 63.25 | 1.38 | 39.55 | 76.01 | 0.63 | 1421.16 | 4110.82 | 0.97 | 37.70 | 64.12 | 0.52 | 1.23 | 2.23 |
| | Season | 91.10 | 209.51 | 0.79 | 118.66 | 228.03 | 0.63 | 6608.73 | 16794.05 | 0.87 | 81.29 | 129.59 | 0.46 | 0.45 | 1.37 |
| | Year | 423.07 | 920.40 | 0.74 | 474.63 | 912.11 | 0.63 | 13628.52 | 64193.54 | 1.30 | 116.74 | 253.36 | 0.74 | -0.46 | 0.87 |
| ARID MOPEX | Day | 0.00 | 0.15 | - | 1.27 | 2.31 | 0.58 | 13.36 | 54.01 | 1.21 | 3.66 | 7.35 | 0.67 | 5.40 | 7.81 |
| | Month | 9.13 | 55.67 | 1.44 | 38.55 | 70.33 | 0.58 | 1329.37 | 3569.40 | 0.91 | 36.46 | 59.74 | 0.48 | 1.00 | 2.34 |
| | Season | 69.89 | 202.28 | 0.97 | 115.65 | 211.00 | 0.58 | 6289.11 | 18484.56 | 0.98 | 79.30 | 135.96 | 0.53 | 0.37 | 1.64 |
| | Year | 381.36 | 878.77 | 0.79 | 462.60 | 844.02 | 0.58 | 11933.15 | 54745.76 | 1.28 | 109.24 | 233.98 | 0.73 | -0.41 | 0.83 |
| CONT CAMELS | Day | 0.00 | 0.57 | - | 2.25 | 3.19 | 0.35 | 11.44 | 64.58 | 1.40 | 3.38 | 8.04 | 0.82 | 2.26 | 4.70 |
| | Month | 58.74 | 88.78 | 0.41 | 68.51 | 97.06 | 0.34 | 1529.66 | 4577.49 | 1.00 | 39.11 | 67.66 | 0.53 | 0.71 | 1.99 |
| | Season | 203.99 | 281.15 | 0.32 | 205.52 | 291.19 | 0.34 | 5124.76 | 20848.61 | 1.21 | 71.59 | 144.39 | 0.67 | -0.01 | 0.95 |
| | Year | 796.50 | 1197.26 | 0.40 | 822.10 | 1164.76 | 0.34 | 11734.23 | 63049.45 | 1.37 | 108.32 | 251.10 | 0.79 | -0.50 | 1.14 |
| CONT MOPEX | Day | 0.03 | 0.87 | - | 2.08 | 3.16 | 0.41 | 9.68 | 62.68 | 1.46 | 3.11 | 7.92 | 0.87 | 2.90 | 5.33 |
| | Month | 60.05 | 86.34 | 0.36 | 63.35 | 96.11 | 0.41 | 966.07 | 4001.93 | 1.22 | 31.08 | 63.26 | 0.68 | 0.83 | 1.62 |
| | Season | 209.10 | 277.72 | 0.28 | 190.04 | 288.34 | 0.41 | 3920.88 | 18210.45 | 1.29 | 62.62 | 134.95 | 0.73 | 0.03 | 1.00 |
| | Year | 765.73 | 1143.82 | 0.40 | 760.18 | 1153.35 | 0.41 | 11497.26 | 53327.77 | 1.29 | 107.23 | 230.92 | 0.73 | -0.41 | 1.09 |
| TEMP CAMELS | Day | 0.00 | 0.70 | - | 2.90 | 5.71 | 0.65 | 26.61 | 135.48 | 1.34 | 5.16 | 11.64 | 0.77 | 2.09 | 4.80 |
| | Month | 79.12 | 165.40 | 0.71 | 88.14 | 173.68 | 0.65 | 1872.09 | 6241.27 | 1.08 | 43.27 | 79.00 | 0.58 | 0.39 | 1.24 |
| | Season | 251.52 | 508.24 | 0.68 | 264.42 | 521.04 | 0.65 | 6496.89 | 28687.61 | 1.26 | 80.60 | 169.37 | 0.71 | 0.18 | 1.12 |
| | Year | 1014.72 | 2181.86 | 0.73 | 1057.68 | 2084.18 | 0.65 | 23220.68 | 128322.97 | 1.39 | 152.38 | 358.22 | 0.81 | -0.66 | 0.91 |
| TEMP MOPEX | Day | 0.04 | 1.09 | - | 2.59 | 5.28 | 0.68 | 27.73 | 144.50 | 1.36 | 5.27 | 12.02 | 0.78 | 2.88 | 5.34 |
| | Month | 71.48 | 151.99 | 0.72 | 78.98 | 160.71 | 0.68 | 1458.72 | 5823.13 | 1.20 | 38.19 | 76.31 | 0.67 | 0.48 | 1.29 |
| | Season | 224.14 | 445.36 | 0.66 | 236.93 | 482.12 | 0.68 | 4861.81 | 26089.63 | 1.37 | 69.73 | 161.52 | 0.79 | 0.05 | 1.12 |
| | Year | 923.61 | 2004.61 | 0.74 | 947.70 | 1928.48 | 0.68 | 18274.22 | 136058.9 | 1.53 | 135.18 | 368.86 | 0.93 | -0.63 | 0.85 |

The range in median values decreases in all regions when moving from a monthly aggregation to seasonal (CAMELS arid 1.38 to 0.79, CAMELS continental 0.41 to 0.32, CAMELS temperate 0.71 to 0.68, MOPEX arid 1.44 to 0.97, MOPEX continental 0.36 to 0.28, MOPEX temperate 0.72 to 0.66). Median ranges continue to contract in arid regions at an annual scale (CAMELS 0.74, MOPEX 0.79). Continental and temperate regions show a slight expansion in median ranges between seasonal and annual aggregations. The range in mean values is uniform in each region over all temporal scales (CAMELS arid 0.63, CAMELS temperate 0.65, MOPEX arid 0.58, MOPEX continental 0.41, MOPEX temperate 0.68), with a minor change of 0.35 to 0.34 in CAMELS continental daily to monthly. All basins within each region demonstrate minimal variability in the mean and proportional aggregation. The range in variance is slightly wider for daily and annual aggregations in all regions for both CAMELS and MOPEX. The range in annual variance increases when moving to an annual aggregation except for MOPEX continental, which remains at a normalized value of 1.29 between seasonal and annual. This suggests interannual variability may be more pronounced than intra-seasonal fluctuations, attributed to the accumulation of extreme precipitation events or shifting between wet and dry. The range of standard deviation values mimics the variability in variance values with the smallest ranges for monthly and seasonal aggregations and minor increases at daily and annual aggregations for all regions except MOPEX continental (seasonal and annual 0.73). Differences in precipitation patterns can become more apparent over longer periods of time. Precipitation variability has been shown to increase over longer time scales under a warming climate (Pendergrass et al., 2017; Zhang et al., 2021), The distribution in all regions tends to become more Gaussian as the aggregation increases from daily to annual, which is to be expected.

The minimum and maximum median, mean, variance, standard deviation, and skewness values for mean temperature in degrees Celsius along with the range values are shown in Table 5. Overall, temperature variability in each climate region decreases with temporal aggregation with daily values showing the highest variability and annual values the lowest for both CAMELS and MOPEX datasets. There is minimal variability in the central tendency in all climate regions, with a slightly narrower spread in the

mean values compared to the median values. In continental regions, the minimum seasonal median value for all CAMELS basins is 0.81 ˚C and 0.96 ˚C for MOPEX basins, which is due to a few colder than average winters in a watershed located in Montana.

Variance in mean temperature is smallest in annual aggregations for all regions because it is based on annual averages, which smooths the extreme values. In contrast, the variability in skewness is greatest at annual aggregations in both CAMELS and MOPEX. Aggregation at an annual scale reduces variance among mean temperature values but at the same time, fewer data points increase the sensitivity to extremes, which can shift the distribution.

**Table 5. Minimum (min), maximum (max), and range of median, mean, variance, standard deviation, and skewness for MOPEX (M) and CAMELS (C) mean daily, monthly, seasonal, and mean temperature (TAIR) in degree Celsius. Values are based on all basins within a climate region. Range is maximum minus minimum.**

| | TAIR | Median | | | Mean | | | Variance | | | Standard Deviation | | | Skewness | | |
|---|---|---|---|---|---|---|---|---|---|---|---|---|---|---|---|---|
| | °C | Min | Max | Range | Min | Max | Range | Min | Max | Range | Min | Max | Range | Min | Max | Range |
| ARID CAMELS | Day | 8.91 | 22.77 | 13.86 | 8.70 | 21.54 | 12.84 | 48.08 | 139.15 | 91.07 | 6.93 | 11.8 | 4.87 | -0.68 | 0.12 | 0.80 |
| | Month | 9.08 | 21.98 | 12.90 | 8.64 | 21.51 | 12.87 | 36.97 | 113.08 | 76.11 | 6.03 | 10.63 | 4.60 | -0.22 | 0.14 | 0.36 |
| | Season | 8.91 | 21.97 | 13.06 | 8.63 | 21.5 | 12.87 | 30.14 | 92.47 | 62.33 | 5.49 | 9.62 | 4.13 | -0.24 | 0.13 | 0.37 |
| | Year | 8.74 | 21.57 | 12.83 | 8.70 | 21.54 | 12.84 | 0.30 | 1.14 | 0.84 | 0.54 | 1.07 | 0.53 | -0.30 | 0.51 | 0.81 |
| ARID MOPEX | Day | 9.08 | 22.81 | 13.73 | 9.08 | 21.57 | 12.49 | 43.84 | 123.74 | 79.90 | 6.62 | 11.12 | 4.50 | -0.68 | 0.15 | 0.83 |
| | Month | 9.12 | 22.00 | 12.88 | 9.02 | 21.54 | 12.52 | 36.49 | 103.1 | 66.61 | 6.04 | 10.15 | 4.11 | -0.23 | 0.19 | 0.42 |
| | Season | 9.00 | 21.90 | 12.90 | 9.02 | 21.54 | 12.52 | 29.49 | 84.07 | 54.58 | 5.43 | 9.17 | 3.74 | -0.23 | 0.16 | 0.39 |
| | Year | 9.04 | 21.59 | 12.55 | 9.08 | 21.57 | 12.49 | 0.27 | 0.86 | 0.59 | 0.52 | 0.93 | 0.41 | -0.33 | 0.45 | 0.78 |
| CONT CAMELS | Day | 1.26 | 14.38 | 13.12 | 0.38 | 13.48 | 13.10 | 75.41 | 141.07 | 65.66 | 8.68 | 11.88 | 3.20 | -0.40 | -0.19 | 0.21 |
| | Month | 1.04 | 13.92 | 12.88 | 0.33 | 13.43 | 13.10 | 59.41 | 119.00 | 59.59 | 7.71 | 10.91 | 3.20 | -0.22 | 0.08 | 0.30 |
| | Season | 0.81 | 13.54 | 12.73 | 0.33 | 13.43 | 13.10 | 49.13 | 99.53 | 50.40 | 7.01 | 9.98 | 2.97 | -0.23 | 0.10 | 0.33 |
| | Year | 1.47 | 13.38 | 11.91 | 0.38 | 13.48 | 13.10 | 0.32 | 0.96 | 0.64 | 0.57 | 0.98 | 0.41 | -0.18 | 0.22 | 0.40 |
| CONT MOPEX | Day | 1.16 | 14.09 | 12.93 | 1.45 | 13.32 | 11.87 | 73.49 | 125.15 | 51.66 | 8.57 | 11.19 | 2.62 | -0.33 | -0.04 | 0.29 |
| | Month | 1.34 | 13.75 | 12.41 | 1.40 | 13.27 | 11.87 | 59.02 | 107.77 | 48.75 | 7.68 | 10.38 | 2.70 | -0.14 | 0.12 | 0.26 |
| | Season | 0.96 | 13.29 | 12.33 | 1.40 | 13.27 | 11.87 | 48.77 | 89.26 | 40.49 | 6.98 | 9.45 | 2.47 | -0.14 | 0.16 | 0.30 |
| | Year | 1.52 | 13.27 | 11.75 | 1.45 | 13.32 | 11.87 | 0.31 | 0.72 | 0.41 | 0.56 | 0.85 | 0.29 | -0.19 | 0.34 | 0.53 |
| TEMP CAMELS | Day | 2.72 | 20.03 | 17.31 | 2.73 | 19.03 | 16.30 | 52.24 | 102.48 | 50.24 | 7.36 | 10.12 | 2.76 | -0.53 | -0.03 | 0.50 |
| | Month | 2.36 | 19.04 | 16.68 | 2.68 | 19.00 | 16.32 | 41.80 | 82.09 | 40.29 | 6.46 | 9.06 | 2.60 | -0.14 | 0.11 | 0.25 |
| | Season | 2.57 | 19.23 | 16.66 | 2.69 | 18.99 | 16.30 | 34.08 | 67.80 | 33.72 | 5.84 | 8.23 | 2.39 | -0.12 | 0.12 | 0.24 |
| | Year | 2.87 | 18.98 | 16.11 | 2.73 | 19.03 | 16.30 | 0.28 | 0.88 | 0.60 | 0.53 | 0.94 | 0.41 | -0.28 | 0.50 | 0.78 |
| TEMP MOPEX | Day | 2.75 | 20.06 | 17.31 | 3.11 | 19.06 | 15.95 | 53.08 | 92.52 | 39.44 | 7.29 | 9.62 | 2.33 | -0.51 | 0.04 | 0.55 |
| | Month | 2.54 | 19.16 | 16.62 | 3.07 | 19.03 | 15.96 | 41.59 | 77.30 | 35.71 | 6.45 | 8.79 | 2.34 | -0.15 | 0.12 | 0.27 |
| | Season | 2.83 | 19.26 | 16.43 | 3.06 | 19.03 | 15.97 | 34.05 | 64.06 | 30.01 | 5.84 | 8.00 | 2.16 | -0.13 | 0.15 | 0.28 |
| | Year | 3.23 | 19.06 | 15.83 | 3.11 | 19.06 | 15.95 | 0.25 | 0.59 | 0.34 | 0.50 | 0.77 | 0.27 | -0.29 | 0.55 | 0.84 |

The coefficient of variation (CV) was calculated for each catchment on all temporal scales for precipitation (Fig. 6). Daily

precipitation shows considerably high variation, with CAMELS mean CV values of 3.28, 2.39, and 2.12 and MOPEX mean CV values of 3.23, 2.42, and 2.22 in arid, continental, and temperate regions respectively (Fig. 6a). Considerably high variation is still observed on monthly scales (Fig. 6b) but decreases to moderate variability for seasonal temporal aggregations for all regions, and low variability, less than one, on an annual scale. The normalized ranges for precipitation variance in Table 4 indicate that annual totals are the most variable while the CV demonstrates decreasing variability from a daily to annual scale. While both are measures

of variability, they differ in how they express dispersion and their sensitivity to scale. Variance is unit dependent and is sensitive to magnitude while the CV is normalized relative to the mean. This suggests that at short time scales, precipitation is more event-driven whereas at longer scales, climate patterns dominate. Temperature demonstrates a consistent decrease in variability from daily to annual temporal aggregation for all regions and is not shown.

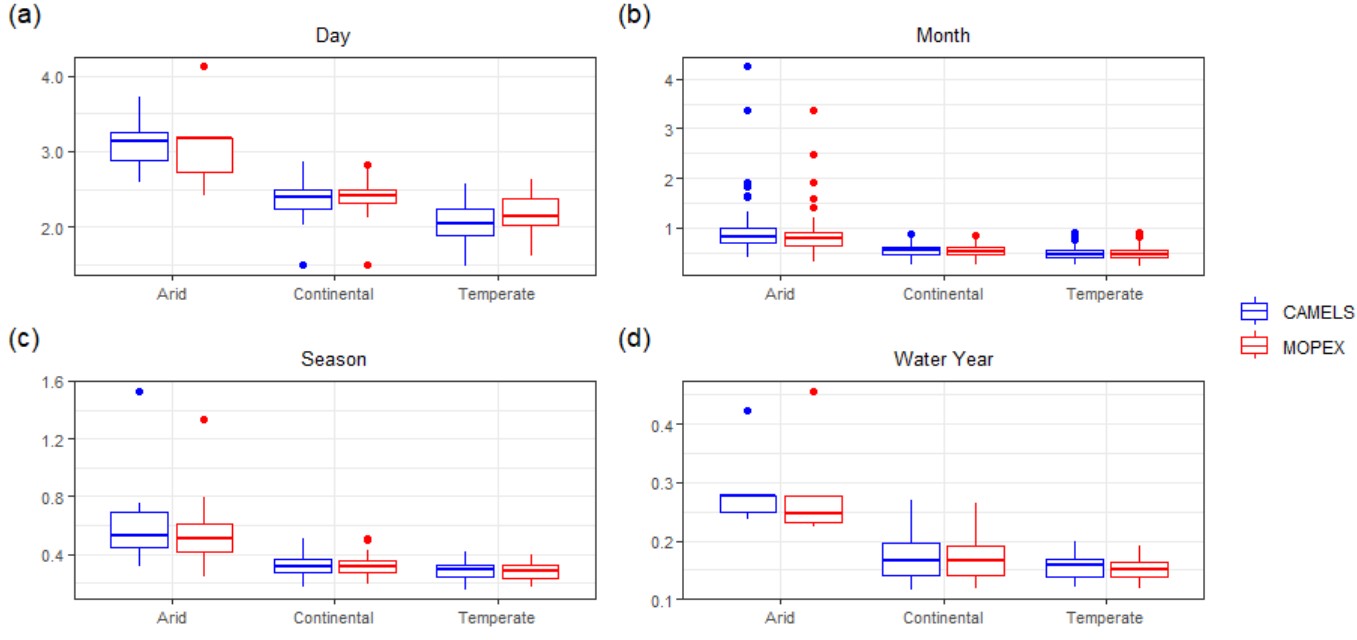

Precipitation coefficient of variation

**Figure 6. Coefficient of variation of precipitation for CAMELS (blue) and MOPEX (red) for each climate region, shown by temporal aggregation a) day, b) month, c) season, and d) water year. Each boxplot represents the value of all basins within the climate region based on total precipitation (mm). Note progressively declining y-axis range from (a) to (d).**

Two-sided interval estimates were computed to determine the uncertainty within each dataset. The daily mean precipitation and temperature were calculated for each basin and the corresponding 95-percent confidence intervals were established by bootstrapping, using 10,000 samples with replacement. The results in Table 4 illustrate that overall daily precipitation means are larger for CAMELS than for MOPEX (except for arid regions); however, it is noteworthy that the confidence intervals, shown in Table 7, exhibit overlap for most regions, suggesting similar degrees of uncertainty. The most pronounced divergence in precipitation means, a difference of 7 percent, is observed in temperate catchments where the overall CAMELS daily mean is 3.73 mm day$^{-1}$ and MOPEX is 3.50 mm day$^{-1}$.

When examining total mean monthly precipitation, both datasets exhibit comparable monthly fluctuations (Fig. 7), but CAMELS exhibits a small positive bias in non-arid climate regions. Arid regions display the most variability, with the largest confidence intervals (± 13.74 mm month$^{-1}$ for CAMELS and ± 12.93 mm month$^{-1}$ for MOPEX) observed in June, and the smallest (± 7.00 mm month$^{-1}$ for CAMELS and ± 7.73 mm month$^{-1}$ for MOPEX) observed in November. Despite this variability, these regions show the greatest temporal consistency between MOPEX and CAMELS values, with total precipitation highest in May and June and lowest in April (Fig. 7a). Additionally, arid regions demonstrate the most notable overlap of the mean values and confidence intervals of the two datasets. Continental regions show an increase in total monthly precipitation in May, June, and July (Fig. 7b). There is the least amount of variation in February (± 4.12 mm month$^{-1}$ for CAMELS and ± 3.79 mm month$^{-1}$ for MOPEX), contrasting with the largest in July (± 6.31 mm month$^{-1}$ for CAMELS and ± 6.08 mm month$^{-1}$ for MOPEX). Temperate regions show decreased precipitation in August, September, and October with less overlap between dataset confidence intervals (Fig. 7c). The smallest confidence intervals differ between datasets with April (± 4.92 mm month$^{-1}$) for CAMELS and July (± 4.54 mm month$^{-1}$) for MOPEX, but both share the largest amount of variability in December (± 8.19 mm month$^{-1}$ for CAMELS and ± 10.21 mm month$^{-1}$ for MOPEX).

Average total monthly precipitation

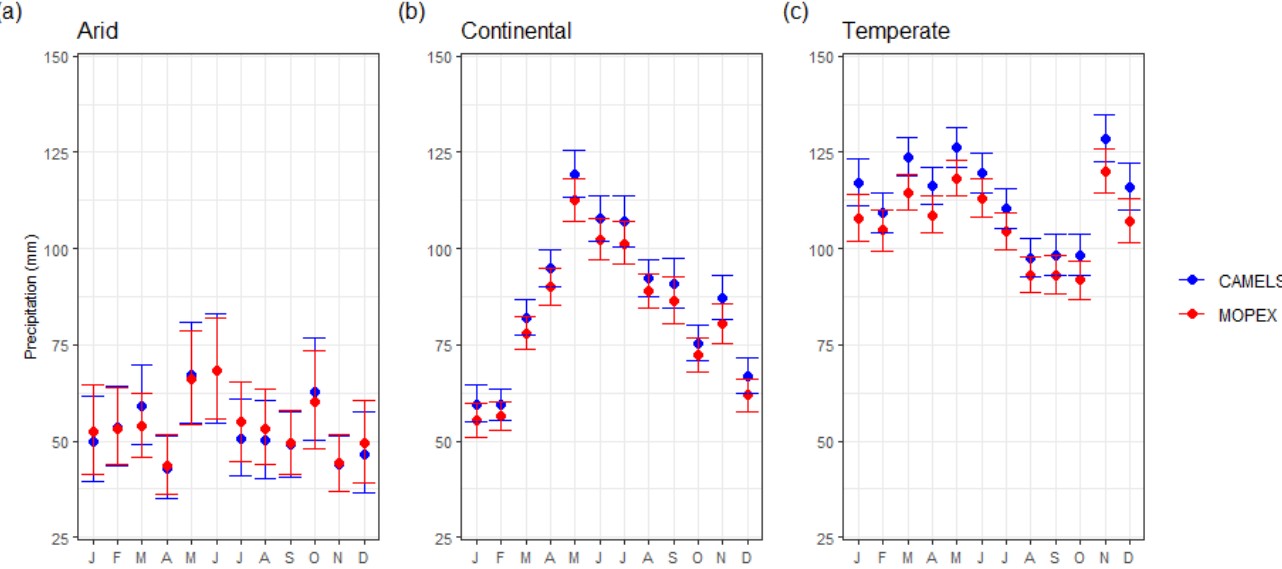

**Figure 7. Average total monthly precipitation for CAMELS (blue) and MOPEX (red) by a) arid, b) continental, and c) temperate climate region. The mean value is determined using all basins within the climate region and each corresponding month for 1981-2000. Error bars represent two-sided 95 % confidence interval, derived from bootstrapping with replacement for 10,000 replicates.**

Seasonal precipitation confidence intervals exhibit the most variability yet also the greatest consistency in arid regions (Fig. 8), which coincides with monthly precipitation analyses (Fig. 7). The range of potential values decreases in continental and temperate regions. MOPEX values are larger than CAMELS in arid regions in the summer and winter seasons (which corresponds to larger monthly values in December, January, June, July, and August). For arid regions (Fig. 8a), the greatest variance is in the winter season ($\pm$ 25.24 mm season$^{-1}$ for CAMELS and $\pm$ 25.37 mm season$^{-1}$ for MOPEX). Continental regions (Fig. 8b) show the greatest uncertainty in summer for CAMELS ($\pm$ 10.58 mm season$^{-1}$) and fall for MOPEX ($\pm$ 10.24 mm season$^{-1}$). Temperate regions (Fig. 8c) have the largest differences in variance between datasets with little to no overlap of confidence intervals, notably in the spring. Winter has the greatest confidence intervals for CAMELS ($\pm$ 17.69 mm season$^{-1}$) and MOPEX ($\pm$ 22.89 mm season$^{-1}$).

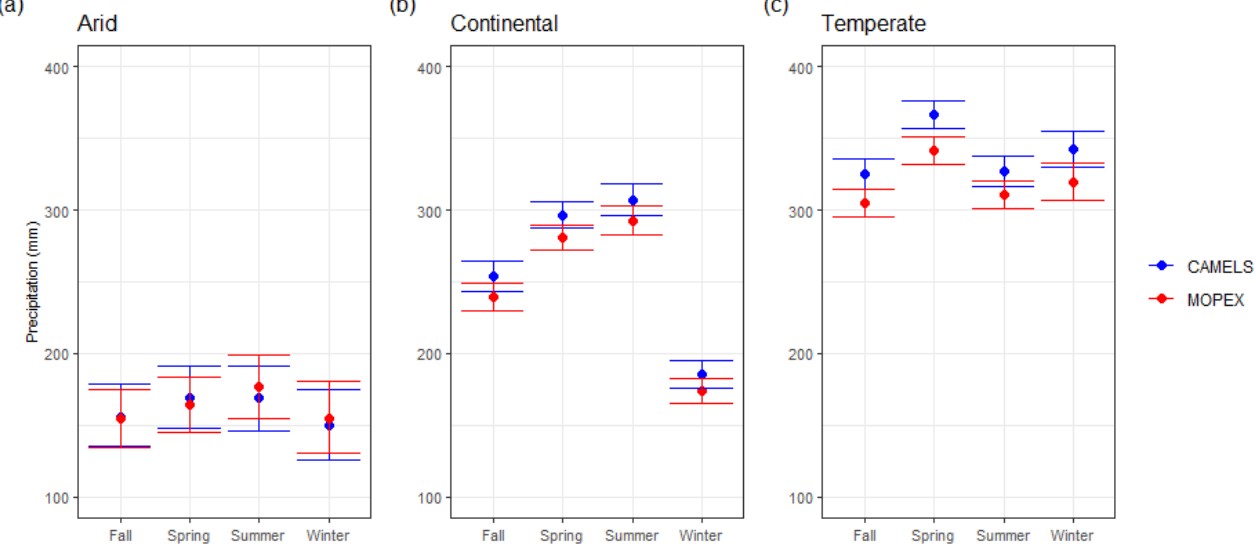

Average total seasonal precipitation

**Figure 8. Average total seasonal precipitation for CAMELS (blue) and MOPEX (red) by a) arid, b) continental, and c) temperate climate region. The mean value is determined using all basins within the climate region and each corresponding season for 1981-2000. Error bars represent two-sided 95 % confidence interval, derived from bootstrapping with replacement for 10,000 replicates.**

For average total annual precipitation, arid regions exhibit the highest variability within each individual dataset, however, their mean values remain similar between the two datasets (Fig. 9a); other climate regions exhibit a small positive precipitation bias for CAMELS. Arid region confidence intervals are greater for CAMELS (between $\pm$ 70.91 mm year$^{-1}$ in 1996 and $\pm$ 326.94 mm year$^{-1}$ in 1987) than MOPEX (between $\pm$ 33.96 mm year$^{-1}$ in 1996 and $\pm$ 298.17 mm year$^{-1}$ in 1985). Annual means in continental (Fig. 9b) and temperate regions (Fig. 9c) are consistently higher in CAMELS, but confidence intervals do overlap with MOPEX. The smallest uncertainty is in continental regions with intervals slightly larger for CAMELS ($\pm$ 48.03 mm year$^{-1}$ in 1986 to $\pm$ 136.19 mm year$^{-1}$ in 1996) compared to MOPEX ($\pm$ 43.57 mm year$^{-1}$ in 1992 to $\pm$ 126.91 mm year$^{-1}$ in 1996). Temperate regions have greater uncertainty associated with MOPEX values ($\pm$ 92.22 mm year$^{-1}$ in 1992 to $\pm$ 221.13 mm year$^{-1}$ in 1982) rather than CAMELS ($\pm$ 72.39 mm year$^{-1}$ in 1981 to $\pm$ 152.43 mm year$^{-1}$ in 1995).

Average total annual precipitation

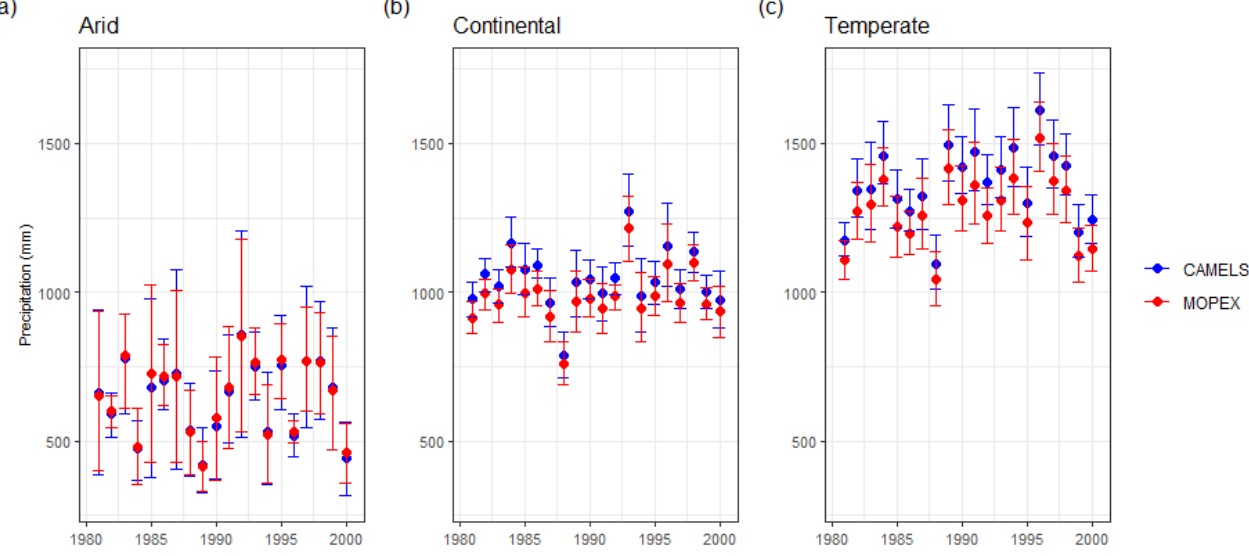

**Figure 9. Average total annual precipitation for CAMELS (blue) and MOPEX (red) by a) arid, b) continental, and c) temperate climate region. The mean value is determined using all basins within the climate region and each corresponding water year for 1981-2000. Error bars represent two-sided 95 % confidence interval, derived from bootstrapping with replacement for 10,000 replicates.**

Evaluation of daily temperature indicates a consistent pattern discerned in Tables 5 and 8. The means for daily temperature are consistently larger for MOPEX with the largest differences (mean difference of 0.62 ˚C day$^{-1}$) observed in arid regions. Monthly temperatures show consistent trends in both datasets, with higher temperatures in July and August and lower temperatures in January and December in all regions. MOPEX and CAMELS are quite similar in their mean values and monthly variability (Fig. 10). Akin to precipitation, arid regions contain the most variability, followed by temperate regions. The largest uncertainty is in December for CAMELS (± 1.37 ˚C month$^{-1}$) and MOPEX (± 1.32 ˚C month$^{-1}$) in arid and continental regions (CAMELS ± 0.46 ˚C month$^{-1}$, MOPEX ± 0.40 ˚C month$^{-1}$) and in February for temperate regions (CAMELS ± 0.45 ˚C month$^{-1}$, MOPEX ± 0.43 ˚C month$^{-1}$). The smallest uncertainty is in July (CAMELS ± 0.80 ˚C month$^{-1}$, MOPEX ± 0.71 ˚C month$^{-1}$) for arid regions, October (CAMELS ± 0.32 ˚C month$^{-1}$, MOPEX ± 0.31 ˚C month$^{-1}$) for continental regions, and August (CAMELS ± 0.36 ˚C month$^{-1}$, MOPEX ± 0.33 ˚C month$^{-1}$) for temperate regions.

Average monthly temperature

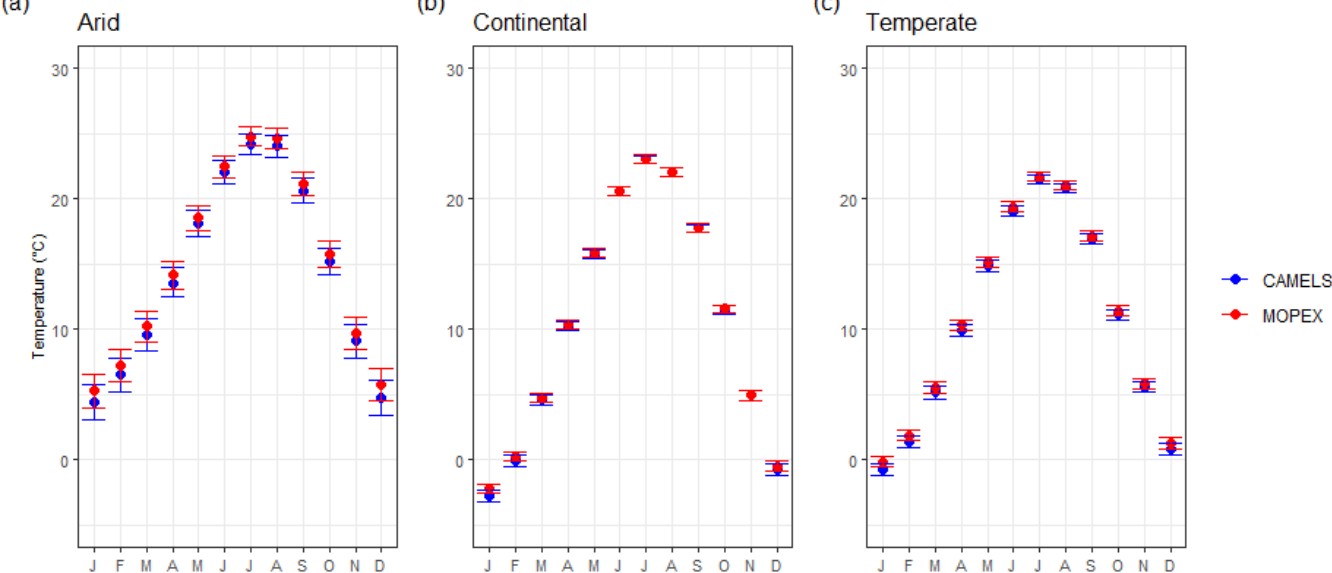

Figure 10. Average monthly mean temperature for CAMELS (blue) and MOPEX (red) by a) arid, b) continental, and c) temperate climate region. The mean value is determined using all basins within the climate region and each corresponding month for 1981-2000. Error bars represent two-sided 95 % confidence interval, derived from bootstrapping with replacement for 10,000 replicates.

Seasonal temperature is also the most variable in arid regions with confidence intervals ranging from ± 0.77 ˚C to ± 1.28˚C season$^{-1}$ compared to intervals ranging from ± 0.30 ˚C to ± 0.40 ˚C season$^{-1}$ for continental and temperate regions (Fig. 11). Winter is consistently the most variable season among all regions, resulting in the largest confidence intervals.

Average seasonal temperature

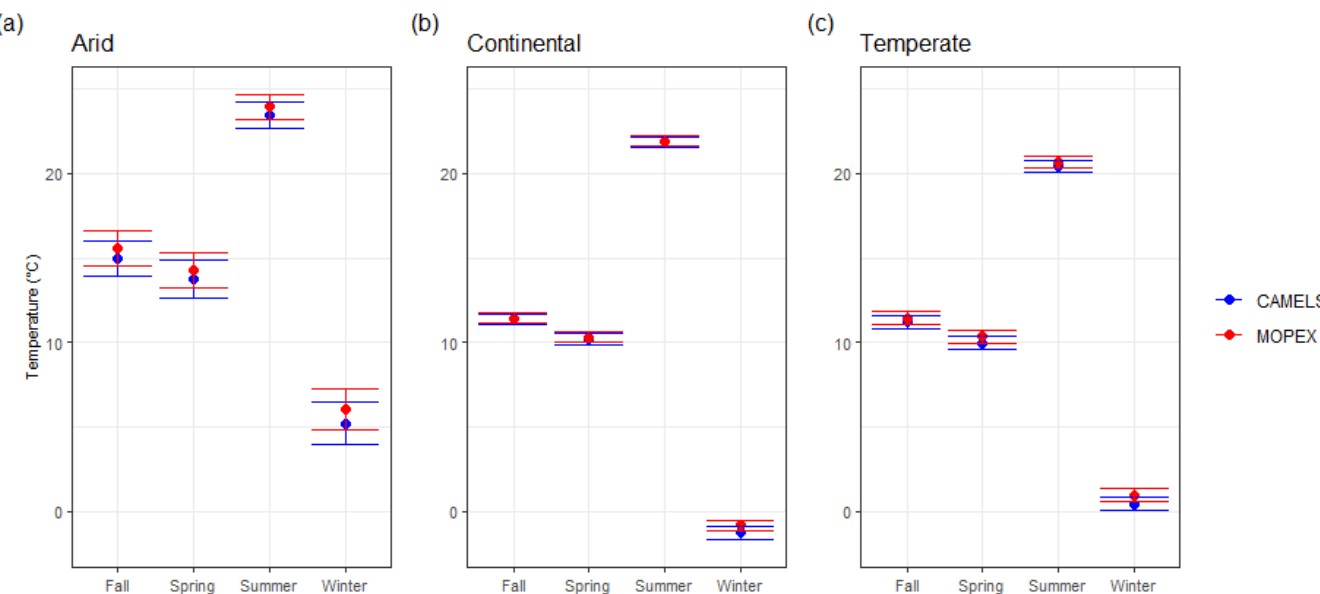

Figure 11. Average mean seasonal temperature for CAMELS (blue) and MOPEX (red) by a) arid, b) continental, and c) temperate climate region. The mean value is determined using all basins within the climate region and each corresponding season for 1981-2000. Error bars represent two-sided 95 % confidence interval, derived from bootstrapping with replacement for 10,000 replicates.

Annually, for temperature, arid confidence intervals are more than double the range of those found in continental and temperate regions (Fig. 12), strongly influenced by the small number of arid sites (Table 3). MOPEX means are consistently larger than CAMELS indicating a warm bias with the largest bias in arid regions (Fig. 12a). Continental regions have the most similarity between mean values and the smallest amount of uncertainty, with confidence intervals ranging from ± 1.24 ˚C year$^{-1}$ to ± 1.47 ˚C year$^{-1}$ for CAMELS and from ± 1.10 ˚C year$^{-1}$ to ± 1.31 ˚C year$^{-1}$ for MOPEX (Fig. 12b).  For temperate regions, MOPEX has a slightly smaller variance compared to CAMELS (Fig. 12c) with confidence intervals ranging from ± 1.44 ˚C year$^{-1}$ to ± 1.73 ˚C year$^{-1}$ (MOPEX) versus ± 1.47 ˚C year$^{-1}$ to ± 1.82 ˚C year$^{-1}$ (CAMELS).

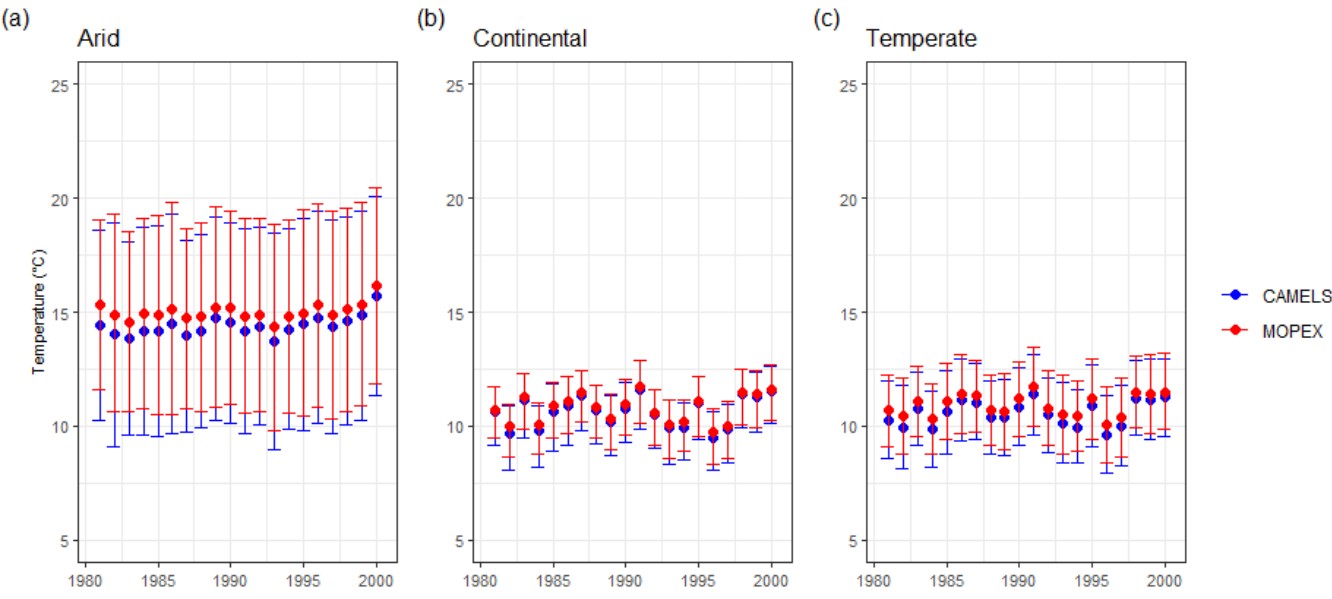

**Figure 12. Average annual temperature for CAMELS (blue) and MOPEX (red) by a) arid, b) continental, and c) temperate climate region. The mean value is determined using all basins within the climate region and each corresponding water year for 1981-2000. Error bars represent two-sided 95 % confidence interval, derived from bootstrapping with replacement for 10,000 replicates.**

**4.2 Uncertainty and variability between datasets**

Important differences between the datasets are detailed below, but in general in time-aggregated values MOPEX exhibits higher temperature, while CAMELS exhibits higher precipitation. Statistical and bootstrapping results from Sect. 4.1 supports these findings. The comparison of paired observations via binomial sign test, CAMELS values minus MOPEX, indicates that individual daily MOPEX values for precipitation and temperature are generally larger than CAMELS; in contrast, when CAMELS precipitation values are aggregated monthly, seasonally, or annually, they are typically larger than MOPEX in continental and temperate regions (Fig. 13). This analysis is based solely on the counts of negative (MOPEX > CAMELS), positive (CAMELS > MOPEX), and zero values (CAMELS = MOPEX). The magnitudes of the differences are not incorporated. Out of the 7,305 days recorded for each basin, precipitation values for MOPEX surpass CAMELS 48 % (62,638 days out of 131,490 total days) and 49 % (86,496 days out of 175,320 total days) of the time in continental and temperate climates respectively, and 40 % of the time in arid regions (Fig. 13a). In arid climates, MOPEX and CAMELS precipitation values are equal 46 % of total days, while in continental and temperate climates, they are equal 28 % and 22 % of total days. The same binomial test was conducted to analyse total monthly precipitation for each catchment. Direct comparisons were made for each month across all water years (i.e. January

1981, January 1982), tallying negative and positive differences, resulting in 240 months per catchment. When aggregated on a monthly scale, CAMELS typically exhibits greater total monthly precipitation, particularly in continental (66 %) and temperate (69 %) regions. Identical ("SAME", Fig. 13b) total values are negligible. In contrast, arid regions indicate larger MOPEX values in 55 % of all months and only 3.25 % of all months have the same total value (Fig. 13b). Seasonal comparisons, based on 80 seasons per catchment, indicate the same pattern with total precipitation greater in MOPEX for 52 % of all seasons in arid regions, and CAMELS greater in continental and temperate regions for 76 % and 78 % of all seasons respectively (Fig. 13c). On an annual scale, 20 years per watershed, the comparison reveals that total precipitation for arid regions is evenly split, with CAMELS and MOPEX dominating 51 % and 49 % of all years respectively. In contrast, continental and temperate regions are largely dominated by CAMELS, constituting 88 % of all years (Fig. 13d). All comparisons, except for arid seasonal and annual, failed to reject the null hypothesis, which expects a median difference of zero between paired observations.

Figure 13. Sign tally results from non-parametric binomial sign test for a) daily, b) monthly, c) seasonal, and d) annual precipitation values. Counts on the y-axis reflect the number of basins (gauges) within each climate region times the number of temporal periods. All results are based on CAMELS minus MOPEX values. Positive values (CAMELS) indicate that CAMELS > MOPEX (blue bars), negative (MOPEX) values indicate MOPEX > CAMELS (red bars), and zero (SAME) indicate that CAMELS = MOPEX (black bars).

Regarding temperature, MOPEX exceeds CAMELS 72 % of total days in arid, 58 % in continental, and 65 % in temperate regions respectively (Fig. 14a). These regions all exhibit the same mean daily temperature values (CAMELS = MOPEX) only 0.03 % of total days. On a monthly scale, MOPEX mean temperature values are larger for all regions, with arid at 81 %, continental at 58 %, and temperate at 74 % of total months with no equal values (Fig. 14b). Seasonal temperature is greater for MOPEX values 85 %, 58 %, and 77 % of all seasons for arid, continental, and temperate regions (Fig. 14c). As for annual mean temperatures, MOPEX values are greater for arid regions in 91 % of all years, while continental and temperate regions show MOPEX dominance in 65 % and 79 % of all years respectively (Fig. 14d). All temperature differences were statistically significant.

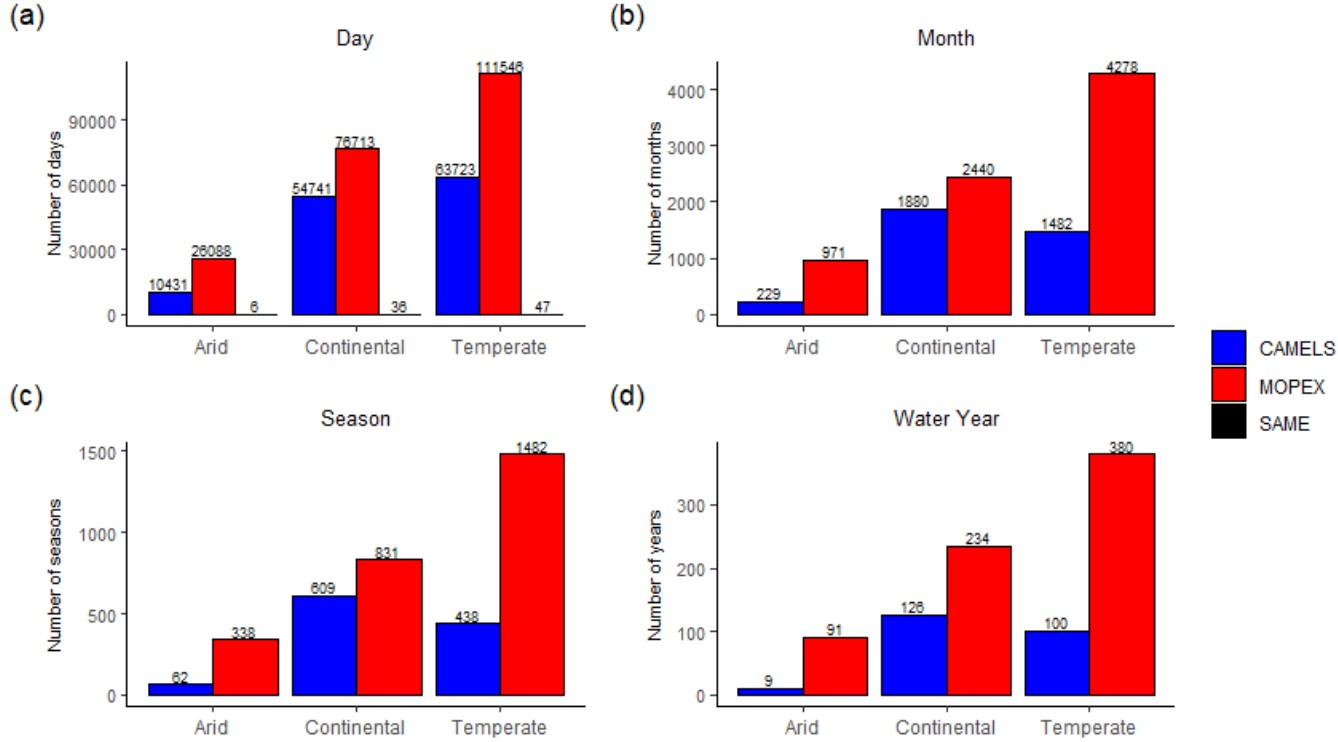

**Figure 14. Sign tally results from non-parametric binomial sign for a) daily, b) monthly, c) seasonal, and d) annual temperature values. Counts on the y-axis reflect the number of basins (gauges) within each climate region times the number of temporal periods. All results are based on CAMELS minus MOPEX values. Positive values (CAMELS) indicate that CAMELS > MOPEX (blue bars), negative (MOPEX) values indicate MOPEX > CAMELS (red bars), and zero (SAME) indicate that CAMELS = MOPEX (black bars).**

The numerical differences between each pair of same day precipitation values, CAMELS minus MOPEX, reveal substantial differences for extreme events. Specifically, there are 20 instances of daily precipitation values differing by more than 100 mm in separate comparisons across all catchments. This indicates notable variations in daily precipitation values between the two datasets. Daily values do not consistently coincide, as exemplified by the comparison of the same maximum precipitation events for each climate region between 1981 and 2000 (Table 6). In the temperate region, for instance, CAMELS reports the maximum precipitation (181.04 mm day$^{-1}$) occurring on 7 April 1983 for gauge 02479300 while MOPEX, for the same date, reports a precipitation total of 64.07 mm. MOPEX reports the maximum precipitation (183.25 mm day$^{-1}$) as occurring on 20 January 1993 at the same gauge (while CAMELS shows a precipitation value of 59.73 mm day$^{-1}$). Consequently, this study does not recommend direct daily comparisons between MOPEX and CAMELS due to discrepancies in single precipitation events.

**Table 6. Largest precipitation event on record for each climate region. Max indicates the maximum daily measurement on record for that dataset between 1981–2000 along with the corresponding value in the other dataset on that date for comparison.**

| Climate | GaugeID | Date | CAMELS (max) | MOPEX | GaugeID | Date | MOPEX (max) | CAMELS |
|---------|---------|------|--------------|-------|---------|------|-------------|--------|
| Arid | 08171300 | 17 Oct 1998 | 152.12 mm | 216.23 mm | 08171300 | 17 Oct 1998 | 216.23 mm | 152.12 mm |
| Continental | 02016000 | 5 Nov 1985 | 126.82 mm | 11.46 mm | 03237500 | 1 Mar 1997 | 140.08 mm | 68.68 mm |
| Temperate | 02479300 | 7 Apr 1983 | 181.04 mm | 64.07 mm | 02479300 | 20 Jan 1993 | 183.25 mm | 59.73 mm |

A positive precipitation bias for CAMELS is visible for all watersheds within a climate region for all temporal aggregations (positive values, Fig. 15). Monthly, precipitation biases for arid regions range from -52.57 mm month⁻¹ to 99.28 mm month⁻¹, continental regions range from -57.60 mm month⁻¹ to 103.22 mm month⁻¹, and temperate regions are between -102.68 mm month⁻¹ to 117.29 mm month⁻¹, comparing 240 months per catchments (Fig. 15a). Seasonal precipitation biases for arid, continental, and temperate regions are -92.21 mm season⁻¹ to 94.65 mm season⁻¹, -64.04 mm season⁻¹ to 137.32 mm season⁻¹, and -123.50 mm season⁻¹ to 174.88 mm season⁻¹ (Fig. 15b). Total annual precipitation bias ranges are -174.43 mm year⁻¹ to 160.03 mm year⁻¹, -111.77 mm year⁻¹ to 315.40 mm year⁻¹, and -256.88 mm year⁻¹ to 405.25 mm year⁻¹ based on 20 years per catchments (Fig. 15c).

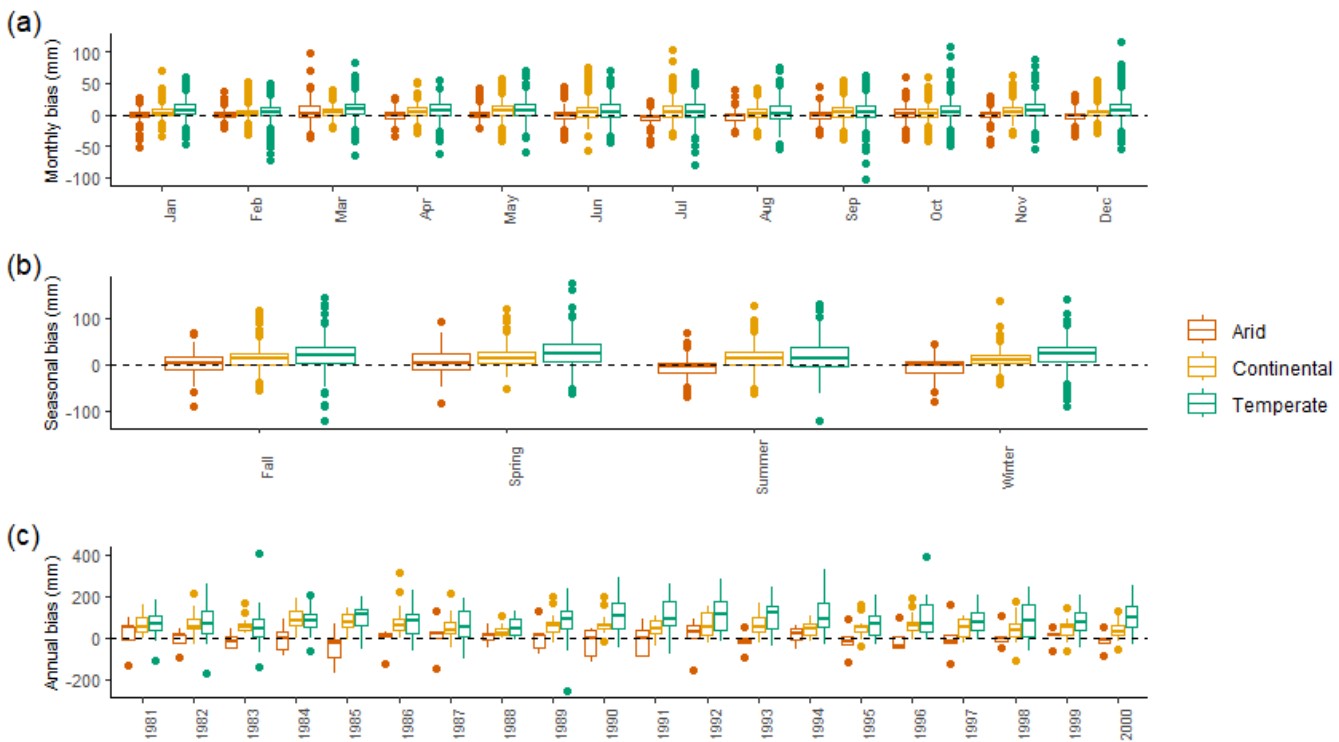

**Figure 15. Monthly (a), seasonal (b), and annual (c) precipitation biases. All basins are combined by climate region (arid, continental, and temperate) and box plots represent the number of observations indicated in Table 3. Precipitation biases are based on total values of CAMELS minus MOPEX. Positive values indicate CAMELS > MOPEX and negative values indicate MOPEX > CAMELS.**

A negative temperature bias for CAMELS vs MOPEX is visible for all watersheds in a climate region for all temporal aggregations (negative values, Fig. 16). Daily temperature values differ between the datasets by as much as ± 28 ˚C day⁻¹ with MOPEX demonstrating a greater positive bias (Fig. 14a). Monthly temperature biases for arid regions range from -5.29 ˚C to 2.00 ˚C month⁻¹, continental regions range from -6.43 ˚C to 0.70 ˚C month⁻¹, and temperate regions range from -5.51 ˚C to 2.26 ˚C month⁻¹ (Fig. 16a). Seasonal temperature biases range from -2.71 ˚C to 1.70 ˚C season⁻¹, -4.24 ˚C to 0.53 ˚C season⁻¹, and -2.84 ˚C to 0.85 ˚C season⁻¹ (Fig. 16b) and mean annual temperature biases decrease to -2.17 ˚C to 0.09 ˚C year⁻¹, -2.44 ˚C to 0.39 ˚C year⁻¹, and -1.89 ˚C to 0.36 ˚C year⁻¹ (Fig. 16c) for arid, continental, and temperate regions respectively.

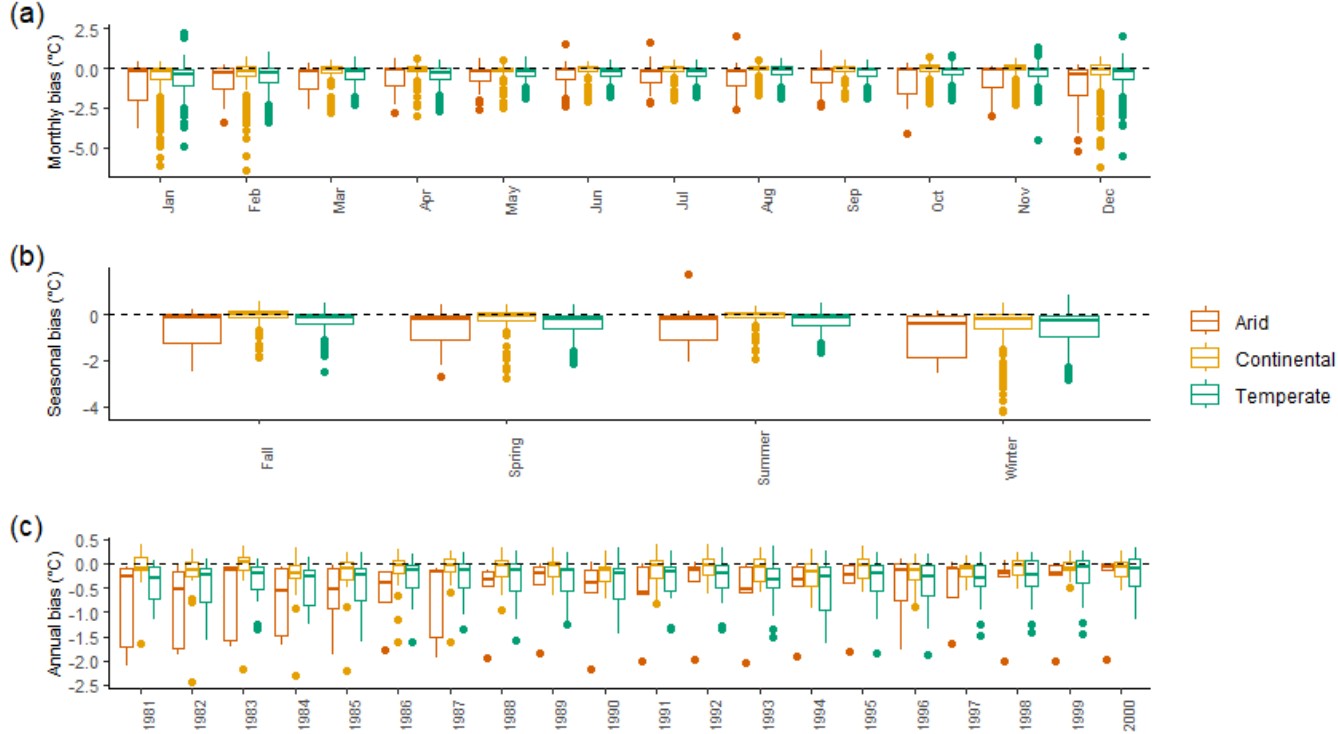

 **Figure 16. Monthly (a), seasonal (b), and annual (c) temperature biases. All basins are combined by climate region (arid, continental, and temperate) and box plots represent the number of observations indicated in Table 3. Temperature biases are based on mean values of CAMELS minus MOPEX. Positive values indicate CAMELS > MOPEX and negative values indicate MOPEX > CAMELS.**

While the ranges of biases for precipitation and temperature are shown in Fig. 15 and Fig. 16 respectively, the magnitude of
differences between MOPEX and CAMELS precipitation and temperature values is clarified by averaging biases over all basins
in a climate region for daily, monthly, seasonal, and annual time aggregations for 1981–2000 (Fig. 17). Given that the differences
are either negative (MOPEX > CAMELS) or positive (CAMELS > MOPEX), the mean reflects the overall bias since equal
differences will negate each other. In direct pairwise comparisons, MOPEX daily precipitation values tend to be larger than
CAMELS, however, when CAMELS values exceed MOPEX, the numerical difference is greater. Daily averages (not shown) for
precipitation bias are -0.02 mm day$^{-1}$ (MOPEX > CAMELS) for arid regions, 0.15 mm day$^{-1}$ (CAMELS > MOPEX) for continental,
and 0.23 mm day$^{-1}$ (CAMELS > MOPEX) for temperate regions, indicating a wet bias in arid regions for MOPEX and a wet bias
for CAMELS in continental and temperate regions. When precipitation values are aggregated on a monthly scale (Fig. 17a),
CAMELS values exceed MOPEX values by 2.94 mm month$^{-1}$ (Feb) to 6.79 mm month$^{-1}$ (May) in continental regions and by 4.41
mm month$^{-1}$ (Aug) to 9.31 mm month$^{-1}$ (Mar) in temperate regions. In arid climates, CAMELS exceeds MOPEX by 0.18 mm
month$^{-1}$ (Feb), 5.07 mm month$^{-1}$ (Mar), 1.11 mm month$^{-1}$ (May), and 2.56 mm month$^{-1}$ (Oct) while MOPEX exceeds CAMELS
(negative values) by 2.58 mm month$^{-1}$ (Jan), 0.84 mm month$^{-1}$ (Apr), 0.14 mm month$^{-1}$ (Jun), 4.38 mm month$^{-1}$ (Jul), 3.16 mm
month$^{-1}$ (Aug), 0.42 mm month$^{-1}$ (Sep), 0.36 mm month$^{-1}$ (Nov), and 2.88 mm month$^{-1}$ (Dec). Average seasonal precipitation
differences are larger for CAMELS in continental regions (Fig. 17b), ranging between 12.13 mm season$^{-1}$ (DJF) to 15.66 mm
season$^{-1}$ (MAM), and temperate regions, ranging between 16.78 mm season$^{-1}$ (JJA) to 24.88 mm season$^{-1}$ (MAM). Average arid
precipitation differences are larger in JJA and DJF by 7.68 mm season$^{-1}$ and 5.27 mm season$^{-1}$ for MOPEX values and larger in
SON and MAM by 1.79 mm season$^{-1}$ and 5.34 mm season$^{-1}$ for CAMELS. Mean annual differences mirror the biases observed in

monthly aggregations (Fig. 17c). Annual CAMELS precipitation values are 1.62 mm year$^{-1}$ to 12.05 mm year$^{-1}$ larger (1981, 1987, 1988, 1989, 1992, 1994, 1997, 1998, 1999) than MOPEX, while MOPEX values are 5.63 mm year$^{-1}$ to 46.91 mm year$^{-1}$ larger than CAMELS for annual totals (1982, 1983, 1984, 1985, 1986, 1990, 1991, 1993, 1995, 1996, 2000) in arid regions. CAMELS values

in continental regions are 27.84 mm year$^{-1}$ to 89.23 mm year$^{-1}$ larger, and temperate regions are 51.46 mm year$^{-1}$ to 112.03 mm year$^{-1}$ larger than MOPEX values.

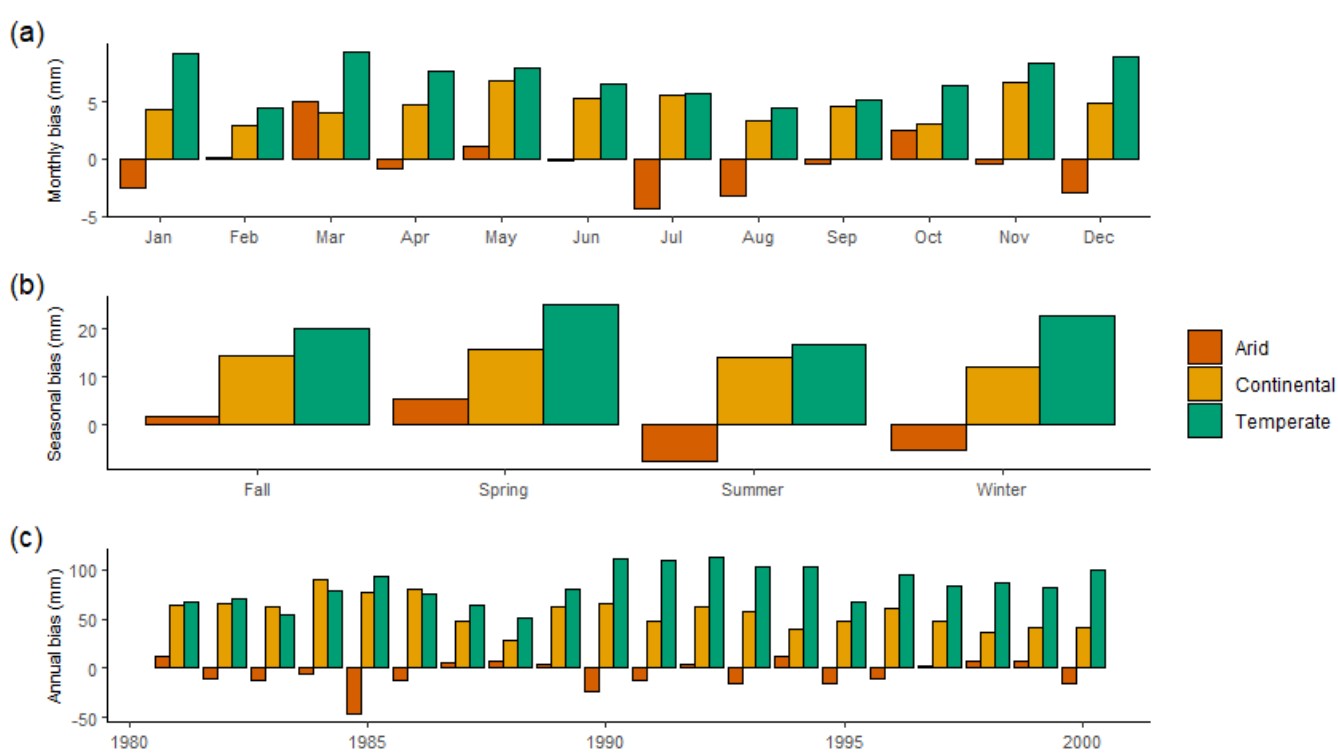

**Figure 17. Magnitude of precipitation bias averaged over all watersheds in a climate region (arid, continental, and temperate) based on a) monthly, b) seasonal, and c) annual totals. All differences are CAMELS minus MOPEX values. Positive bias indicates CAMELS > MOPEX while negative bias indicates MOPEX > CAMELS.**

The average of daily temperature differences indicated MOPEX values were greater than CAMELS by 0.62 ˚C day$^{-1}$ for arid basins,

0.15 ˚C day$^{-1}$ for continental basins, and 0.35 ˚C day$^{-1}$ for temperate basins, suggesting a warmer bias in all MOPEX values. For monthly aggregations, temperature exhibits larger values for MOPEX by 0.41 ˚C month$^{-1}$ to 0.95 ˚C month$^{-1}$, 0.01 ˚C month$^{-1}$ to 0.54 ˚C month$^{-1}$, and 0.24 ˚C month$^{-1}$ to 0.64 ˚C month$^{-1}$ in arid, continental, and temperate regions respectively (Fig. 18a). Seasonally, temperature differences indicate a warm MOPEX bias with average differences of 0.51 ˚C season$^{-1}$ to 0.85 ˚C season$^{-1}$ in arid, 0.03 ˚C season$^{-1}$ to 0.41 ˚C season$^{-1}$ in continental, and 0.25 ˚C season$^{-1}$ to 0.52 ˚C season$^{-1}$ in temperate regions (Fig.

18b). Mean annual temperature differences indicate MOPEX is greater than CAMELS by 0.43 ˚C year$^{-1}$ to 0.86 ˚C year$^{-1}$, 0.07 ˚C year$^{-1}$ to 0.29 ˚C year$^{-1}$, and 0.23 ˚C year$^{-1}$ to 0.47 ˚C year$^{-1}$ for arid, continental, and temperate regions respectively (Fig. 18c).

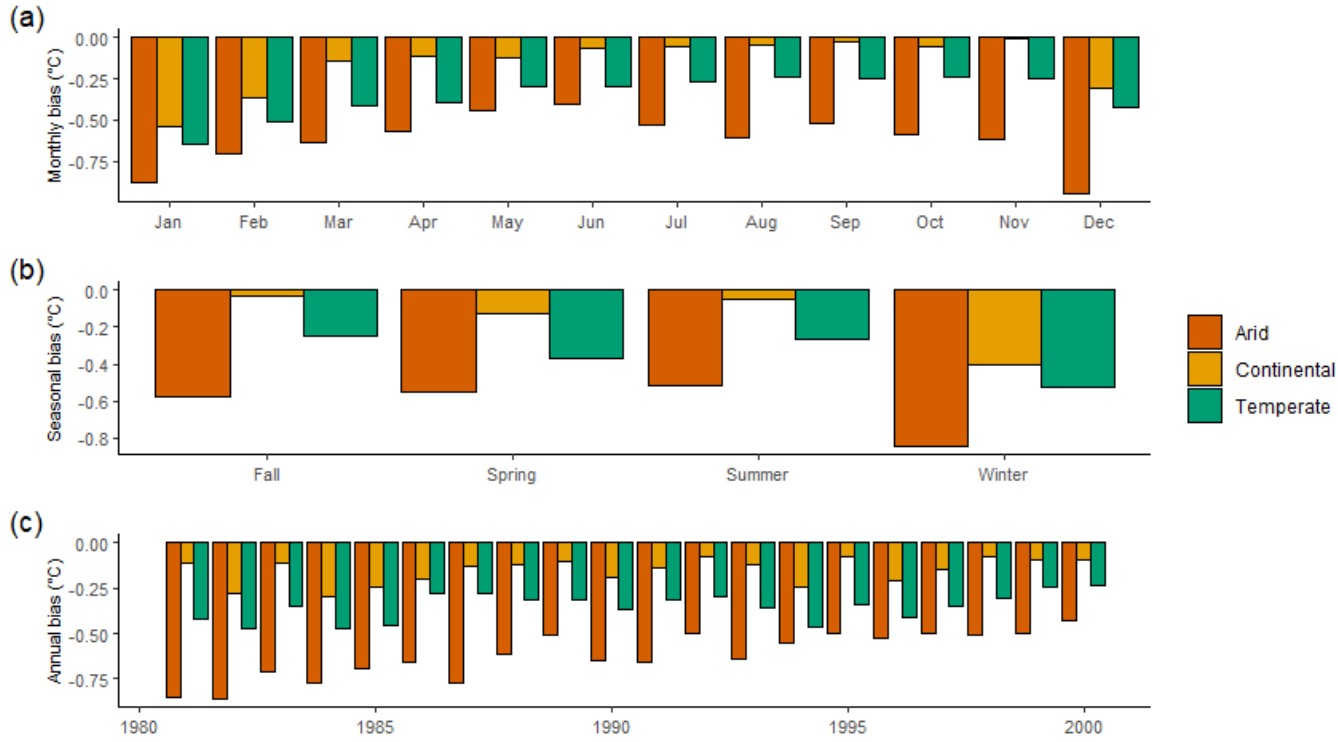

**Figure 18. Magnitude of temperature bias averaged over all watersheds in a climate region (arid, continental, and temperate) based on a) monthly, b) seasonal, and c) annual totals. All differences are CAMELS minus MOPEX values. Positive bias indicates CAMELS > MOPEX while negative bias indicates MOPEX > CAMELS.**

Spatial distribution of precipitation and temperature mean biases between the two datasets shows some geographic concentration, especially of positive (CAMELS) bias for precipitation in the Eastern U.S. (Fig. 19a). Arid regions show an overall wet bias for MOPEX (two watersheds have a slight wet bias for CAMELS) while continental and temperate regions have a wet bias for CAMELS for all temporal aggregations. Temperature biases in Fig. 19b show an overall warm bias for MOPEX for all regions with the exception of four continental watersheds and two temperate watersheds.

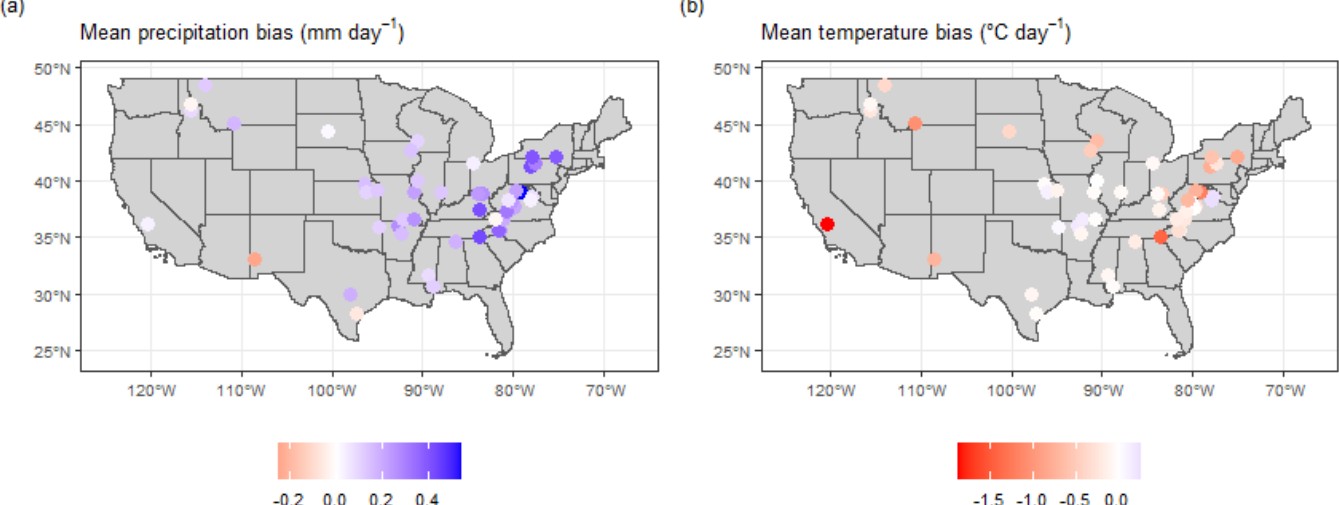

**Figure 19. Overall bias for a) mean precipitation at each gauge location based on CAMELS minus MOPEX. Color bar represents bias in mm per day where negative values (red) indicate a MOPEX bias and positive values (blue) indicate a CAMELS wet bias. Overall bias for b) mean temperature at each gauge location based on CAMELS minus MOPEX. Color bar represents bias in degrees Celsius per day where negative values (red) indicate a MOPEX bias and positive values (blue) indicate a CAMELS warm bias.**

Overall statistics for precipitation are shown in Table 7 and were calculated over all shared basins within a climate region. Temperature statistics are shown in Table 8. The mean values and corresponding confidence intervals are based on the averages derived from bootstrapping results, shown in Figs. 7-9 for monthly, seasonal, and annual precipitation values and Figs. 10-12 for monthly, seasonal, and annual temperature values. The tables highlight the commensurate central tendencies, variabilities, and dispersion values within the datasets and provide insight into the comparisons between the datasets.

**Table 7. Overall statistics for MOPEX (M) and CAMELS (C) precipitation totals by climate region. Bootstrapping mean values for each climate region and the lower and upper confidence limits are based on two-sided 95 % confidence interval and 10,000 replicates with replacement. Median, variance, standard deviation, and skew are based on the average of all values for each basin within a region.**

| | PRCP (mm) | Median C | Median M | Mean ±CI C | Mean ±CI M | Variance C | Variance M | St Dev C | St Dev M | Skew C | Skew M |
|---|---|---|---|---|---|---|---|---|---|---|---|
| ARID | Day | 0.00 | 0.06 | 1.76 ±0.06 | 1.78 ±0.06 | 33.12 | 32.78 | 5.50 | 5.51 | 5.18 | 6.82 |
| | Month | 36.32 | 37.25 | 53.65 ±10.59 | 54.13 ±10.22 | 2912.56 | 2697.72 | 52.86 | 51.03 | 1.55 | 1.52 |
| | Season | 142.71 | 142.06 | 159.08 ±22.75 | 160.56 ±22.03 | 11287.61 | 10923.90 | 104.63 | 102.62 | 0.85 | 0.83 |
| | Year | 644.74 | 646.79 | 643.74 ±182.57 | 649.56 ±165.36 | 37205.76 | 24725.11 | 185.53 | 180.40 | 0.14 | 0.13 |
| CONT | Day | 0.03 | 0.18 | 2.85 ±0.04 | 2.70 ±0.04 | 46.45 | 42.53 | 6.74 | 6.45 | 3.99 | 4.51 |
| | Month | 76.40 | 73.17 | 86.86 ±5.20 | 82.17 ±4.86 | 2918.08 | 2502.34 | 53.38 | 49.32 | 1.13 | 1.12 |
| | Season | 248.45 | 235.51 | 257.56 ±9.79 | 243.68 ±9.61 | 11252.31 | 9913.34 | 103.43 | 96.78 | 0.49 | 0.53 |
| | Year | 1039.78 | 978.47 | 1042.31 ±79.39 | 986.09 ±76.57 | 33962.67 | 29675.00 | 180.26 | 167.92 | 0.34 | 0.33 |
| TEMP | Day | 0.10 | 0.39 | 3.73 ±0.04 | 3.50 ±0.04 | 61.98 | 60.14 | 7.69 | 7.55 | 3.56 | 4.20 |
| | Month | 105.73 | 98.73 | 113.44 ±6.03 | 111.42 ±6.46 | 3484.74 | 3051.77 | 57.99 | 54.04 | 0.84 | 0.90 |
| | Season | 330.78 | 309.13 | 344.33 ±12.30 | 330.48 ±14.36 | 12668.97 | 11042.11 | 109.89 | 101.87 | 0.54 | 0.55 |
| | Year | 1364.52 | 1279.35 | 1393.13 ±122.75 | 1337.09 ±158.06 | 48010.82 | 42250.29 | 213.19 | 197.93 | 0.18 | 0.18 |

**Table 8. Overall statistics for MOPEX (M) and CAMELS (C) precipitation totals by climate region. Bootstrapping mean values for each climate region and the lower and upper confidence limits are based on two-sided 95 % confidence interval and 10,000 replicates with replacement. Median, variance, standard deviation, and skew are based on the average of all values for each basin within a region.**

| | TAIR (°C) | Median C | Median M | Mean ±CI C | Mean ±CI M | Variance C | Variance M | St Dev C | St Dev M | Skew C | Skew M |
|---|---|---|---|---|---|---|---|---|---|---|---|
| ARID | Day | 14.77 | 15.33 | 14.39 ±0.10 | 15.01 ±0.10 | 70.17 | 65.68 | 8.17 | 7.93 | -0.32 | -0.25 |
| | Month | 14.57 | 15.08 | 14.35 ±1.07 | 14.97 ±1.04 | 56.51 | 53.95 | 7.33 | 7.18 | -0.05 | -0.02 |
| | Season | 14.49 | 14.98 | 14.39 ±1.05 | 15.01 ±1.04 | 46.06 | 43.94 | 6.61 | 6.48 | -0.04 | -0.01 |
| | Year | 14.36 | 14.99 | 14.39 ±4.44 | 15.01 ±4.29 | 0.59 | 0.45 | 0.74 | 0.65 | 0.22 | 0.12 |
| CONT | Day | 11.32 | 11.37 | 10.61 ±0.06 | 10.77 ±0.06 | 103.63 | 98.58 | 10.13 | 9.89 | -0.34 | -0.26 |
| | Month | 11.01 | 11.13 | 10.57 ±0.37 | 10.72 ±0.34 | 84.44 | 81.65 | 9.14 | 8.99 | -0.12 | -0.08 |
| | Season | 10.88 | 10.94 | 10.60 ±0.36 | 10.75 ±0.32 | 70.25 | 67.79 | 8.34 | 8.20 | -0.10 | -0.07 |
| | Year | 10.65 | 10.81 | 10.62 ±1.33 | 10.77 ±1.19 | 0.59 | 0.51 | 0.76 | 0.71 | 0.04 | 0.06 |
| TEMP | Day | 11.14 | 11.35 | 10.57 ±0.05 | 10.92 ±0.04 | 80.52 | 76.14 | 8.94 | 8.70 | -0.36 | -0.28 |
| | Month | 10.56 | 10.93 | 10.47 ±0.40 | 10.89 ±0.38 | 63.60 | 61.69 | 7.95 | 7.83 | -0.08 | -0.05 |
| | Season | 10.62 | 10.94 | 10.49 ±0.38 | 10.92 ±0.36 | 52.41 | 50.88 | 7.21 | 7.11 | -0.05 | -0.03 |
| | Year | 10.52 | 10.91 | 10.51 ±1.64 | 10.94 ±1.57 | 0.46 | 0.38 | 0.67 | 0.62 | 0.10 | 0.14 |

To assess the magnitude of disparities between CAMELS and MOPEX, we also examined the mean absolute error (MAE), with temperate regions exhibiting greatest MAE in precipitation (Fig. 20), and arid regions greatest in temperature (Fig. 21). The overall error considers both positive and negative differences equally, with the magnitude escalating from 1.30 mm day$^{-1}$ in arid regions to 2.70 mm day$^{-1}$ in continental and 3.19 mm day$^{-1}$ in temperate regions for daily precipitation. Monthly precipitation MAE ranges from 2.37 mm month$^{-1}$ to 5.72 mm month$^{-1}$ in arid regions, 3.74 mm month$^{-1}$ to 6.80 mm month$^{-1}$ for continental regions, and 5.48 mm month$^{-1}$ to 9.62 mm month$^{-1}$ in temperate regions (Fig. 20a). Seasonal MAE ranges from 1.79 mm season$^{-1}$ to 7.68 mm season$^{-1}$, 12.13 mm season$^{-1}$ to 15.66 mm season$^{-1}$, and 16.78 mm season$^{-1}$ to 24.88 mm season$^{-1}$ for arid, continental, and temperate regions (Fig. 20b). Annual MAE ranges from 1.62 mm year$^{-1}$ to 46.91 mm year$^{-1}$, 27.84 mm year$^{-1}$ to 89.23 mm year$^{-1}$, and 51.46 mm year$^{-1}$ to 112.03 mm year$^{-1}$ for arid, continental, and temperate regions (Fig. 20c).

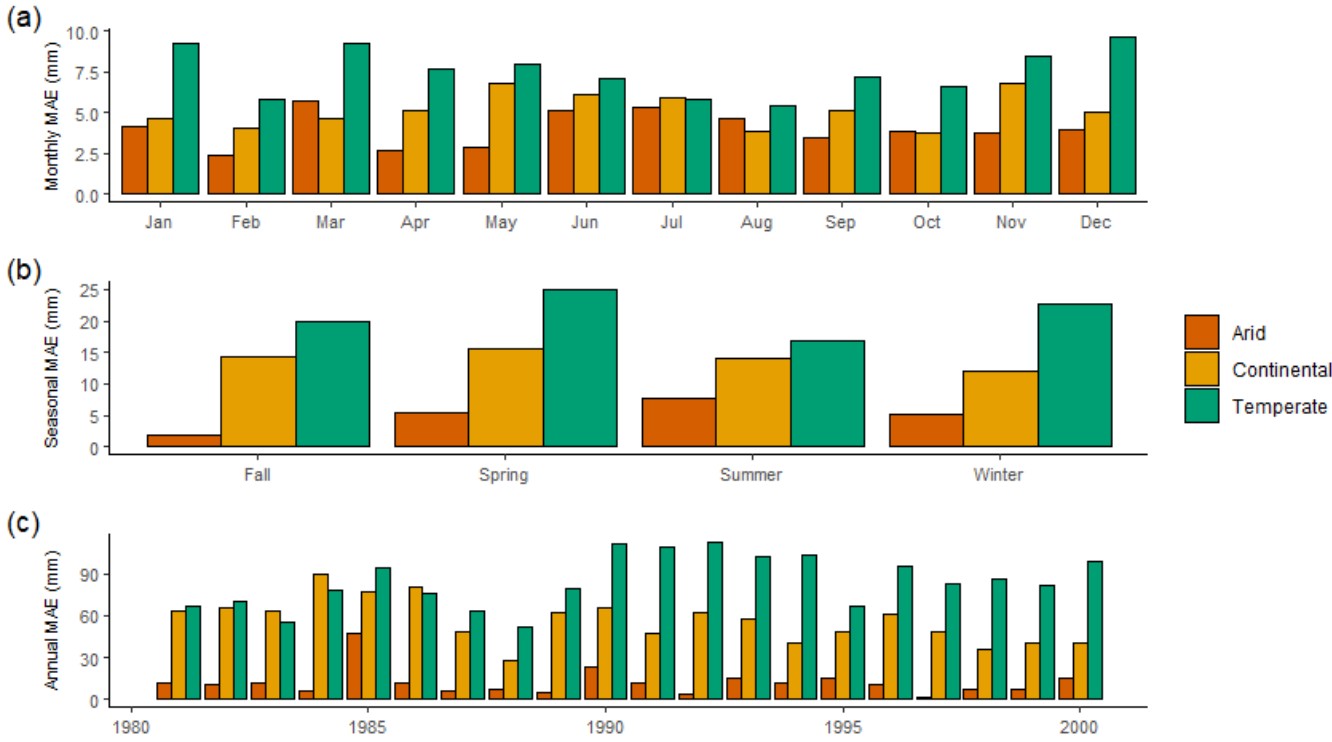

Precipitation mean absolute error

**Figure 20. Mean absolute error for a) monthly, b) seasonal, and c) annual precipitation. MAE is based on mean totals between all basins within a climate region (arid, continental, and temperate).**

Daily temperature MAE averages 1.01 ˚C day$^{-1}$ in arid regions, 0.87 ˚C day$^{-1}$ in continental, and 0.82 ˚C day$^{-1}$ in temperate regions. Monthly temperature MAE ranges from 0.41 ˚C month$^{-1}$ to 0.95 ˚C month$^{-1}$, 0.04 ˚C month$^{-1}$ to 0.54 ˚C month$^{-1}$, and 0.24 ˚C month$^{-1}$ to 0.64 ˚C month$^{-1}$ for arid, continental, and temperate regions respectively (Fig. 21a). Seasonal MAE for

temperature ranges from 0.51 ˚C season$^{-1}$ to 0.85 ˚C season$^{-1}$, 0.03 ˚C season$^{-1}$ to 0.41 ˚C season$^{-1}$, and 0.25 ˚C season$^{-1}$ to 0.52 ˚C season$^{-1}$ for arid, continental, and temperate regions (Fig. 21b). Annual temperature MAE ranges from 0.43 ˚C year$^{-1}$ to 0.86 ˚C year$^{-1}$, 0.07 ˚C year$^{-1}$ to 0.29 ˚C year$^{-1}$, and 0.23 ˚C year$^{-1}$ to 0.47 ˚C year$^{-1}$ for arid, continental, and temperate regions (Fig. 21c).

Temperature mean absolute error

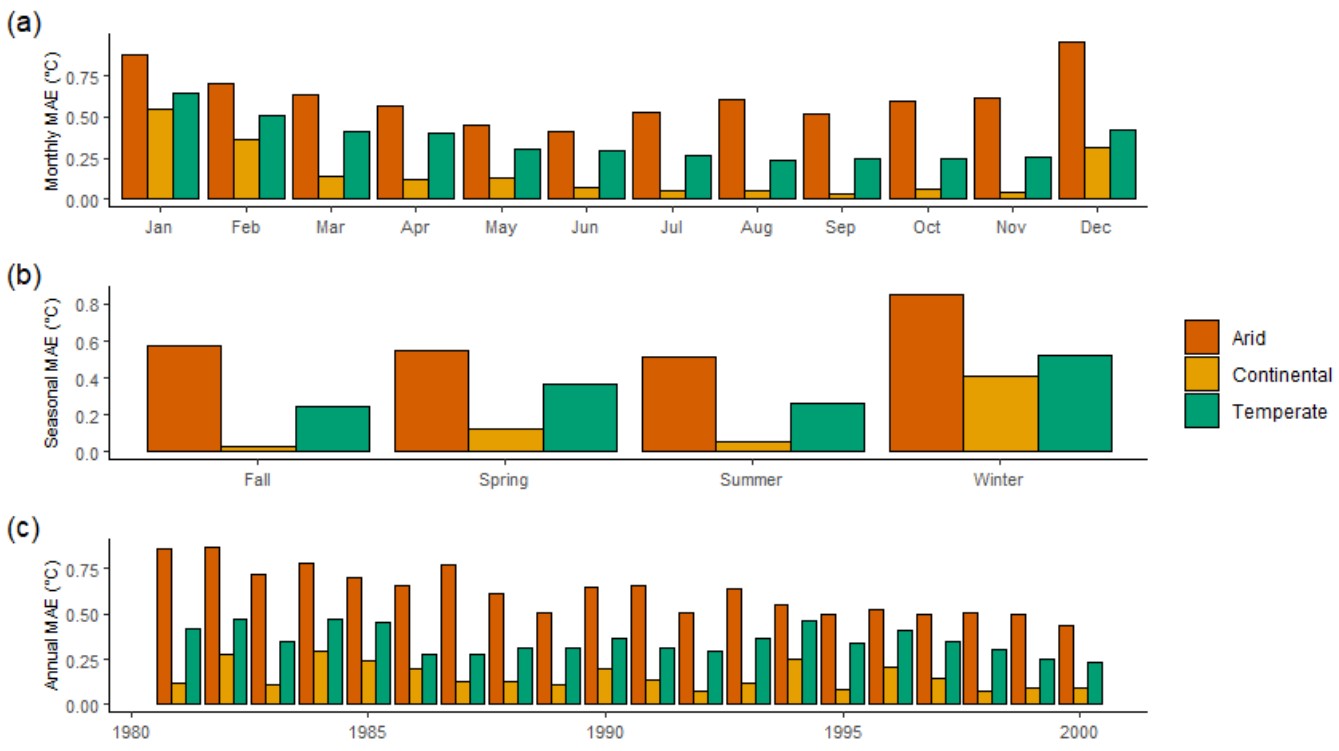

**Figure 21. Mean absolute error for a) monthly, b) seasonal, and c) annual temperature. MAE is based on means between all basins within a climate region (arid, continental, and temperate).**

The statistical results for all regions are summarized in Table 9 and are calculated over all days, months, seasons, and water years (refer to Table 3). Overall statistics remove the observed fluctuations in monthly, seasonal, and annual data but provide a generalized value by climate region. The margin of error (MOE) was derived from the standard error (SE) of the difference of the means and coincides with the bootstrapping results. Arid regions have the largest MOE for precipitation and temperature.

**Table 9. Statistical results for comparisons between CAMELS and MOPEX values for all basins within a climate region. Analyses were conducted over the total number of values. Bias represents mean CAMELS minus mean MOPEX. Standard error (SE), margin of error (MOE), mean absolute error (MAE), and Spearman rank ($R^2$) are also based on mean values.**

| | | Precipitation (mm) | | | | | Temperature (˚C) | | | | |
|---|---|---|---|---|---|---|---|---|---|---|---|
| | | Bias | SE | MOE | MAE | $R^2$ | Bias | SE | MOE | MAE | $R^2$ |
| ARID | Day | -0.02 | 0.04 | ± 0.08 | 1.30 | 0.665 | -0.62 | 0.07 | ± 0.14 | 1.01 | 0.992 |
| | Month | -0.48 | 2.24 | ± 4.38 | 3.99 | 0.975 | -0.62 | 0.37 | ± 0.72 | 0.62 | 0.996 |
| | Season | -1.45 | 8.02 | ± 15.71 | 5.02 | 0.977 | -0.62 | 0.59 | ± 1.16 | 0.62 | 0.993 |
| | Year | -5.82 | 35.90 | ± 70.37 | 12.09 | 0.964 | -0.62 | 0.72 | ± 1.41 | 0.62 | 0.986 |
| CONT | Day | 0.15 | 0.03 | ± 0.05 | 2.70 | 0.571 | -0.16 | 0.04 | ± 0.08 | 0.87 | 0.994 |
| | Month | 4.69 | 1.13 | ± 2.22 | 5.16 | 0.973 | -0.15 | 0.20 | ±0.40 | 0.16 | 0.999 |
| | Season | 13.88 | 3.92 | ± 7.69 | 14.05 | 0.978 | -0.15 | 0.33 | ±0.64 | 0.15 | 0.998 |
| | Year | 56.21 | 14.95 | ± 29.31 | 56.22 | 0.956 | -0.15 | 0.22 | ±0.42 | 0.15 | 0.989 |
| TEMP | Day | 0.23 | 0.03 | ± 0.05 | 3.19 | 0.609 | -0.35 | 0.03 | ± 0.06 | 0.82 | 0.995 |
| | Month | 4.67 | 1.11 | ± 2.18 | 7.52 | 0.962 | -0.36 | 0.17 | ± 0.33 | 0.35 | 0.999 |
| | Season | 13.85 | 3.88 | ± 7.60 | 21.07 | 0.964 | -0.35 | 0.27 | ± 0.52 | 0.35 | 0.998 |
| | Year | 56.04 | 18.99 | ± 37.22 | 84.28 | 0.949 | -0.35 | 0.27 | ± 0.53 | 0.35 | 0.996 |

745 Fligner-Killeen's test for equality of variances indicated that arid regions are the most similar in precipitation variance for all temporal aggregations except daily (Table 10). Statistically significant differences between variances were found for continental regions (daily, monthly, seasonally) and temperate regions (daily and monthly) but not on an annual basis. Temperature values are more consistent with statistically significant differences between variances indicated for daily values only. These results are corroborated by observations previously presented and outlined in Sect. 4.1, as shown in Tables 4 and 5, and Figs. 7 through 12.

**Table 10. Fligner-Killeen's test for homogeneity of variance results. Df1 is the number of groups minus 1, statistic is Chi-squared. P-values are reported as * p < 0.05, ** p < 0.01, *** p < 0.001.**

| | | Precipitation | | | Temperature | | |
|---|---|---|---|---|---|---|---|
| | | df1 | Chi-squared | p | df1 | Chi-squared | p |
| ARID | Day | 1 | 9661.2 | **<0.001*** | 1 | 38.22 | **<0.001*** |
| | Month | 1 | 0.15 | 0.70 | 1 | 1.16 | 0.28 |
| | Season | 1 | 0.10 | 0.75 | 1 | 1.00 | 0.32 |
| | Year | 1 | 0.09 | 0.76 | 1 | 1.35 | 0.25 |
| CONT | Day | 1 | 14432 | **<0.001*** | 1 | 60.52 | **<0.001*** |
| | Month | 1 | 23.24 | **<0.001*** | 1 | 2.86 | 0.09 |
| | Season | 1 | 4.96 | **0.03*** | 1 | 1.98 | 0.16 |
| | Year | 1 | 0.69 | 0.41 | 1 | 2.62 | 0.11 |
| TEMP | Day | 1 | 11096 | **<0.001*** | 1 | 89.83 | **<0.001*** |
| | Month | 1 | 21.54 | **<0.001*** | 1 | 3.32 | 0.07 |
| | Season | 1 | 2.68 | 0.10 | 1 | 1.86 | 0.17 |
| | Year | 1 | 0.67 | 0.41 | 1 | 2.69 | 0.10 |

Differences in the mean values, evaluated by Welch's t-test, indicated that there were no statistically significant differences in the 755 mean for daily, monthly, seasonal, and annual arid precipitation (Table 11). Despite the largest variance, arid regions are the most similar with the smallest amount of bias between means. Differences in mean precipitation values in continental and temperate

regions are statistically significant on all temporal scales, however, the largest difference between mean values is only 6.36 % (daily temperate). Temperature mean differences are only statistically significant at daily aggregations for all climate regions, with the exception of monthly temperature.

**Table 11. Welch's t-test for comparison of means with unequal variance results. The n represents the number of values, df is calculated degrees of freedom, statistic is calculated t.  P-values are reported as * p < 0.05, ** p < 0.01, *** p < 0.001.**

| | | Precipitation | | | | Temperature | | | |
|---|---|---|---|---|---|---|---|---|---|
| | | n | df | statistic | p | n | df | statistic | p |
| ARID | Day | 36525 | 73045.58 | -0.37 | 0.71 | 36525 | 72950.21 | -8.70 | **<0.001*** |
| | Month | 1200 | 2393.65 | -0.22 | 0.83 | 1200 | 2395.78 | -1.70 | 0.09 |
| | Season | 400 | 797.34 | -0.18 | 0.85 | 400 | 797.19 | -1.05 | 0.29 |
| | Year | 100 | 197.15 | -0.16 | 0.87 | 100 | 197.54 | -0.86 | 0.39 |
| CONT | Day | 131490 | 262467.77 | 5.91 | **<0.001*** | 131490 | 262734.92 | -3.76 | **<0.001*** |
| | Month | 4320 | 8589.14 | 4.14 | **<0.001*** | 4320 | 8632.98 | -0.75 | 0.45 |
| | Season | 1440 | 2867.28 | 3.58 | **<0.001*** | 1440 | 2875.99 | -0.47 | 0.64 |
| | Year | 360 | 715.67 | 3.76 | **<0.001*** | 360 | 711.07 | -0.71 | 0.48 |
| TEMP | Day | 175320 | 350554.08 | 8.72 | **<0.001*** | 175320 | 350322.75 | -10.64 | **<0.001*** |
| | Month | 5760 | 11481.67 | 6.32 | **<0.001*** | 5760 | 11513.38 | -2.11 | **0.03*** |
| | Season | 1920 | 3829.89 | 5.43 | **<0.001*** | 1920 | 3836.39 | -1.32 | 0.19 |
| | Year | 480 | 957.97 | 4.44 | **<0.001*** | 480 | 956.60 | -1.29 | 0.20 |

The non-parametric Wilcoxon signed-rank test (Helsel et al., 2020) was also conducted to evaluate the median differences and the
765 results indicated statistically significant differences for daily precipitation in all regions, and for monthly, seasonal, and annual precipitation in continental and temperate regions. Temperature median differences were only statistically significant for daily values.

As previously noted, there are no discrepancies in runoff between MOPEX and CAMELS datasets because both contain identical
daily streamflow values sourced from the USGS. However, evapotranspiration estimates derived from the water balance approach will differ due to variations in precipitation, since runoff remains consistent across the datasets. Runoff and water balance derived evapotranspiration were included in correlation analyses to evaluate the relationships among all variables for both datasets and to determine consistency in the strength and direction of their associations.

Daily precipitation Spearman Rank correlation values between CAMELS (_C) and MOPEX (_M) ranged from 0.58 to 0.74, 0.48 to 0.86, and 0.46 to 0.88 for arid, continental, and temperate regions respectively (Fig. 22). The highest precipitation correlation values were observed for monthly and seasonal aggregations (red cells, Fig. 22), with annual values following closely. Monthly precipitation correlations are the lowest in July and August for all regions (0.84 to 0.91). Monthly and seasonal aggregations are the most consistent between MOPEX and CAMELS, followed by annual and then daily for precipitation and temperature.
Temperature shows a high similarity between MOPEX and CAMELS for all temporal aggregations and regions, ranging from 0.99 to 1.0. Correlations between runoff and precipitation are positive for all regions and temporal aggregations in both datasets with the largest difference of 0.08 in daily continental (0.14 for MOPEX, 0.22 for CAMELS).  Water balance evapotranspiration values show improved agreement, greater than 0.90, for monthly, seasonal, and annual aggregations. Daily evapotranspiration coefficients are between 0.59 to 0.64, indicating less consistency between MOPEX and CAMELS.

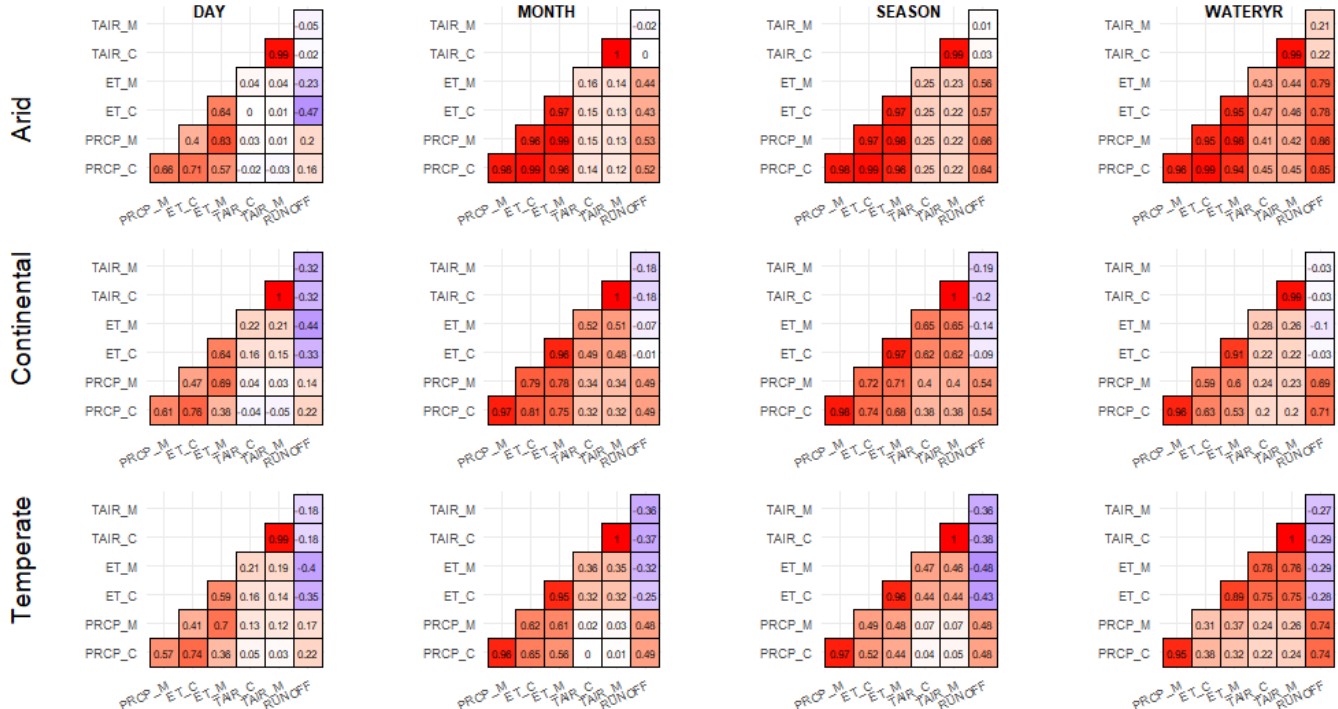

**Figure 22. Spearman rank correlation values between CAMELS (_C) and MOPEX (_M) datasets for precipitation, temperature, and water balance calculated evapotranspiration. Runoff represents both datasets.**

Runoff efficiency is the amount of precipitation that becomes runoff and can be used to evaluate trends and climate impact. This coefficient provides an additional metric of dataset compatibility. The annual efficiency for each basin was determined for CAMELS and MOPEX using total precipitation and total runoff and then plotted, resulting in an $R^2$ value of 0.988 for all climate regions combined (Fig. 23). This correlation was conducted to illustrate the annual compatibility of the datasets and the ability of both to convey consistent attributes among watersheds for derived parameters, such as runoff efficiency.

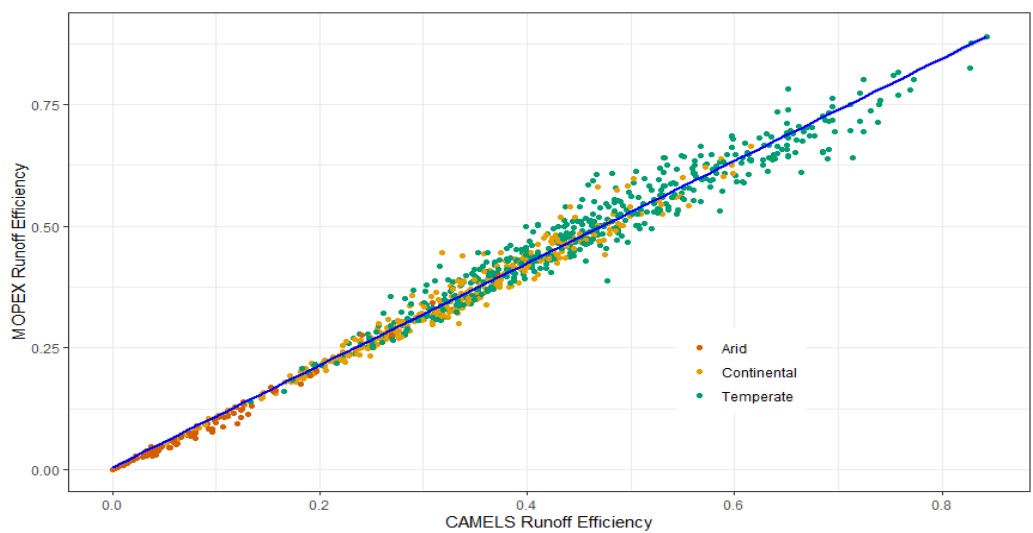

**Figure 23.  Annual runoff coefficient (runoff/precipitation) for each basin. Colored points represent climate region (arid, continental, and temperate). The blue line indicates the best linear fit.**

**4.3 Evaluation of precipitation and temperature extremes between datasets**

While data extremes were not the focus of this study, a few precipitation and temperature extreme indices were evaluated for CAMELS and MOPEX values. The number of heavy precipitation days, where daily precipitation is greater than or equal to 10 mm, are more prolific in CAMELS, consistent with the wet bias (Fig. 24a). Despite the difference in the number of days, the two datasets show the same trends over time from 1981 to 2000. The number of dry days (precipitation < 1mm) per year are greater in CAMELS for all climate regions, with the largest discrepancies in arid regions (Fig. 24b). This study has shown that CAMELS has a wet bias for continental and temperate regions and MOPEX has a wet bias for arid regions (Fig. 17). The differences in the number of dry days show that CAMELS daily precipitation values are overall, larger than MOPEX values (Fig. 24b).

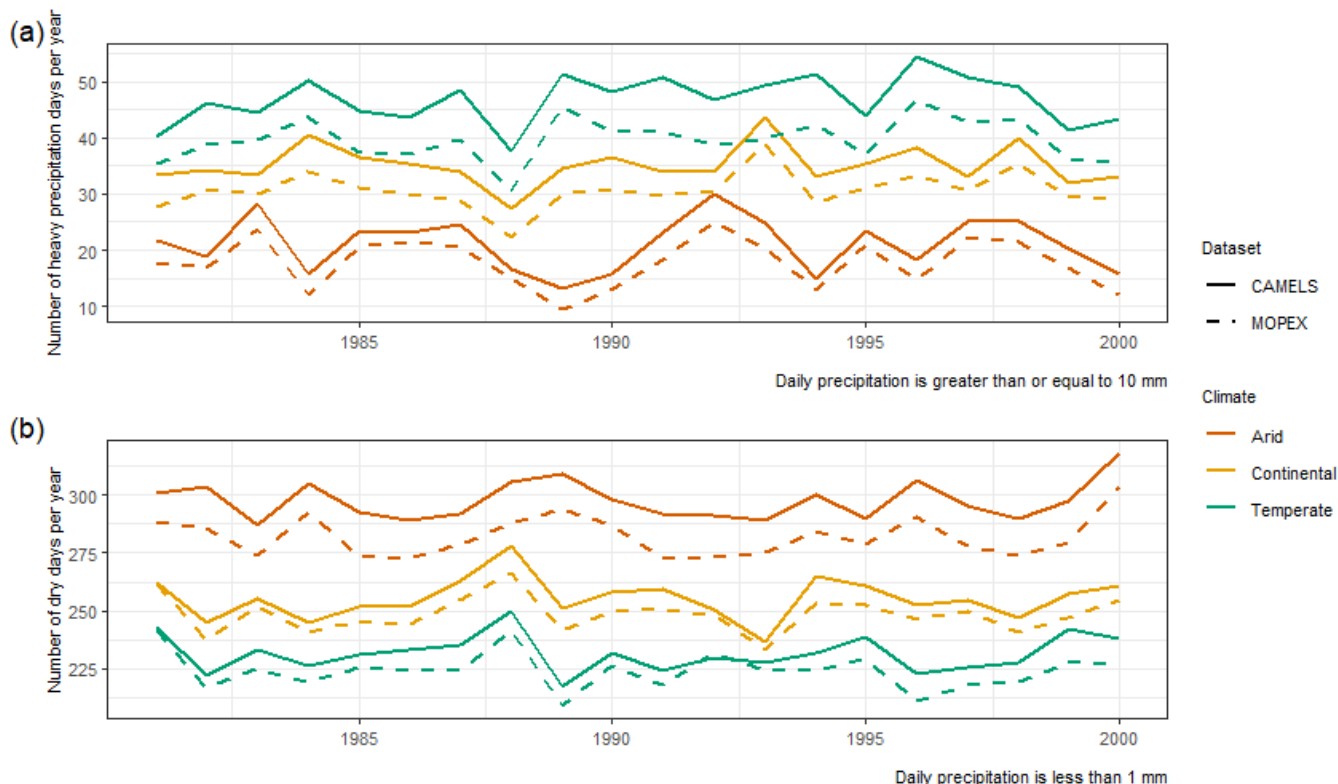

**Figure 24. Precipitation indices which show the a) annual count of heavy precipitation days where precipitation is ≥ 10 mm and b) annual count of dry days where precipitation amount is < 1mm. Colors represent the climate region (arid, continental, and temperate), dashed lines represent MOPEX, and solid lines represent CAMELS.**

The extremely wet day rainfall, R99p, represents the annual total precipitation when daily rainfall is greater than the $99^{th}$ percentile and when plotted for both datasets by climate region, very similar trends are observed (Fig. 25). In a broad temporal context, analysis consistently shows that precipitation values tend to be larger in CAMELS, regardless of the temporal scale considered beyond paired daily values. This pattern is observed in monthly, seasonal, and annual aggregations as well as summarized daily mean for continental and temperate regions.

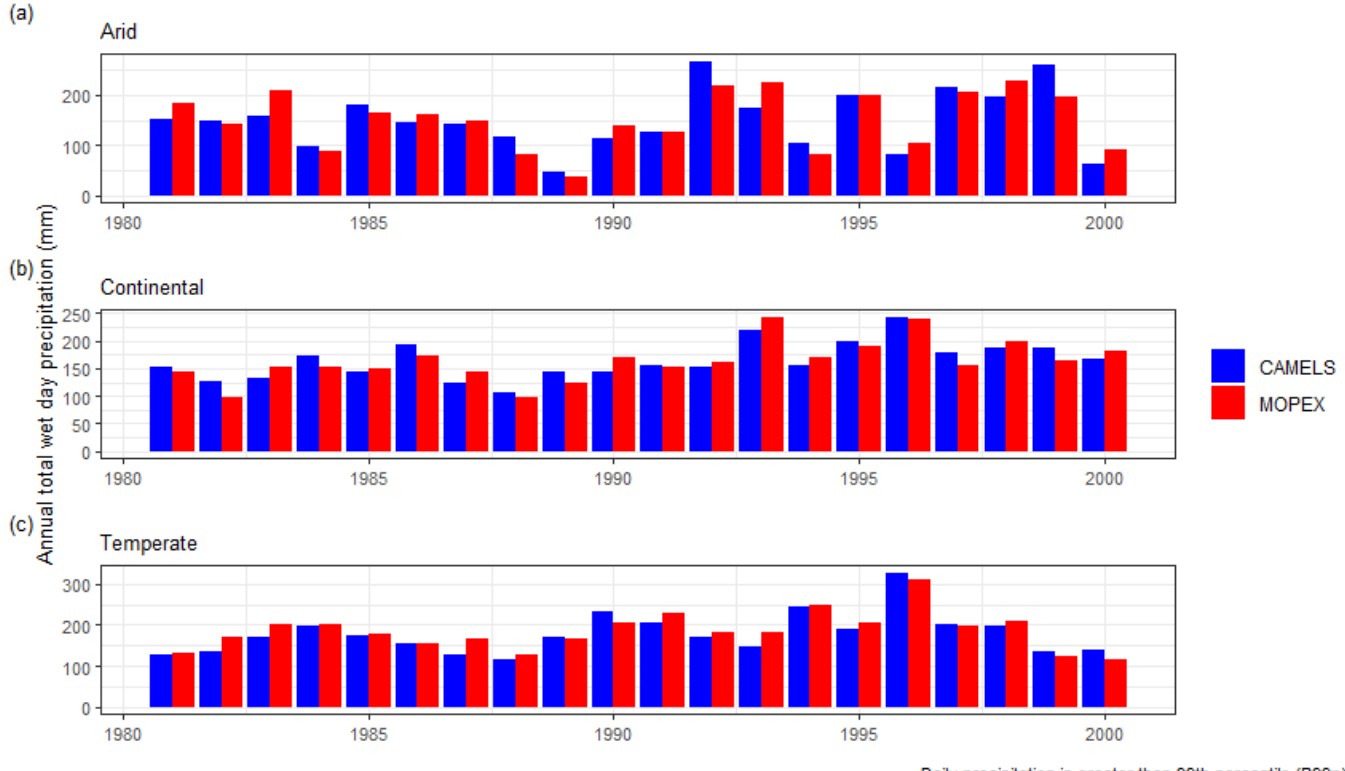

**Figure 25. Annual total precipitation when daily precipitation amount on a wet day (≥ 1mm) is greater than the 99th percentile for a) arid, b) continental, and c) temperate regions. Colors (blue and red) represent the dataset.**

In contrast, temperature values exhibit a different trend, with MOPEX consistently showing larger values irrespective of the temporal aggregation or climate region (Fig. 18). The number of frost days (Fig. 26a) indicate the annual count per year where temperature falls below 0 ˚C. CAMELS has a greater number of cold days which corresponds to warmer MOPEX bias. The warm MOPEX bias is most prevalent in arid regions when evaluating the number of summer days per year, the annual count of days with temperatures above 25 ˚C (Fig. 26b).

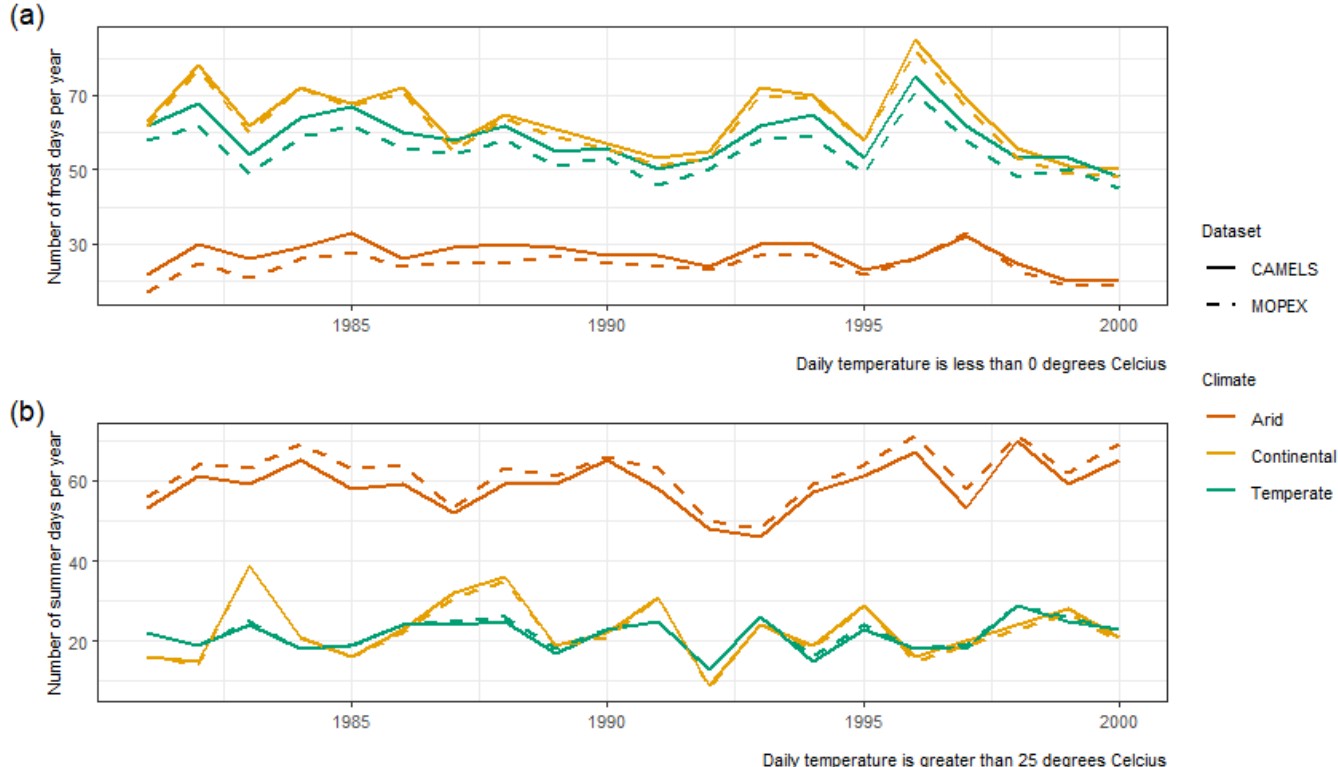

**Figure 26. Temperature indices which show the a) annual count of frost days where temperature is < 0 ˚C and b) annual count of summer days where temperature is > 25 ˚C. Colors represent the climate region (arid, continental, and temperate), dashed lines represent MOPEX, and solid lines represent CAMELS.**

## 4.4 Validation

Hydrologic models are used to simulate real world processes and range from simple conceptual models to complex physically based models. Choosing a suitable model is highly dependent on the purpose and scale. The input data required depends on the spatial and temporal distributions evaluated in a model, but precipitation and temperature are fundamental. Inherent biases in input

data can skew modeling results. Machine learning (ML) was used instead of hydrologic models (i.e. SWAT, VIC, SAC-SMA) because ML models provide a data-driven, model-agnostic approach that focuses on the relationships between inputs and outputs without relying on predefined process-based assumptions (Herrera et al., 2022). Four machine learning models were used to predict runoff at daily, monthly, seasonal, and annual scales for MOPEX and CAMELS. The objective is not to determine model suitability, rather evaluate the performance of each dataset.  The RMSE, MAE, R2, and bias of predicted versus observed runoff

serve as dataset comparisons.

On a daily scale, CAMELS has a slightly lower RMSE and MAE than MOPEX for all regions and a better R2, although the values are quite low, less than 0.3, shown in Table 12. A good fit is not expected with daily data which will have multiple zero values for precipitation.

**Table 12. Machine learning model metrics for predicted versus observed total daily runoff using total daily precipitation and mean daily temperature data as inputs for CAMELS (C) and MOPEX (M).**

| Day | ML Model | RMSE C | RMSE M | MAE C | MAE M | R2 C | R2 M | Bias C | Bias M |
|---|---|---|---|---|---|---|---|---|---|
| ARID | Linear Regression | 0.73 | 0.82 | 0.21 | 0.21 | 0.15 | 0.08 | -0.01 | 0.00 |
| | Random Forest | 0.70 | 0.84 | 0.20 | 0.22 | 0.21 | 0.04 | -0.01 | 0.01 |
| | Gradient Boosting | 0.69 | 0.82 | 0.20 | 0.21 | 0.25 | 0.09 | 0.00 | 0.00 |
| | SVR | 0.77 | 0.85 | 0.17 | 0.17 | 0.12 | 0.06 | -0.06 | -0.06 |
| CONT | Linear Regression | 1.99 | 2.09 | 0.89 | 0.93 | 0.21 | 0.06 | 0.00 | -0.01 |
| | Random Forest | 1.95 | 2.14 | 0.85 | 0.95 | 0.25 | 0.05 | 0.01 | 0.01 |
| | Gradient Boosting | 1.89 | 2.06 | 0.83 | 0.90 | 0.29 | 0.09 | -0.01 | -0.01 |
| | SVR | 1.99 | 2.15 | 0.73 | 0.77 | 0.29 | 0.08 | -0.39 | -0.46 |
| TEMP | Linear Regression | 2.62 | 2.83 | 1.41 | 1.48 | 0.17 | 0.06 | 0.00 | -0.01 |
| | Random Forest | 2.60 | 2.87 | 1.38 | 1.48 | 0.19 | 0.07 | 0.01 | 0.01 |
| | Gradient Boosting | 2.51 | 2.77 | 1.34 | 1.41 | 0.24 | 0.10 | -0.01 | -0.01 |
| | SVR | 2.63 | 2.88 | 1.21 | 1.26 | 0.23 | 0.11 | -0.60 | -0.69 |

At the monthly aggregation, Table 13, CAMELS narrowly outperforms MOPEX with lower RMSE and MAE values. The R2 values are extremely similar between datasets in all regions, and both exhibit the same positive biases with all ML models except for SVR, which underpredicts runoff and results in negative biases for both datasets. The results indicate that the predictive performance of the models is very similar across both datasets, suggesting a high degree of consistency between them.

**Table 13. Machine learning model metrics for predicted versus observed total monthly runoff using total monthly precipitation and mean monthly temperature data as inputs for CAMELS (C) and MOPEX (M).**

| Month | ML Model | RMSE C | RMSE M | MAE C | MAE M | R2 C | R2 M | Bias C | Bias M |
|---|---|---|---|---|---|---|---|---|---|
| ARID | Linear Regression | 7.73 | 10.64 | 5.12 | 5.94 | 0.37 | 0.37 | 0.62 | 0.64 |
| | Random Forest | 8.08 | 10.82 | 3.82 | 5.32 | 0.31 | 0.37 | 0.12 | 0.30 |
| | Gradient Boosting | 8.15 | 11.13 | 4.00 | 5.54 | 0.33 | 0.35 | 0.37 | 0.44 |
| | SVR | 7.00 | 10.65 | 3.48 | 4.34 | 0.36 | 0.39 | -0.91 | -1.42 |
| CONT | Linear Regression | 23.42 | 23.45 | 17.16 | 16.95 | 0.40 | 0.41 | 0.42 | 0.31 |
| | Random Forest | 22.61 | 22.81 | 16.05 | 16.03 | 0.45 | 0.44 | 0.48 | 0.29 |
| | Gradient Boosting | 21.49 | 21.87 | 15.40 | 15.56 | 0.50 | 0.48 | 0.45 | 0.51 |
| | SVR | 21.69 | 22.02 | 14.80 | 14.93 | 0.50 | 0.49 | -2.74 | -2.99 |
| TEMP | Linear Regression | 41.55 | 42.09 | 28.96 | 29.27 | 0.32 | 0.31 | 0.74 | 0.78 |
| | Random Forest | 41.01 | 40.97 | 28.13 | 28.18 | 0.36 | 0.36 | 1.39 | 1.27 |
| | Gradient Boosting | 38.73 | 38.91 | 26.52 | 26.84 | 0.41 | 0.41 | 1.21 | 0.75 |
| | SVR | 38.71 | 39.21 | 25.29 | 25.85 | 0.42 | 0.41 | -5.00 | -5.20 |

Seasonally, Table 14, the main discrepancies between the datasets are in continental regions, where CAMELS runoff predictions are lower than those from MOPEX by approximately 4 to 5 mm. This difference, while evident, is relatively small and may not have significant implications for broader regional or long-term studies. For instance, seasonal runoff values in continental regions range from 0.3 mm in one basin (JJA 1988) to 423.88 mm (MAM 1996) in another basin. This effect of these biases would be more pronounced for basins with very little runoff in a specific season, but this issue is not unique to these datasets. Any dataset used on such a fine, basin-specific scale may exhibit similar biases.

**Table 14. Machine learning model metrics for predicted versus observed total seasonal runoff using total seasonal precipitation and mean seasonal temperature data as inputs for CAMELS (C) and MOPEX (M).**

| Season | ML Model | RMSE C | RMSE M | MAE C | MAE M | R2 C | R2 M | Bias C | Bias M |
|---|---|---|---|---|---|---|---|---|---|
| ARID | Linear Regression | 21.60 | 22.66 | 14.52 | 14.65 | 0.37 | 0.31 | 0.48 | 0.41 |
|  | Random Forest | 16.96 | 20.25 | 10.36 | 11.27 | 0.61 | 0.55 | 1.29 | 1.02 |
|  | Gradient Boosting | 18.25 | 21.17 | 11.26 | 12.15 | 0.59 | 0.54 | 2.27 | 1.32 |
|  | SVR | 18.63 | 21.33 | 9.27 | 9.49 | 0.53 | 0.39 | -2.49 | -4.35 |
| CONT | Linear Regression | 59.11 | 51.78 | 43.00 | 38.57 | 0.44 | 0.43 | -4.41 | -0.12 |
|  | Random Forest | 54.75 | 54.91 | 39.47 | 40.55 | 0.52 | 0.58 | -4.45 | -0.99 |
|  | Gradient Boosting | 53.19 | 50.00 | 39.01 | 36.76 | 0.55 | 0.47 | -5.00 | 0.07 |
|  | SVR | 53.93 | 50.94 | 37.02 | 35.79 | 0.55 | 0.47 | -9.69 | -4.22 |
| TEMP | Linear Regression | 97.33 | 96.28 | 70.64 | 70.71 | 0.31 | 0.33 | 0.66 | 0.46 |
|  | Random Forest | 92.10 | 92.02 | 67.26 | 65.56 | 0.40 | 0.40 | 2.23 | 2.28 |
|  | Gradient Boosting | 87.83 | 88.08 | 66.09 | 63.93 | 0.44 | 0.44 | 2.54 | 1.92 |
|  | SVR | 91.48 | 91.78 | 64.44 | 64.54 | 0.41 | 0.41 | -7.17 | -9.29 |

The differences in precipitation and temperature between MOPEX and CAMELS become more relevant depending on the scale and objective of the study. For daily or single-month analyses, as well as for very specific seasons, the datasets may not be directly comparable. However, as with any modeling approach, results come with inherent uncertainty, which should be acknowledged when presenting findings. Model results should be accompanied by an uncertainty estimate, reflecting potential biases or discrepancies. Bias correction is an essential part of any modeling process, typically done during the calibration phase (Lehner et al., 2023). In this context, the warm bias in MOPEX and the wet bias in CAMELS are important only when focusing on very fine, basin-specific scales. On larger temporal or spatial scales, these biases are less likely to significantly affect the conclusions, making these two datasets comparable for general hydrological or climate studies. At an annual scale, Table 15, MOPEX and CAMELS have improved R2 and the same predicted runoff biases despite the overall warm MOPEX temperature biases and wet CAMELS precipitation biases present in the data. The similarity in predicted runoff demonstrates the compatibility between the two datasets and that no corrections to the raw data are required at an annual scale.

**Table 15. Machine learning model metrics for predicted versus observed total annual runoff using total annual precipitation and mean annual temperature data as inputs for CAMELS (C) and MOPEX (M).**

| Year | ML Model | RMSE C | RMSE M | MAE C | MAE M | R2 C | R2 M | Bias C | Bias M |
|---|---|---|---|---|---|---|---|---|---|
| ARID | Linear Regression | 33.57 | 32.52 | 28.63 | 26.97 | 0.79 | 0.78 | 17.72 | 19.03 |
|  | Random Forest | 31.82 | 34.20 | 23.87 | 25.44 | 0.58 | 0.60 | 10.86 | 13.29 |
|  | Gradient Boosting | 35.33 | 30.54 | 26.30 | 22.16 | 0.68 | 0.83 | 17.85 | 21.22 |
|  | SVR | 20.18 | 21.24 | 15.95 | 16.53 | 0.81 | 0.81 | 4.27 | 8.06 |
| CONT | Linear Regression | 85.78 | 91.05 | 77.12 | 77.10 | 0.69 | 0.67 | -6.11 | -9.69 |
|  | Random Forest | 91.53 | 94.40 | 82.25 | 77.77 | 0.64 | 0.65 | -10.76 | -10.76 |
|  | Gradient Boosting | 89.53 | 95.91 | 83.76 | 80.24 | 0.64 | 0.64 | -15.48 | -15.15 |
|  | SVR | 80.13 | 85.16 | 74.19 | 70.84 | 0.73 | 0.73 | -11.10 | -15.48 |
| TEMP | Linear Regression | 106.89 | 113.85 | 82.26 | 91.08 | 0.88 | 0.87 | -19.61 | -19.52 |
|  | Random Forest | 121.34 | 126.85 | 92.93 | 103.58 | 0.85 | 0.83 | -17.52 | -23.33 |
|  | Gradient Boosting | 109.74 | 121.89 | 85.29 | 95.15 | 0.87 | 0.84 | -20.21 | -28.71 |
|  | SVR | 101.84 | 118.68 | 79.35 | 94.11 | 0.89 | 0.86 | -19.59 | -33.60 |

Total annual observed runoff is plotted against the predicted runoff for all four ML models in Fig. 27 with a 1:1 reference line. In all regions, MOPEX and CAMELS exhibit similar visual patterns and alignment of the points. In arid regions (Fig. 27a, b), both datasets show distinct clusters of low and high runoff values, reflecting greater variability and defined wet and dry periods. In contrast, continental (Fig. 27c, d) and temperate regions (Fig. 27e, f) display a more even distribution of runoff throughout the year, with both datasets capturing this behavior.

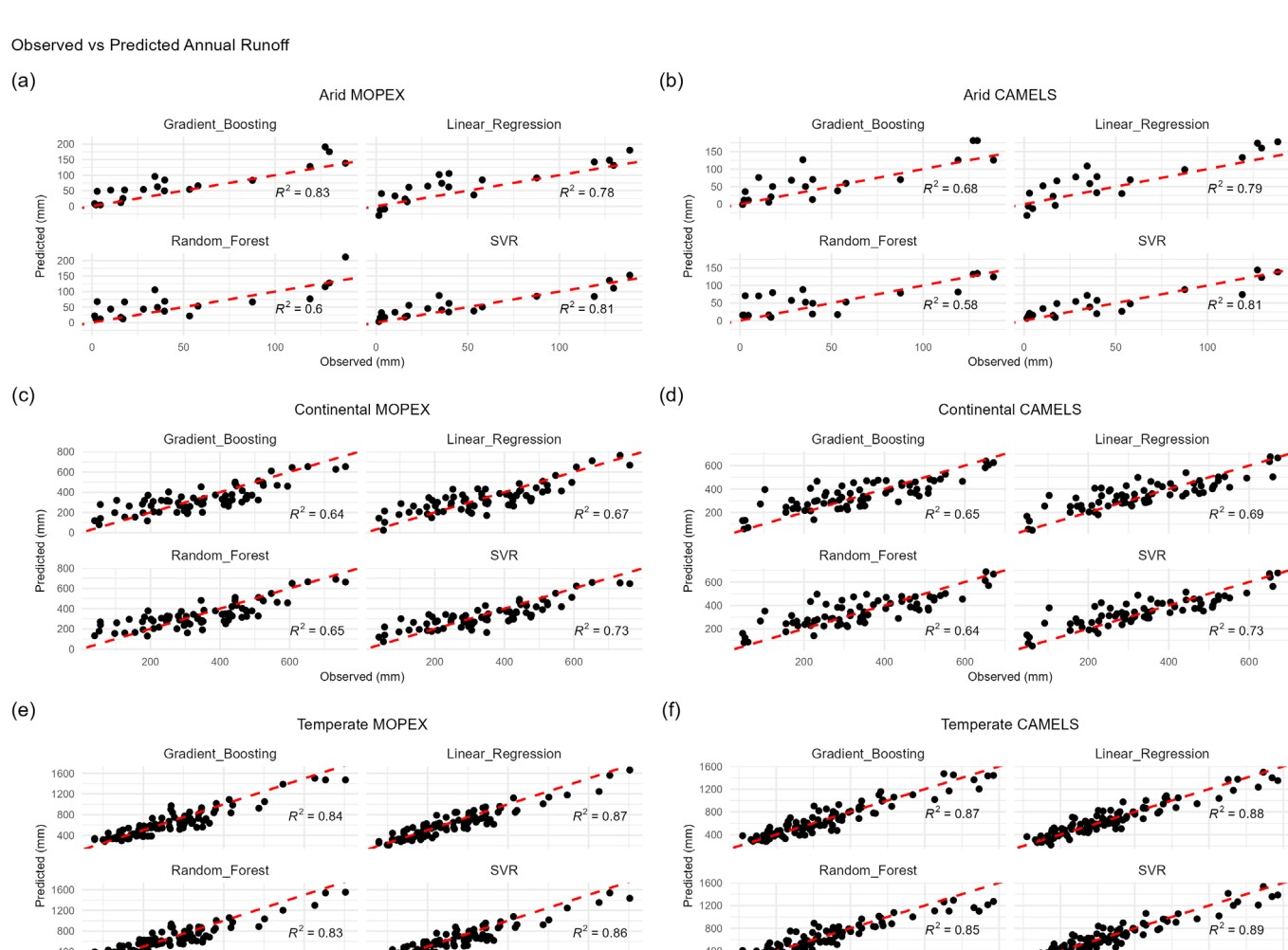

**Figure 27. Observed versus predicted total annual runoff (mm) for each ML model and each climate region. The dashed red line represents the 1:1 reference line.**

In addition to predicting runoff, machine learning was used to differentiate between MOPEX and CAMELS. Data were separated by climate region and then daily, monthly, seasonal, and annual precipitation, temperature, and water balance derived evapotranspiration were used for classifications. The support vector machine performed binary classification, assigning the standardized values to either MOPEX or CAMELS. The classification accuracy values shown in Table 16 represent the model's
ability to identify which dataset the values belong to. If both datasets are considered equal, then the probability of choosing the correct dataset based on a selection of precipitation, temperature, and evapotranspiration values would be 0.5. Accuracy ranges

from 49 to 53 % with most classifications close to 50 %. The model's difficulty to successfully classify the data demonstrates the relative similarity of MOPEX and CAMELS for all climate regions and temporal aggregations.

**Table 16. Classification accuracy results for support vector models.**

|  | Day | Month | Season | Year |
|---|---|---|---|---|
| Arid | 0.517 | 0.490 | 0.495 | 0.480 |
| Continental | 0.505 | 0.521 | 0.529 | 0.523 |
| Temperate | 0.508 | 0.524 | 0.530 | 0.510 |

Annual precipitation similarity was evaluated using the double mass curve (Searcy et al., 1960), a comparative analysis that can identify changes in values over time, examine data consistency, and provide validation (Gao et al., 2017). Cumulative values of two variables plotted against each other should display a linear relationship if the ratio between them is constant. Breaks in the
910 slope can indicate changes in the data and the time it occurred. When cumulative precipitation values are plotted for CAMELS and MOPEX, the slopes are 0.99, 1.06, and 1.07 for arid (Fig. 28a), continental (Fig. 28b), and temperate regions (Fig. 28c) respectively. There are apparent trends in the residuals for continental (Fig. 28e) and temperate (Fig. 28f) regions which could be due to bias, however, the residuals are small, within ± 60 mm.

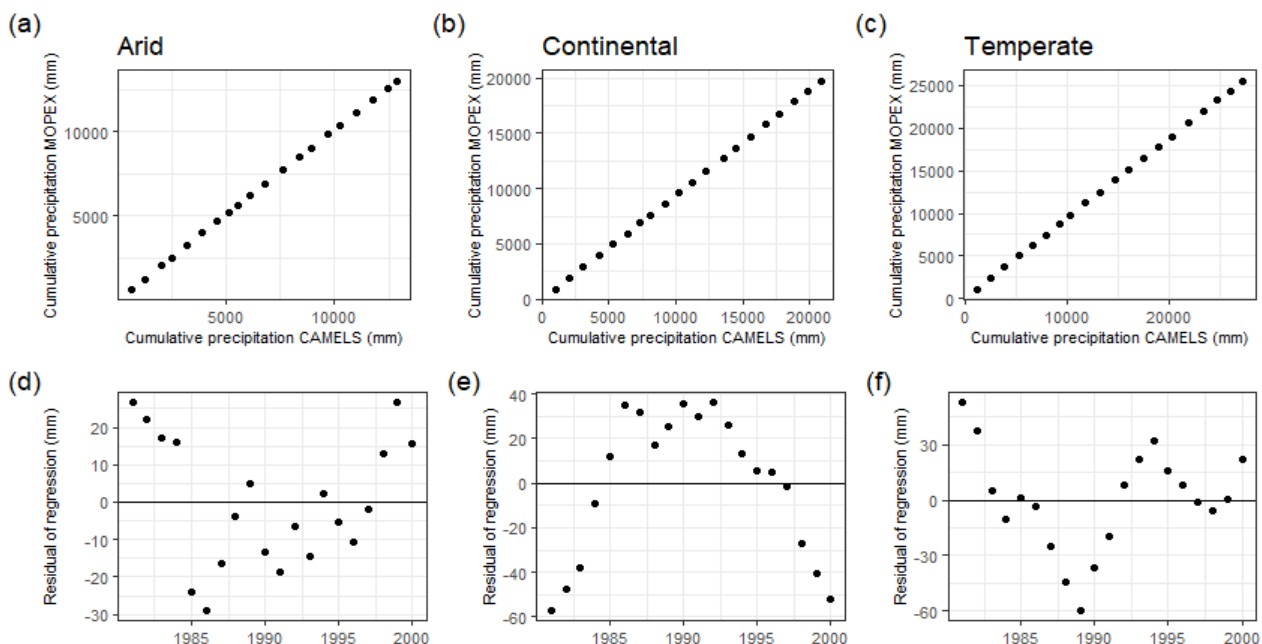

**Figure 28. Cumulative annual precipitation of CAMELS versus MOPEX for each climate region a) arid, b) continental, c) temperate for all water years 1981 – 2000. Residuals of regression are shown for each double mass curve d) arid, e) continental, f) temperate.**

**5 Discussion**

The comprehensive results above indicate important biases between the CAMELS and MOPEX datasets which vary in potential importance by climate region, geographic location and degree of temporal aggregation. The findings underscore the need for
careful consideration of dataset disparities, acknowledging the impact of temporal scale and methodology. As one would expect, comparability between CAMELS and MOPEX increases with greater temporal aggregation. The uncertainty and variability within each dataset were evaluated independently by calculating variance, standard deviation, and mean confidence intervals

with bootstrapping (Tables 4-5, 7-8). The uncertainty and variability between the datasets were evaluated by paired tests, bias, MAE, MOE, $R^2$, and hypothesis tests. By these measures monthly, seasonal, and annual precipitation and temperature values are comparable for MOPEX and CAMELS (Table 9). "Magnitudes of difference or trends within data products may be comparable to the magnitude of difference between data products (Levy et al., 2017)." Essentially, the datasets share similar uncertainties and variances.

Statistically significant differences between variance and mean were more prevalent for precipitation than temperature, however, bootstrapping results indicate that both datasets have similar uncertainties for their mean values, with frequent overlap of the confidence intervals (Tables 7-8). The most substantial differences in mean values are observed in daily aggregations, with discrepancies of 1.13 %, 5.41 %, and 6.36 % for arid, continental, and temperate regions, respectively. At the annual scale, these differences decrease to 0.87 % for arid regions and 4.11 % for temperate regions. Continental regions unexpectedly show an increased difference of 5.54 % at the annual scale, which could be attributed to higher interannual variability or spatial heterogeneity. Median differences in precipitation show significant improvement with temporal aggregation: arid regions decrease from 200 % to 4.88 %, continental regions from 142.86 % to 5.70 %, and temperate regions from 118.37 % to 5.63 %.

Arid regions have the largest margin of error for precipitation, but the smallest percent difference between their daily, monthly, seasonal, and annual mean values. Conversely, for temperature, arid regions have the largest percent difference between mean values in addition to the largest margins of error compared to continental and temperate regions. Correlations improve from 0.57 to 0.67 for daily precipitation values to greater than 0.90 for monthly, seasonal, and annual totals. Temperature correlation values are 0.99 for all regions. Positive correlation between temperature/evapotranspiration and runoff in arid regions is possibly due to increased rates of soil drying and decreased soil moisture retention, which then leads to greater runoff. Intense precipitation events with higher temperatures can also lead to less infiltration and more runoff. Climate patterns, such as rainy or dry seasons, are captured similarly in both datasets and the more consistent pattern may contribute to higher monthly and seasonal correlations compared to annual.

MOPEX has an overall warm bias for all climate regions and a wet bias for arid regions. CAMELS has an overall wet bias for temperate and continental regions. MOPEX is approximately 0.62 ˚C, 0.16 ˚C, and 0.35 ˚C warmer than CAMELS for arid, continental, and temperate regions respectively on all temporal scales. For precipitation, MOPEX is larger than CAMELS by 0.02 mm day$^{-1}$, 0.48 mm month$^{-1}$, 1.45 mm season$^{-1}$, and 5.82 mm year$^{-1}$ for arid regions. CAMELS shows a wet bias for continental, ranging from 0.15 mm day$^{-1}$ to 56.21 mm year$^{-1}$, and temperate, with biases of 0.23 mm day$^{-1}$ to 84.27 mm year$^{-1}$. Other comparison studies have also shown a warm MOPEX bias when compared to Daymet and a wet Daymet bias compared to MOPEX (Essou et al., 2016) and PRISM (Muche et al., 2020). Arid regions have the largest variance and uncertainty which could be due to high evaporation rates, diverse landscapes, or the limited number of stations could influence bootstrapping results, incorporating fewer values compared to temperate or continental landscapes.

Precipitation exhibits a more pronounced positive bias on an annual scale (Fig. 15c), primarily due to the accumulation of small positive biases observed on monthly and seasonal scales. On finer temporal scales, these individual biases may partially offset each other due to seasonality, leading to less noticeable discrepancies yet when data is aggregated annually, consistent overestimations are amplified, resulting in a more evident positive bias. This pattern highlights the presence of a systematic wet bias, where precipitation is consistently overestimated across temporal scales. The effect becomes more apparent at larger aggregation intervals, particularly due to increasing precipitation totals.

For temperature, although larger biases are observed in the 1980s, this pattern is limited to arid regions. Even in later years, the outliers for arid regions remain close to -2 °C. The medians for 1981, 1983, and 1987 are near zero, indicating minimal central tendency bias for those years. When averaged over time, as shown in Fig. 18, there is a slight improvement in biases for arid regions, with a reduction of approximately 0.25 °C from 1981 to 2000. An increase in station density could also contribute to the observed improvement in biases. The GHCNd database, accessed via the National Centers for Environmental Information (NCEI), indicates a noticeable rise in the number of precipitation and temperature stations during the late 1990s and early 2000s. This increased station coverage likely enhanced the spatial representation of observations, reducing biases and improving the accuracy of aggregated data.

Biases have potential consequences for dependent variables like evapotranspiration and runoff. A wet bias in daily precipitation could propagate and inflate the estimates of evapotranspiration and runoff, with hydrological models possibly predicting higher water availability across the system than actually present. Precipitation and temperature data sources differ between MOPEX and CAMELS, resulting in the observed biases, however, statistical analyses and validation in this study demonstrate that the datasets closely resemble one another. Precipitation biases are between -0.25 to 0.54 mm day$^{-1}$ and temperature biases are between -1.88 to 0.27 ˚C day$^{-1}$. These biases exhibit little influence on general hydrologic indices such as runoff efficiency (Fig. 23). Paired same-day comparisons of CAMELS versus MOPEX indicate that the largest discrepancies are between daily values for precipitation and temperature. Combining these datasets is not recommended for evaluating daily events such as maximum 1 day precipitation, however, both datasets exhibit the same general trends for seasonality and climatic indices (Sect. 4.3).

Climate data inherently carries a degree of uncertainty, and the biases described above derive from varying data sources and interpolation of those data to the watershed footprint. Factors such as heterogeneity, gaps in coverage, and interpolation methods contribute to deviations from precision. Data sources can include gauges, satellites, or a combination of both. No two datasets will be identical, as discrepancies occur across various scales. The primary objective is to choose the most appropriate, quality-controlled, accurate, and representative dataset for the research at hand. Previous comparison studies have highlighted the inconsistencies and biases between data products (Levy et al., 2017; Mallakpour and Villarini, 2016; Prat and Nelson, 2015; Sun et al., 2018).

Differences are expected between the datasets based on their sources and processing. Essou et al. (2016) compared three gridded datasets that interpolated daily temperature and precipitation values from the same observation network and noted that "the differences between gridded products may largely be attributed to the interpolation schemes which differ substantially from one dataset to another." A key distinction between the MOPEX and CAMELS datasets lies in their methodological foundations. While both rely on ground-based weather stations as their primary source of meteorological input, they differ in how these observations were processed and spatially interpolated. Daily observations from a network of weather stations are made available by NOAA's National Centers for Environmental Information (NCEI). These data undergo quality assurance checks and processing prior to dissemination. However, uncertainties inherent to station observations remain due to limitations in instrumentation, despite adhere to established standards and calibration protocols, such as those outlined by the National Weather Service (https://www.weather.gov/coop/standards).

MOPEX used observed, gauge-based inputs of precipitation and temperature from Cooperative Observer Program (COOP) and Snowpack Telemetry Network (SNOTEL) weather stations to estimate mean areal values at the catchment scale (Schaake et al.,

2006). For precipitation, MOPEX employed the Mean Areal Precipitation (MAP) methodology developed by the National Weather Service River Forecast Systems (NWSRFS), which combined an inverse distance weighting algorithm with monthly climatological means from PRISM to enhance spatial representativeness (Daly et al., 1993). In contrast, CAMELS derived its meteorological forcing data, specifically precipitation and temperature, from Daymet version 2, a gridded dataset that interpolates and extrapolates surface observations from Global Historical Climatology Network (GHCN) stations, including those from COOP (Thornton et al., 2014). Daymet version 2 employs a Gaussian convolution kernel interpolation method to produce spatially and temporally consistent values across the CONUS. These gridded values were then spatially averaged to the catchment scale within CAMELS (Newman et al., 2015).

Evapotranspiration values in CAMELS are estimated via the SAC-SMA hydrological model and as our results highlight, these model-based ET values can sometimes produce implausible behavior – such as overestimation or ET demand exceeding available precipitation. As MOPEX does not provide ET data, it can be calculated as a water balance residual. This approach benefits from empirical grounding, particularly in well-instrumented basins, but may suffer in regions with poor data quality or significant anthropogenic influences that are not explicitly accounted for in the water balance in addition to the uncertainties present in precipitation and runoff estimates (Carter et al., 2018).

MOPEX provides a longer historical record, which is valuable for evaluating long-term trends and hydroclimate variability. CAMELS is particularly well-suited for regional-scale hydrological analyses and climate sensitivity studies, especially in areas where gauge coverage is minimal or where spatial variability in meteorology is high (Addor et al., 2017; Newman et al., 2015). For spatially extensive or gridded analyses where consistency and meteorological realism are priorities, CAMELS offers advantages. Thus, when using daily data, the choice between CAMELS and MOPEX depends on the application.

**6 Conclusions**

In this study, we evaluated two large sample datasets, MOPEX and CAMELS, comparing precipitation and temperature. The current MOPEX dataset contains data for 431 watersheds within the CONUS while CAMELS includes data for 671 watersheds. The datasets were combined for this study and 47 common basins were compared for water years 1981 through 2000 on daily, monthly, seasonal, and annual scales. Precipitation, temperature, and streamflow data are areally weighted by delineated boundaries available as shapefiles. The main conclusions from the statistical comparison between CAMELS and MOPEX at daily, monthly, seasonal, and annual scales are summarized as follows:

1) The relevance of differences between MOPEX and CAMELS depends on the study's scale and purpose.
2) Daily pairwise comparisons are not recommended due to the variability in extreme precipitation event measurements. However, both datasets capture similar patterns and basin behavior, for example, when evaluating the number of rainy days or dry days per year.
3) Comparison improves significantly with monthly, seasonal, and annual aggregations. Despite temperature and precipitation biases, MOPEX and CAMELS show similar predicted runoff at the annual scale, requiring no raw data corrections. Monthly, seasonal, and annual values are comparable, as their differences are within expected uncertainty ranges.

4) Compatibility is constrained by basin water balance and requires basin averaged values, i.e. ET values from model output CAMELS time series must be used with caution, and often cannot be reconciled with MOPEX or other water-balance based estimates

5) All modeling results should include uncertainty estimates. Bias correction is typically performed during calibration, addressing dataset specific biases.

The comparative analysis of the MOPEX and CAMELS datasets reveals distinct biases and variability patterns across climate regions and temporal scales. CAMELS generally exhibits a positive precipitation bias at monthly, seasonal, and annual aggregations in continental and temperate regions, while MOPEX shows higher daily precipitation values, particularly in arid regions. Temperature analysis highlights a consistent warm bias in MOPEX across all regions and time scales, with notable disparities in daily values. Despite these differences, the datasets show overlapping confidence intervals for many metrics, suggesting similar levels of uncertainty.

The observed variations, particularly for extreme precipitation events, underscore the necessity for cautious interpretation of dataset-specific results. For applications requiring precision, such as hydrological modeling or climate analysis, direct substitution of daily values between MOPEX and CAMELS is not advisable without considering these biases. Instead, leveraging insights from both datasets can provide a more comprehensive understanding of regional and temporal climate characteristics.

Ongoing research aims to extend the MOPEX dataset from 2003 to 2023 and the CAMELS dataset from 2014 to 2023, leveraging the Daymet dataset (Thornton et al., 2021). MOPEX and CAMELS will be integrated into a cohesive resource that combines catchment attributes and human impact classifications based on the GAGESII framework (Falcone et al., 2010). The enhanced dataset will support model calibration and freshwater balance studies at the watershed scale (in progress, Sink et al.). Basin-scale analyses and forecasts are expected to benefit from more precise water balance constraints, improving their accuracy and predictive power.

*Code and data availability.* A repository with R Code used for analyses and resulting data is available from https://github.com/k-sink/Toward_merging. Colors are based on https://jfly.uni-koeln.de/color/#pallet by Okabe-Ito (2008). Although the MOPEX dataset (Schaake et al., 2006) was publicly available at the time of analysis, recent attempts to access the data at http://hydrology.nws.noaa.gov/pub/gcip/-mopex/US_Data were unsuccessful. The CAMELS dataset (Addor et al., 2017) is available from https://gdex.ucar.edu/dataset/camels.html. Streamflow data and attributes are available from https://waterdata.usgs.gov/nwis/inventory.

*Author contributions*. KS and TB conceptualized the research project. KS developed and performed the formal analysis. KS prepared the paper with contributions from the co-author.

*Competing interests*. The authors declare that they have no conflict of interest.

*Acknowledgements*. Thank you to my advisor, Tom Brikowski, for his helpful discussions and comments during this research project. Data processing and visualization were performed with R (R Core Team, 2024). UTD SESS contribution #1728.

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
