# Peer review of "Toward merging MOPEX and CAMELS hydrometeorological datasets: compatibility and statistical comparison"

_EGUsphere, 2024_

## Author Comment (AC1)

**Comments by RC1, Anonymous Referee #1 in blue, our responses in black.**

Overall, this is an interesting study comparing two commonly utilized catchment data sources. The analysis for the continental US (CONUS) appears to demonstrate statistically significant differences in the aggregate regarding temperature and precipitation. The differences shown are important, however greater attention to explaining the differences, and their significance would significantly improve the manuscript. The use of machine learning is not clearly articulated in the work and its significance is not yet clear. Greater attention should be paid to discussing the impacts of these differences on future modeling efforts as well.

**Thank you for your thoughtful and constructive feedback on our manuscript. We have carefully considered each of your suggestions and have made the following revisions accordingly. Specific changes:**

The manuscript would also benefit from a clear statement of the goals of the research, i.e. is the goal to show that the two data sets are equivalent and therefore can be merged? Or is to identify where the two data sets differ and to explain why they are different, with the goal of adjusting one, or the other to allow merging? See line 45 for the first time this is made clear in the text. I would suggest clearly stating this in the abstract as well

**The abstract was revised to read**

This study compares two large hydrometeorological datasets, the Model Parameter Estimation Experiment (MOPEX), and the Catchment Attributes and Meteorology for Large-sample Studies (CAMELS), with the aim of quantifying differences that might impact their mergers. This comparison focuses on 47 shared watersheds within the continental United States spanning daily, monthly, seasonal, and annual scales for the overlapping water years of 1981-2000. Results indicate significant differences between the datasets at daily timesteps, highlighting the challenge of high temporal resolution data reconciliation; however, compatibility markedly improves with temporal aggregation at monthly, seasonal, and annual scales. Systematic biases are evident, with MOPEX showing a warm bias for temperature and CAMELS displaying a wet bias for precipitation. Studies considered monthly or longer trends may use either or both datasets, as the biases are not significant for low-temporal resolution analyses. Studies focusing on high resolution hydrological characteristics, such as daily precipitation events, the frequency of wet and dry days per month or single-basin dynamics, may require a statistical bias correction to ensure accuracy. The variability between the datasets is comparable to the variability within each dataset and is neither a useful criterion for dataset selection nor a barrier to potential merger. In any case, model outputs should be calibrated against observational refence data to account for systematic errors. Statistical analyses demonstrate that both datasets are representative of climatic conditions, trends, and extreme events. Our findings validate the results of previous research employing either dataset. Furthermore, this study serves as a foundation for the merging and extension of MOPEX and CAMELS datasets without any alterations, providing a comprehensive, long-term dataset suitable for hydrological modelling and climate analyses while maintaining comparability across basin and temporal scales.

**Beginning at line 56, after the sentence ending with "watershed-based datasets" we removed the remaining text (lines 56 to 59) and clarified the study to read as follows:**

Our findings show that while MOPEX and CAMELS exhibit systematic biases, they can still be merged or reliably compared without requiring corrections beyond smaller time scales (i.e. a single day, month, or season). Statistical adjustments to daily data depend on study objectives, as no single method fits all needs. Raw data or direct model outputs typically require bias correction, and we intend for our results to help researchers determine necessary adjustments using appropriate methods, including equidistant quantile matching (EDCDFm) for temperature and quantile delta mapping (QDM) or PresRATe for precipitation (Lehner et al., 2023a; Pierce et al., 2015).

**Line 103. How will this study address "uncertainties within the data sets? This is an unclear statement**

**Thank you for your comment. We have clarified the statement as follows**

This study provides researchers with detailed analyses regarding the uncertainties within the datasets and between them for a 20-year period through quantitative measurements of dispersion, distribution, central tendency, interval estimates, and statistical tests.

**Lines 140-150. This is a confusing paragraph for those not intimately familiar with either data set. You state there are large discrepancies between the CAMEL SAC model ET and CAMEL-WB. Why is this important when comparing CAMELS to MOPEX, the goal of this work? Please expand this section and make it clear why these differences in ET with CAMELS is important to the goal of this work.**

**Thank you for the suggestion. We have expanded the section and included two additional figures to aid in clarification. The following text was added immediately after Figure 2, beginning at line 140. The addition also includes two figures.**

Terrestrial evapotranspiration is difficult to measure directly but can be evaluated using lysimeters or eddy covariance towers on local scales. Evapotranspiration can be estimated on a larger scale using satellite remote sensing or land surface models but these carry with them inherent biases due to varying algorithms, spatial resolutions, calibration, and input data (Long et al., 2014). Many studies have shown that derived ET products fail to reconcile the terrestrial water budget on multiple temporal scales (Carter et al., 2018). A water balance approach is commonly used on a catchment scale and with observed streamflow obtained from a measured outlet (Han et al., 2015). A water balance sets ET (mm) equal to the precipitation (mm) minus basin runoff (mm), with water storage and net groundwater flux assumed to be zero on an annual scale.

The MOPEX dataset does not contain daily ET. Studies that have made use of MOPEX data obtain ET via the water balance approach using the precipitation and observed runoff (Berghuijs et al., 2014; Coopersmith et al., 2012; Sawicz et al., 2014). As mentioned previously, CAMELS provides three different daily forcing datasets (Daymet, Maurer, NLDAS), which do not contain ET, in addition to three Sacramento Soil Moisture and Accounting Model (SAC-SMA) generated time series from each of the forcing datasets. In this study, daily ET values from the model output time series using Daymet forcing variables (CAMELS SAC-SMA) were compared to the water balance derived ET (CAMELS WB) using CAMELS catchment averaged Daymet precipitation and USGS runoff values to

evaluate any notable differences between methods and facilitate comparison of MOPEX and CAMELS. The CAMELS SAC-SMA model derived ET values are typically greater than the values derived from the CAMELS WB, which become more prominent at an annual scale, as plotted in Fig. 3. When the individual annual differences between CAMELS SAC-SMA model estimated ET and CAMELS WB estimated ET are averaged together, SAC-SMA model estimations are approximately 13 mm larger in arid regions (Fig. 3a), 36 mm larger in continental regions (Fig. 3b), and 50 mm larger in temperate regions (Fig. 3c).

[Figure]

**Figure 3. Total annual evapotranspiration for a) arid, b) continental, and c) temperate regions. The annual values are the overall mean of all basin totals in a region. The model output ET (CAMELS SAC-SMA), water balance derived MOPEX ET (MOPEX), and water balance derived CAMELS ET (CAMELS WB) are shown in each plot.**

Higher ET values lead to reduced runoff. As shown in Fig. 4, estimated ET values from CAMELS SAC-SMA model were subtracted from the provided CAMELS (Daymet) precipitation data to calculate estimated runoff (SAC_RUN), which was then compared to observed runoff (OBS_RUN). Incorporating ET values from the model output time series as an input variable to a hydrologic

model may result in slightly lower discharge estimates, primarily reflecting the influence of ET values rather than actual runoff conditions.

[Figure]

**Figure 4. Total annual runoff for a) arid, b) continental, and c) temperate regions. The boxplots represent the annual totals for all basins in a region, with measured observed runoff (OBS_RUN) and water balance calculated runoff (PRCP minus ET) using CAMELS SAC-SMA model output ET (SAC_RUN).**

**Figure 3 (page 7) was renumbered to Figure 5 and the subsequent figures in the manuscript and captions were updated to reflect the insertion of the new figures.**

**The text resumes with the original line 140 "A Budyko diagram …" and ends with "4.88% in continental regions from line 153. The text was slightly modified to read as follows**

A Budyko diagram, plotting evaporative versus aridity indices, clarifies the predominant hydrologic processes versus climate type (Fig. 5) for the common basins. The CAMELS SAC-SMA evapotranspiration (solid symbols, Fig. 5) exhibits large discrepancies from CAMELS-WB (open symbols, Fig. 5) and MOPEX ET values for most catchments (arrows, Fig. 5). Furthermore, several CAMELS SAC-SMA gauges plotted above the water limit (i.e. to extreme values in the Budyko context) and were 10 to 12 % larger than the water-balance-calculated evapotranspiration indices. The higher SAC-SMA model-derived ET for CAMELS could reflect additional non-precipitation sources of water to the catchment, but that was not evaluated in this study. The largest discrepancies between model-derived ET/P and water balance derived ET/P for CAMELS for each climate region are 46 %, 12 %, and 60 % for arid, continental, and temperate regions respectively.

Average discrepancies for both CAMELS evapotranspiration values are largest in arid regions, 12 %, followed by average discrepancies of 11 % in temperate regions, and 5 % in continental regions.

**The next portion of the sentence in line 153 "Studies that have made use of MOPEX data obtain evapotranspiration via the water balance approach" was deleted and relocated to the inserted text presented on page 3 of this response document.**

**The text then continues with "Differences between …" from line 160.**

**The remainder of the original sentence on line 155 beginning with "further research …" and ending with "snowmelt is negligible." on line 158 was moved to immediately after the original Figure 3 (now figure 5), concluding Section 3.1**

Tables 4 and 5: These tables need far more explaining. The text indicates that they are internal variability of the two data sets, yet in each case, only a single mean is presented. The text is unclear as the tables do not provide the reader with any form of comparison here. The text indicates "within" the data sets, but the tables appear to provide "between" the data sets. Please expand section 4.1 to be clearer here.

**Thank you for your comment. We agree that this could be clarified, and we have made changes to assist the reader with comparisons. After the end of the sentence on line 251 ("both within and between them."), we modified the remaining text for Section 4 to read:**

By considering multiple statistics, we can evaluate the representativeness of each dataset and identify any systematic differences that may need further investigation. If both datasets exhibit similar means and variability within a climate region, it suggests that their distributions are comparable. Differences in variance and skewness, on the other hand, highlight potential biases between the datasets. Though there are consistent biases, they are minimal for aggregations beyond a daily time step, making them suitable for combined application in climate studies and hydrologic modelling at monthly, seasonal, or annual aggregations. Although efforts were made to distinguish results for internal analyses within datasets and intercomparisons between datasets, the results are often presented together to provide a clearer understanding of how each dataset behaves independently, while also enabling direct cross-dataset evaluation. Consequently, some overlap does occur.

**We have relocated the original tables 4 and 5 to Section 4.2 because they coincide with the other results for between datasets. Original tables 4 and 5 (renumbered to 7 and 8) are now located at line 506, immediately after the caption for original figure 17. All tables have been renumbered accordingly in the captions and manuscript text. The text preceding the relocated tables, beginning at line 506 reads:**

Overall statistics for precipitation are shown in Table 7 and were calculated over all shared basins within a climate region. Temperature statistics are shown in Table 8. The mean values and corresponding confidence intervals are based on the averages derived from bootstrapping results, shown in Figs. 7-9 for monthly, seasonal, and annual precipitation values and Figs. 10-12 for monthly, seasonal, and annual temperature values. The tables highlight the commensurate central tendencies, variabilities, and dispersion values within the datasets and provide insight into the comparisons between the datasets.

**The manuscript resumes after the inclusion of the relocated tables (original tables 4 and 5 now tables 7 and 8) with original text beginning on line 506 "To assess the magnitude…"**

**We have included two new tables, now Tables 4 and 5 that show the internal variability of the datasets with the minimum and maximum values for each statistic. These tables show the range of median, mean, variance, standard deviation, and skewness values for the basins within each climate region.  We have edited the text in Section 4.1 (removing original lines 254 through 260) so that it now begins on line 254 with the following:**

[revised manuscript text omitted]

**Manuscript resumes with the precipitation coefficient of variation figure, now Figure 6.**

**Line 272. Does the fact that averaging over greater temporal scales reduce the dispersion a major finding here? it would seem like this would be an expected result?.**

**We agree and have removed the sentence.**

**Line 323. It's not surprising that the variation in arid region precipitation is greater but what does " remain the most consistent" in the text mean? Consistent between data sets? Please be specific.**

**Thank you for your comment. We have modified the portion of the text to now read:**

For average total annual precipitation, arid regions exhibit the highest variability within each individual dataset, however, their overall temporal mean values remain similar between the two datasets

**Line 375: Some discussion of why these differences exist would be valuable here. . A bit of speculation will be helpful and appropriate.**

**Thank you for your suggestion. Our objective is to present the results as concisely as possible and reserve our speculations for the discussion section. See discussion beginning on line 668. We hope to maintain the structure of our paper. We also added significantly to our machine learning section and indicated below.**

**Line 630, Section 4.4  It is not fully apparent why machine learning validation was undertaking for this work and how it helps in the analysis.  Please justify its use in more clarity.**

**Thank you for this comment. We have reworked Section 3.3 to explain the reasoning behind machine learning validation. We also included additional analyses to provide further support for this form of validation. The section now reads as follows:**

[revised manuscript text omitted]

Domingos, P. (2012). A few useful things to know about machine learning. In *Communications of the ACM* (Vol. 55, Issue 10, pp. 78–87). https://doi.org/10.1145/2347736.2347755

Herrera, P. A., Marazuela, M. A., & Hofmann, T. (2022). Parameter estimation and uncertainty analysis in hydrological modeling. In *Wiley Interdisciplinary Reviews: Water* (Vol. 9, Issue 1). John Wiley and Sons Inc. https://doi.org/10.1002/wat2.1569

Kratzert, F., Klotz, D., Herrnegger, M., Sampson, A. K., Hochreiter, S., & Nearing, G. S. (2019). Toward Improved Predictions in Ungauged Basins: Exploiting the Power of Machine Learning. *Water Resources Research*, *55*(12), 11344–11354. https://doi.org/10.1029/2019WR026065

Lehner, F., Nadeem, I., & Formayer, H. (2023). Evaluating skills and issues of quantile-based bias adjustment for climate change scenarios. *Advances in Statistical Climatology, Meteorology and Oceanography*, *9*(1), 29–44. https://doi.org/10.5194/ascmo-9-29-2023

Long, D., Longuevergne, L., & Scanlon, B. R. (2014). Uncertainty in evapotranspiration from land surface modeling, remote sensing, and GRACE satellites. *Water Resources Research*, *50*(2), 1131–1151. https://doi.org/10.1002/2013WR014581

Nearing, G. S., Kratzert, F., Sampson, A. K., Pelissier, C. S., Klotz, D., Frame, J. M., Prieto, C., & Gupta, H. V. (2021). What Role Does Hydrological Science Play in the Age of Machine Learning? In *Water Resources Research* (Vol. 57, Issue 3). Blackwell Publishing Ltd. https://doi.org/10.1029/2020WR028091

Pendergrass, A. G., Knutti, R., Lehner, F., Deser, C., & Sanderson, B. M. (2017). Precipitation variability increases in a warmer climate. *Scientific Reports*, *7*(1). https://doi.org/10.1038/s41598-017-17966-y

Pierce, D. W., Cayan, D. R., Maurer, E. P., Abatzoglou, J. T., & Hegewisch, K. C. (2015). *Improved Bias Correction Techniques for Hydrological Simulations of Climate Change\**. https://doi.org/10.1175/JHM-D-14-0236.s1

Xu, T., & Liang, F. (2021). Machine learning for hydrologic sciences: An introductory overview. In *Wiley Interdisciplinary Reviews: Water* (Vol. 8, Issue 5). John Wiley and Sons Inc. https://doi.org/10.1002/wat2.1533

Yokoo, K., Ishida, K., Ercan, A., Tu, T., Nagasato, T., Kiyama, M., & Amagasaki, M. (2022). Capabilities of deep learning models on learning physical relationships: Case of rainfall-runoff modeling with LSTM. *Science of the Total Environment*, *802*. https://doi.org/10.1016/j.scitotenv.2021.149876

Zhang, W., Furtado, K., Wu, P., Zhou, T., Chadwick, R., Marzin, C., Rostron, J., & Sexton, D. (2021). Increasing precipitation variability on daily-to-multiyear time scales in a warmer world. In *Sci. Adv* (Vol. 7). https://www.science.org

---

## Author Comment (AC2)

**Comments by RC2, Anonymous Referee #2 in blue, our responses in black.**

This manuscript presents a detailed comparison between two widely used streamflow and meteorological datasets for the continental United States, MOPEX and CAMELS, investigating their consistency and discrepancies from daily to annual scales. The study is based on a carefully designed statistical analysis and is relevant to the hydrological modeling and large-sample hydrology communities. The work is rigorous, and the results are clearly communicated and well discussed. I have a few remarks and suggestions for improvement that the authors might find useful.

**Thank you for your insightful feedback on our manuscript. We have carefully considered each of your suggestions and have made the following revisions accordingly. Specific changes:**

In the abstract and elsewhere, the term 'bias' is used to describe the differences between MOPEX and CAMELS. Since bias is typically defined with respect to a reference or ground truth, it would be helpful to clarify that this refers to relative bias (i.e., systematic differences between datasets), rather than absolute error. While this becomes clearer within the manuscript, the abstract might mislead readers into thinking that MOPEX is definitively too warm or CAMELS too wet.

**Thank you for your comment. We have made the necessary updates to the abstract and text, indicating that the bias is systematic for clarification.**

The manuscript could benefit from a more in-depth discussion of which dataset may be more reliable under certain conditions. Lines 685–687 touch upon this subject but could be expanded. For instance, CAMELS uses Daymet meteorological forcing, which could be potentially considered more reliable for regional hydrological analyses. However, its evapotranspiration values are derived from the SAC-SMA hydrologic model and, as the authors show, can exhibit implausible behavior. These trade-offs, i.e., between modern gridded meteorological inputs and model-based ET estimates, deserve a more explicit discussion to help guide dataset selection for different hydrological applications.

**Thank you for your suggestions. The final paragraph in the discussion section was edited. The three sentences beginning on line 725 with "MOPEX estimated daily mean" were replaced with:**

A key distinction between the MOPEX and CAMELS datasets lies in their methodological foundations. While both rely on ground-based weather stations as their primary source of meteorological input, they differ in how these observations were processed and spatially interpolated. Daily observations from a network of weather stations are made available by NOAA's National Centers for Environmental Information (NCEI). These data undergo quality assurance checks and processing prior to dissemination. However, uncertainties inherent to station observations remain due to limitations in instrumentation, despite adherence to

established standards and calibration protocols, such as those outlined by the National Weather Service (https://www.weather.gov/coop/standards).

MOPEX used observed, gauge-based inputs of precipitation and temperature from Cooperative Observer Program (COOP) and Snowpack Telemetry Network (SNOTEL) weather stations to estimate mean areal values at the catchment scale (Schaake et al., 2006). For precipitation, MOPEX employed the Mean Areal Precipitation (MAP) methodology developed by the National Weather Service River Forecast Systems (NWSRFS), which combined an inverse distance weighting algorithm with monthly climatological means from PRISM to enhance spatial representativeness (Daly et al., 1993). In contrast, CAMELS derived its meteorological forcing data, specifically precipitation and temperature, from Daymet version 2, a gridded dataset that interpolates and extrapolates surface observations from Global Historical Climatology Network (GHCN) stations, including those from COOP (Thornton et al., 2014). Daymet version 2 employs a Gaussian convolution kernel interpolation method to produce spatially and temporally consistent values across the CONUS. These gridded values were then spatially averaged to the catchment scale within CAMELS (Newman et al., 2015).

Evapotranspiration values in CAMELS are estimated via the SAC-SMA hydrological model and as our results highlight, these model-based ET values can sometimes produce implausible behavior – such as overestimation or ET demand exceeding available precipitation. As MOPEX does not provide ET data, it can be calculated as a water balance residual. This approach benefits from empirical grounding, particularly in well-instrumented basins, but may suffer in regions with poor data quality or significant anthropogenic influences that are not explicitly accounted for in the water balance in addition to the uncertainties present in precipitation and runoff estimates (Carter et al., 2018).

MOPEX provides a longer historical record, which is valuable for evaluating long-term trends and hydroclimate variability. CAMELS is particularly well-suited for regional-scale hydrological analyses and climate sensitivity studies, especially in areas where gauge coverage is minimal or where spatial variability in meteorology is high (Addor et al., 2017; Newman et al., 2015). For spatially extensive or gridded analyses where consistency and meteorological realism are priorities, CAMELS offers advantages. Thus, when using daily data, the choice between CAMELS and MOPEX depends on the application.

**Line 725: Please provide a citation for the NCDC COOP and SNOTEL datasets used in MOPEX. Additionally, a brief explanation of the nature of these data sources, including their observational basis and common sources of uncertainty, would help readers better understand the reliability and limitations of the meteorological data used in these databases.**

**We have added the citations which are indicated below along with a brief description of COOP and SNOTEL, that is incorporated into the previous response above.**

Operated by National Resources Conservation Service, US Department of Agriculture. Snowpack Telemetry Network (https://www.wcc.nrcs.usda.gov/factpub/sntlfct1.html).

Wuertz, David; Lawrimore, Jay; and Korzeniewski, Bryant (2018). Cooperative Observer Program (COOP) Hourly Precipitation Data (HPD). NOAA National Centers for Environmental Information. doi:10.25921/p7j8-2170.

**Figure 2: Could the authors clarify the meaning of the blue color in the map? It's not evident from the caption or figure description.**

**We have added the following to the caption of Figure 2**

The blue colors in southern Florida represent regions within the tropical climate group, which is not represented in this study.

**Section 3.2.2: Please include references for all the statistical tests used (e.g., Fligner-Killeen test, Welch's t-test).**

**References were added for all applicable statistical tests.**

Aho, K.A. (2013). Foundational and applied statistics for biologists using R. Chapman & Hall/CRC Press, 596 p.

Fligner, M.A. and Killeen, T.J. (1976). Distribution-free two-sample tests for scale. Journal of the American Statistical Association, 71(353), 210-213. https://doi.org/10.1080/01621459.1976.10481517

Helsel, D.R., Hirsch, R.M., Ryberg, K.R., Archfield, S.A., and Gilroy, E.J. (2020). Statistical methods in water resources. USGS Techniques and Methods 4 – A3.

Welch, B.L. (1951). On the comparison of several mean values: an alternative approach. Biometrika, 38(3-4), 330-336. https://doi.org/10.1093/biomet/38.3-4.330

---

## Author Response (AR1)

Line numbers refer to the updated revision pdf with tracking changes enabled, not the original submission. The content of the added text, tables, and figures were detailed in our individual responses to the referee comments. We did not include them again in this response for brevity's sake.

We sincerely want to thank the reviewers and editor for their time and comments. Your suggestions have greatly contributed to the improvement of our manuscript.

Comments by RC1, RC2 in blue, our responses in black

**Anonymous Referee #1 Comments**

Overall, this is an interesting study comparing two commonly utilized catchment data sources. The analysis for the continental US (CONUS) appears to demonstrate statistically significant differences in the aggregate regarding temperature and precipitation. The differences shown are important, however greater attention to explaining the differences, and their significance would significantly improve the manuscript. The use of machine learning is not clearly articulated in the work and its significance is not yet clear. Greater attention should be paid to discussing the impacts of these differences on future modeling efforts as well.

The manuscript would also benefit from a clear statement of the goals of the research, i.e. is the goal to show that the two data sets are equivalent and therefore can be merged? Or is to identify where the two data sets differ and to explain why they are different, with the goal of adjusting one, or the other to allow merging? See line 45 for the first time this is made clear in the text. I would suggest clearly stating this in the abstract as well

- Revised the abstract to include clear statement of the goals.
- Edited and added text beginning at line 70 to address the results of the study along with the goals.

Line 103. How will this study address "uncertainties within the data sets? This is an unclear statement

- Line 129 clarifies the statement regarding the methodology to address uncertainties.

Lines 140-150. This is a confusing paragraph for those not intimately familiar with either data set. You state there are large discrepancies between the CAMEL SAC model ET and CAMEL-WB. Why is this important when comparing CAMELS to MOPEX, the goal of this work? Please expand this section and make it clear why these differences in ET with CAMELS is important to the goal of this work.

- Added text beginning at line 168 to line 184 to expand the section and clarify the differences in evapotranspiration in the CAMELS dataset, the modeled ET and the water balance derived ET.

- Added Figure 3 at line 184 to depict the differences in annual ET for CAMELS model and water balance derived along with MOPEX water balance ET for comparison.
- Added text at line 190 to 196 to explain the new figures 3 and 4.
- Added Figure 4 at line 196 to demonstrate the differences between the modeled ET when used to calculate the runoff (Precipitation – evapotranspiration = runoff) compared to the observed, measured runoff from the USGS.
- Text beginning at line 205 was edited to clarify the depictions in Figure 5.
- Text on lines 216 – 221 was moved to line 239 – 242.

**Tables 4 and 5: These tables need far more explaining. The text indicates that they are internal variability of the two data sets, yet in each case, only a single mean is presented. The text is unclear as the tables do not provide the reader with any form of comparison here. The text indicates "within" the data sets, but the tables appear to provide "between" the data sets. Please expand section 4.1 to be clearer here.**

- Added text at line 360 – 363 and at lines 365 – 368 to clarify the comparisons.
- Added Table 4 at line 389 to include the minimum, maximum and ranges for the median, mean, variance, standard deviation, and skewness for precipitation for both datasets. This table shows the internal variability of the datasets.
- Added text at lines 370 – 380 and lines 394 – 409 to explain results in Table 4.
- Added text at line 411 – 425 to explain results in Table 5.
- Added Table 5 at line 445 to include minimum, maximum, and ranges for median, mean, variance, standard deviation, and skewness for temperature for both datasets. This table shows the internal variability of the datasets.
- Removed sentence at line 452.
- Added text at lines 457 – 461 to explain Table 4 and its relationship with the coefficient of variation results for precipitation.
- Added line 469 to clarify the confidence interval estimates.
- More descriptive text added at lines 564 and 568.
- Text at 694 – 698 was added to explain Tables 7 and 8 (previously Tables 4 and 5), which were relocated to line 699 after the addition of new Tables 4 and 5.

**Line 272. Does the fact that averaging over greater temporal scales reduce the dispersion a major finding here? it would seem like this would be an expected result?.**

- Now line 452-453, was removed.

**Line 323. It's not surprising that the variation in arid region precipitation is greater but what does " remain the most consistent" in the text mean? Consistent between data sets? Please be specific.**

- Modified lines 508-509 for clarity.

**Line 375: Some discussion of why these differences exist would be valuable here. . A bit of speculation will be helpful and appropriate.**

- Added to discussion and machine learning, indicated below.

**Line 630, Section 4.4  It is not fully apparent why machine learning validation was undertaking for this work and how it helps in the analysis.  Please justify its use in more clarity.**

- Added significantly to Section 3.3 to include further machine learning validation. Outlines the procedures.
- Text added at lines 300 – 337 to explain the machine learning models and simulated runoff using linear regression, random forest, gradient boosting, and support vector regression.
- Lines 339 – 342 removed.
- Lines 346 – 351 added and edited to explain binary classification.
- Modified Section 4.4 begins at line 836. Added new text, Tables 12, 13, 14 and 15 to present results from the 4 machine learning models and the predicted compared to observed runoff results.
- Added Figure 27 at line 892 which depicts the predicted vs observed runoff values and the 1:1 reference line for each of the four machine learning models for CAMELS and MOPEX.
- Numbered points in Conclusion beginning at line 1035 were renumbered and revised to include more defined results of the study.

**Anonymous Referee #2 Comments**

This manuscript presents a detailed comparison between two widely used streamflow and meteorological datasets for the continental United States, MOPEX and CAMELS, investigating their consistency and discrepancies from daily to annual scales. The study is based on a carefully designed statistical analysis and is relevant to the hydrological modeling and large-sample hydrology communities. The work is rigorous, and the results are clearly communicated and well discussed. I have a few remarks and suggestions for improvement that the authors might find useful.

In the abstract and elsewhere, the term 'bias' is used to describe the differences between MOPEX and CAMELS. Since bias is typically defined with respect to a reference or ground truth, it would be helpful to clarify that this refers to relative bias (i.e., systematic differences between datasets), rather than absolute error. While this becomes clearer within the manuscript, the abstract might mislead readers into thinking that MOPEX is definitely too warm or CAMELS too wet.

- Modified abstract incorporates these comments.

**The manuscript could benefit from a more in-depth discussion of which dataset may be more reliable under certain conditions. Lines 685–687 touch upon this subject but could be expanded. For instance, CAMELS uses Daymet meteorological forcing, which could be potentially considered more reliable for regional hydrological analyses. However, its evapotranspiration values are derived from the SAC-SMA hydrologic model and, as the authors show, can exhibit implausible behavior. These trade-offs, i.e., between modern gridded meteorological inputs and model-based ET estimates, deserve a more explicit discussion to help guide dataset selection for different hydrological applications.**

- Removed lines 994-997 and added lines 997-1025 to discussion section to expand.

**Line 725: Please provide a citation for the NCDC COOP and SNOTEL datasets used in MOPEX. Additionally, a brief explanation of the nature of these data sources, including their observational basis and common sources of uncertainty, would help readers better understand the reliability and limitations of the meteorological data used in these databases.**

- Citations added and discussion incorporated into lines 997-1025.

**Figure 2: Could the authors clarify the meaning of the blue color in the map? It's not evident from the caption or figure description.**

- Added explanation to figure caption.

**Section 3.2.2: Please include references for all the statistical tests used (e.g., Fligner-Killeen test, Welch's t-test).**

- Added references

*Additional points*

- Renumbered all figures/tables and captions to reflect the updated numbering.
- Fixed minor typos and punctuation.